# Clipped Q-Learning: Your Value Clipping Is Secretly A Robust Operator

**Zhishuai Liu** [1]   **Pan Xu** [1]

## Abstract

We study a simple yet principled modification of classical Q-learning that clips the value estimate in the Bellman backup at a threshold $\lambda$. The resulting algorithm, Clipped Q-Learning, is motivated by a key theoretical insight: the clipped Bellman backup is an unbiased one-sample estimator of a robust Bellman operator that arises naturally from a transition-regularized MDP framework. This formulation corresponds to optimizing performance against a specific class of adversarial dynamics perturbations at test time that reallocate transition probability mass away from high-value states, thereby inducing conservative but stable decision-making. Under this interpretation, Clipped Q-Learning can be viewed as tracking the fixed point of the robust Bellman equation and learning policies that hedge against adversarial dynamics shifts at test time. We analyze two Clipped Q-Learning variants with optimistic exploration bonuses and establish polynomial regret guarantees, demonstrating statistical efficiency. Beyond the tabular setting, our framework suggests that value clipping is a modular mechanism that can be incorporated into general value-based RL algorithms with function approximation. As a proof of concept, we evaluate a clipped Double DQN algorithm on a control task and observe improved robustness.

## 1. Introduction

Q-learning (Watkins & Dayan, 1992) is one of the foundational Reinforcement Learning (RL) algorithms for solving a Markov decision process (MDP). Its deep RL extensions, such as DQN (Mnih et al., 2015) and its variants (Van Hasselt et al., 2016; Hessel et al., 2018), have achieved remark-

[1] Department of Biostatistics and Bioinformatics, Duke University, Durham. Correspondence to: Zhishuai Liu <zhishuai.liu@duke.edu>, Pan Xu <pan.xu@duke.edu>.

*Proceedings of the 43rd International Conference on Machine Learning*, Seoul, South Korea. PMLR 306, 2026. Copyright 2026 by the author(s).

**Standard Q-Learning** update rule:

$$Q_{t+1}(s,a) \leftarrow (1 - \alpha_t)Q_t(s,a) + \alpha_t(\hat{\mathcal{T}}Q_t)(s,a)$$

Here $\hat{\mathcal{T}}$ represents the Bellman Backup:

$$(\hat{\mathcal{T}}Q)(s,a) = r(s,a) + \max_{a'} Q(s',a')$$

↓ Value Clipping

$$(\hat{\mathcal{T}}^\lambda Q)(s,a) = r(s,a) + \min\{\max_{a'} Q(s',a'), \lambda\}$$

↕ Regularized Robust MDP

**Clipped Q-Learning** Bellman Backup:

$$Q(s,a) = r(s,a) + \inf_{P(\cdot) \in \Delta(\mathcal{S})} \left[ \mathbb{E}_{s' \sim P(\cdot)} \left[ \max_{a'} Q(s',a') \right] + \lambda \cdot D_{\mathrm{TV}}(P(\cdot) || P^0(\cdot|s,a)) \right]$$

*Figure 1.* Clipped Q-learning can be interpreted as solving a transition-regularized robust MDP.

able empirical successes in game-playing, robotics control and beyond. Classical MDP theory (Bertsekas, 2025) guarantees that the optimal Q-function $Q^\star(s,a)$ is the unique fixed point of the Bellman optimality operator:

$$(\mathcal{T}Q)(s,a) = r_{s,a} + \gamma \mathbb{E}_{s' \sim P_{s,a}} \left[ \max_{a'} Q(s',a') \right]. \quad (1.1)$$

Q-learning first constructs an unbiased stochastic approximation of $\mathcal{T}Q$ using a single sampled transition $(s,a,r,s')$ as a stochastic target, $(\hat{\mathcal{T}}Q)(s,a) = r + \gamma \max_{a'} Q(s',a')$. Consequently, its update rule

$$Q_{t+1}(s,a) = (1 - \alpha_t)Q_t(s,a) + \alpha_t(\hat{\mathcal{T}}Q_t)(s,a) \quad (1.2)$$

can be interpreted as a stochastic approximation iteration that tracks the fixed point of $\mathcal{T}$. An agent then plans with the approximated optimal Q-function, which guides decision making to achieve the maximum expected cumulative reward when interacting with the MDP environment characterized by the transition kernel $P$. Extensive theoreti-

cal studies have analyzed the convergence of various Q-learning variants in different tabular MDP settings (Jin et al., 2018; Zhang et al., 2020; Qu & Wierman, 2020; Wainwright, 2019), establishing a grounded theoretical understanding of this classical RL algorithm.

In this paper, we introduce a simple yet interesting **value-clipping mechanism** as a modular plug-in to Q-learning, and provide theoretical and intuitive interpretations of its effect on the behavior of the learned Q-function. In particular, under episodic tabular MDPs with $\gamma = 1$, we introduce the Clipped Q-Learning, which distinctively employs a clipped version of Bellman backup

$$(\hat{\mathcal{T}}^\lambda Q)(s,a) = r + \left[ \max_{a'} Q(s',a') \right]_\lambda$$

$$(\mathcal{T}^\lambda Q)(s,a) = r + \mathbb{E}_{s' \sim P_{s,a}} \left[ \max_{a'} Q(s',a') \right]_\lambda$$

and updates its Q-table as [1]

$$Q_{t+1}^\lambda(s,a) = (1-\alpha_t)Q_t^\lambda(s,a) + \alpha_t(\hat{\mathcal{T}}^\lambda Q_t^\lambda)(s,a),$$
$$(1.3)$$

where $\lambda > 0$ is a predefined scalar serving as the clipping threshold. We use the superscript $\lambda$ to distinguish $Q^\lambda(s,a)$ from the $Q$-function induced by the classical Q-learning update (1.2). Notably, the Clipped Q-Learning constructs a new stochastic target, $r + \left[ \max_{a'} Q(s',a') \right]_\lambda$, by combining the immediate reward with a clipped estimate of the next-state value. A few works in the literature have deployed a similar clipping mechanism in Q-learning variants by instantiating $\lambda$ to be the ceiling of all possible values[2]. For example, Jin et al. (2018) employ the value clipping in a Q-learning variant that incorporates UCB-exploration to ensure the value function estimation is properly upper bounded; Chen et al. (2023) show that the value clipping together with a target network is sufficient to provably stabilize Q-learning with linear function approximation, avoiding the infamous deadly triad (Sutton & Barto, 2018). Prior works have primarily used value clipping as a technical device to ensure bounded Q-value estimates. To the best of our knowledge, clipping the Bellman backup at a general threshold $\lambda$ has *not* been previously proposed, and it appears to deviate from the Bellman optimality operator (1.1) when $\lambda$ is chosen below the natural value ceiling. In this work, we explicitly study this form of clipping and show that it admits a principled interpretation: the resulting stochastic target is an unbiased one-sample estimator of a robust Bellman operator arising naturally from a class of transition-regularized MDPs.

---

[1] For a function $V : \mathcal{X} \to \mathbb{R}$ and a scalar $\lambda$, we define $[V(x)]_\lambda := \min\{V(x), \lambda\}$.

[2] Assuming $r \in [0,1]$, $\lambda$ is defined as $1/(1-\gamma)$ in discounted MDPs, and in finite horizon episodic MDPs, $\lambda := H$ is the horizon length.

This interpretation naturally connects our method to a recent line of works on regularized robust Markov decision processes (RRMDPs, Yang et al. (2023); Zhang et al. (2024); Panaganti et al. (2024); He et al. (2025); Tang et al. (2025); Gu et al. (2025)). The RRMDP framework addresses robust decision making to hedge against adversarial dynamics perturbation during the test time. Such robustness properties have profound implications for real-world applications, including autonomous driving, where systems must cope with unexpected road conditions and driving behaviors of other units; robotic control, where stability under modeling errors and external disturbances is critical; and optimal treatment regimes, where decision policies must remain reliable under patient heterogeneity and uncertainty. Viewed through the lens of RRMDPs, the Clipped Q-Learning can be interpreted as tracking the fixed point of the robust Bellman equation (see (4.1) for details) for a class of RRMDPs where the regularization term is defined by the total variation (TV) distance between the nominal transition kernel and target transition kernel multiplied by the regularizer $\lambda$. Thus, the value clipping in (1.3) is secretly adopting a robust Bellman operation that enables robust policy learning. An illustration from Q learning to robust MDPs is presented in Figure 1.

We summarize the contributions of this work as follows:

- We analyze Clipped Q-Learning and show that it corresponds to solving a robust Bellman equation arising from a class of RRMDPs. This interpretation reveals that Clipped Q-Learning learns policies robust to a specific worst-case dynamics perturbation in which probability mass is shifted from high-value successor states to minimal-value states. Through a concrete MDP example, we further illustrate the resulting robust policies favor stable actions under such high-value attacks.

- We prove that Clipped Q-Learning is statistically efficient by analyzing two Clipped Q-Learning variants with UCB exploration and establishing polynomial regret bounds. Our analysis develops several new technical tools tailored to TV-distance–regularized RRMDPs, including a range shrinkage lemma (Proposition B.1), a performance difference lemma (Lemma B.10), and a total variance law (Lemma B.8), which may be of independent interest.

- The value-clipping module can be readily integrated into a broad class of value-based RL algorithms, and naturally extends to large-scale function approximation. We equip Double DQN (Van Hasselt et al., 2016) with the value-clipping module and evaluate it heuristically on the classical control task *Inverted Pendulum*. The experimental results show policies learned by the clipped Double DQN exhibit substantially improved stability under pole-length perturbations.

**Notations:** We denote $\Delta(\mathcal{S})$ as the set of probability measures over some set $\mathcal{S}$. For any number $H \in \mathbb{Z}_+$, we denote $[H]$ as the set of $\{1, 2, \cdots, H\}$. For any function $V : \mathcal{S} \to \mathbb{R}$, we denote $[\mathbb{P}_h V](s, a) = \mathbb{E}_{s' \sim P_h(\cdot|s,a)}[V(s')]$ as the expectation of $V$ with respect to the transition kernel $P_h$, and $[V(s)]_\alpha = \min\{V(s), \alpha\}$, given a scalar $\alpha > 0$, as the truncated value of $V$.

## 2. Related Work

**Q-learning** Q-learning (Watkins et al., 1989) is a foundational RL algorithm for learning optimal state-action value functions via off-policy temporal-difference updates. Its convergence properties in tabular MDPs have been extensively studied in various settings (Watkins & Dayan, 1992; Strehl et al., 2006; Jin et al., 2018; Wainwright, 2019; Qu & Wierman, 2020; Zhang et al., 2020; Chen et al., 2023). Among these works, Jin et al. (2018) and Chen et al. (2023) also use value clipping in their analyses. However, they mostly set the clipping threshold to the (known) value range to guarantee bounded $Q$-value estimates. For a general clipping threshold, the underlying mechanism remains unclear. Practically, DQN (Mnih et al., 2015) and its variants (Van Hasselt et al., 2016; Hessel et al., 2018; Liu et al., 2023) extend Q-learning with neural-network function approximation, and have achieved remarkable empirical successes. Our value-clipping operator can be inserted into these algorithms to promote robust policy learning.

**Distributionally Robust RL** There is a line of work studying robust decision-making under the distributionally robust MDP (RMDP) framework when there is distribution shift between a nominal (training) environment and a target (test) environment (Iyengar, 2005; Nilim & El Ghaoui, 2005; Wiesemann et al., 2013; Zhou et al., 2021; Liu et al., 2022; Shi et al., 2024; Liu & Xu, 2024b; 2025; Blanchet et al., 2024). In RMDPs, the adversary is restricted to choosing transition kernels from an uncertainty set centered around the nominal kernel, and the agent aims to learn a policy that performs well in the worst-case environment within that set. Liu et al. (2022) propose a robust Q-learning algorithm under the RMDP framework; however, their method relies on a simulator to generate an arbitrary number of samples at each state-action pair to compute accurate Bellman backups. Liang et al. (2023) study robust Q-learning using a single trajectory collected from the nominal environment, which is the most similar setting to ours. However, their formulation requires a convoluted three-time-scale update rule for $Q$-function estimation.

**Robust Regularized MDPs** Regularized robust MDPs (RRMDPs) (Yang et al., 2023; Panaganti et al., 2024) have recently been proposed as an alternative to classical RMDPs, replacing hard uncertainty-set constraints with soft regularization terms on transition dynamics. Early work by Zhang et al. (2024) showed that RRMDPs admit a closed-form dual formulation when the regularization is defined via the Kullback–Leibler divergence. Subsequently, Tang et al. (2025) demonstrated that RRMDPs with TV-distance regularization also enjoy a closed-form dual formulation. More recently, He et al. (2025) studied online learning in RRMDPs and identified the fundamental information deficit issue arising from distributional shift between nominal and target environments. Despite these advances, the structure of worst-case transition dynamics in RRMDPs without explicit uncertainty sets remains poorly understood, and a clean, efficient learning algorithm that is straightforward to scale beyond tabular settings has yet to be established.

**Regularized MDPs** There is a rich body of work on regularized Markov Decision Processes. In particular, Geist et al. (2019) develop a general theoretical framework for regularized MDPs, with a focus on policy regularization. Building on this line of work, Derman et al. (2021) show that policy-regularized MDPs are equivalent to reward-robust MDPs, and further establishes that MDPs with both transition and reward uncertainty sets can be reformulated as MDPs with policy and value regularization. In a related direction, (Xu & Mannor, 2010) studies Wasserstein distributionally robust MDPs and characterizes a dual relationship between robustness to model uncertainty and value function regularization. Our work is orthogonal to these prior studies. Rather than focusing on policy or value regularization, we investigate transition-regularized MDPs in the online setting, and propose a robust variant of Q-learning with finite-sample guarantees.

## 3. Preliminary

In this section, we lay out the formulation to derive Clipped Q-Learning. An episodic tabular regularized robust MDP (RRMDP) is denoted as a tuple $(\mathcal{S}, \mathcal{A}, P^0, r, \lambda, D, H)$, where $\mathcal{S}$ is the state space with size $S$, $\mathcal{A}$ is the action space with size $A$, $P^0 = \{P_h^0\}_{h=1}^H$, $P_h^0 : \mathcal{S} \times \mathcal{A} \to \Delta(\mathcal{S})$ are the *nominal* transition kernels, $r = \{r_h\}_{h=1}^H, r_h : \mathcal{S} \times \mathcal{A} \to [0, 1]$ are the reward functions, $\lambda \geq 0$ is the regularizer, $D(\cdot||\cdot)$ is the probability divergence measure and $H$ is the horizon length. For any policy $\pi$, the regularized value function is defined as the infimum, over all transition kernel $P$, of expected cumulative regularized reward $r_h(s, a) + \lambda D(P_h(\cdot|s, a)||P_t^0(\cdot|s, a))$ achieved by policy $\pi$,

$$V_h^{\pi, \lambda}(s) = \inf_P \mathbb{E}^{\pi, P} \left[ \sum_{t=h}^{H} r_t(s_t, a_t) \right. \tag{3.1}$$

$$\left. + \lambda \cdot D(P_t(\cdot|s_t, a_t)||P_t^0(\cdot|s_t, a_t)) \Big| s_h = s \right].$$

Similarly, the regularized Q function is defined as

$$Q_h^{\pi,\lambda}(s,a) = \inf_P \mathbb{E}^{\pi,P}\Bigg[\sum_{t=h}^H r_t(s_t, a_t)$$

$$+ \lambda \cdot D(P_t(\cdot|s_t, a_t)||P_t^0(\cdot|s_t, a_t))\Big|s_h = s, a_h = a\Bigg]. \tag{3.2}$$

In both definitions, the operator $\inf_P$ is taken over the general class of transition kernels

$$\{P = \{P_h\}_{h=1}^H | P_h : \mathcal{S} \times \mathcal{A} \to \Delta(\mathcal{S}), \forall h \in [H]\}.$$

We remark that in stark contrast to widely studied entropy regularized MDPs (Neu et al., 2017; Haarnoja et al., 2018; Eysenbach & Levine, 2021), where the regularization is defined to be the entropy of a policy $\pi$, RRMDP adopts a transition-based regularization as shown in (3.1). More importantly, the regularized value functions for RRMDPs aim to evaluate a policy $\pi$ in the *worst-case* environment. The regularization term controls how far the worst kernel can deviate from the nominal one, by adding compensation term to the cumulative rewards and then taking infimum over all transition kernels. It is clear that the vanilla value functions for standard MDPs are lower bounded by the regularized value functions. Further, we define the optimal regularized value functions, for any $(s, a, h) \in \mathcal{S} \times \mathcal{A} \times [H]$,

$$V_h^{\star,\lambda}(s) = \max_\pi V_h^{\pi,\lambda}(s), \quad Q_h^{\star,\lambda}(s,a) = \max_\pi Q_h^{\pi,\lambda}(s,a).$$

The goal of RRMDPs is to find an optimal policy that achieves the optimal regularized value function, i.e., $\pi^{\star,\lambda} = \arg\max_\pi V_1^{\pi,\lambda}(s), \forall s \in \mathcal{S}$. Solving an RRMDP can be understood as a two-player game between the agent and an adversary, corresponding to a max–min optimization over policies and transition kernels. By definition, the optimal policy $\pi^{\star,\lambda}$ is the solution to the max-min optimization problem and achieves optimal performances in the worst-case environment. Yang et al. (2023); Tang et al. (2025) have shown that the important dynamic programming principles for planning hold under RRMDPs. In particular, we have *Robust Bellman Equations*:

$$Q_h^{\pi,\lambda}(s,a) = r_h(s,a) + \inf_{P'(\cdot) \in \Delta(\mathcal{S})} \big[\mathbb{E}_{s' \sim P'(\cdot)}\big[V_{h+1}^{\pi,\lambda}(s')\big]$$

$$+ \lambda \cdot D(P'(\cdot)||P_h^0(\cdot|s,a))\big], \tag{3.3}$$

$$V_h^{\pi,\lambda}(s) = \mathbb{E}_{a \sim \pi_h(\cdot|s)}\big[Q_h^{\pi,\lambda}(s,a)\big]. \tag{3.4}$$

and *Robust Bellman Optimality Equations*:

$$Q_h^{\star,\lambda}(s,a) = r_h(s,a) + \inf_{P'(\cdot) \in \Delta(\mathcal{S})} \big[\mathbb{E}_{s' \sim P'(\cdot)}\big[V_{h+1}^{\star,\lambda}(s')\big]$$

$$+ \lambda \cdot D(P'(\cdot)||P_h^0(\cdot|s,a))\big], \tag{3.5}$$

$$V_h^{\star,\lambda}(s) = \max_{a \in \mathcal{A}}\big[Q_h^{\star,\lambda}(s,a)\big]. \tag{3.6}$$

Thus, the optimal policy $\pi^{\star,\lambda}$ is greedy with respect to the optimal regularized Q-function. It is the solution to the max-min optimization problem and achieves optimal performances in the worst-case environment. However, this robustness comes at the cost of potential suboptimality in the nominal environment. $\pi^{\star,\lambda}$ may sacrifice performance under the nominal dynamics in order to hedge against worst-case deviations, reflecting an inherent trade-off between robustness and nominal optimality.

The online learning of RRMDPs proceeds as follows. An agent interacts with the *nominal* environment for $K$ episodes to learn the optimal robust policy $\pi^{\star,\lambda}$. Specifically, at the start of episode $k$, the agent selects a policy $\pi^k$ based on the interaction history and receives the initial state $s_1^k$. The agent then executes $\pi^k$ until the end of the episode and collects the resulting trajectory. The goal is to minimize the regret after $K$ episodes, defined as

$$\text{Regret}(K) = \sum_{k=1}^K \big[V_1^{\star,\lambda}(s_1^k) - V_1^{\pi^k,\lambda}(s_1^k)\big].$$

### 3.1. What Does Clipped Q-Learning Learn?

Recall (1.3), Clipped Q-Learning employs a clipped Bellman backup of the form $r + [\max_{a'} Q(s', a')]_\lambda$. We show that this update arises from a class of TV-regularized RRMDPs.

**Definition 3.1.** Define $D(\cdot||\cdot)$ as the TV-distance with an absolute continuity constraint: for any $P, Q \in \Delta(\mathcal{S})$ such that $P \ll Q$, $D_{TV}(P||Q) = \frac{1}{2}\sum_{s \in \mathcal{S}} |P(s) - Q(s)|$; otherwise, $D_{TV}(P||Q) = +\infty$.

With Definition 3.1, TV-regularized RRMDPs prohibit support shift relative to the nominal kernel, meaning transitions to states outside the support of the nominal kernel are not allowed. Consequently, the robust Bellman optimality equation (3.5) involves a TV-regularized optimization over an infinite-dimensional space, which appears challenging to solve. Surprisingly, we show it admits a simple closed-form solution that closely resembles the classical Bellman equation.[3]

**Proposition 3.2.** *Given any probability measure $\mu^0 \in \Delta(\mathcal{S})$, scalar $\lambda > 0$ and value function $V : \mathcal{S} \to [0, H]$, define the support of $\mu^0$ as $supp(\mu^0) := \{s \in \mathcal{S}|\mu^0(s) > 0\}$ and the minimal value of $V$ over $supp(\mu^0)$ as $V_{\min}^{\mu^0} := \min_{s \in supp(\mu^0)} V(s)$. Set $D(\cdot||\cdot)$ as in Definition 3.1, the dual formulation of the regularized optimization problem is:*

$$\inf_{\mu \in \Delta(\mathcal{S})} \big[\mathbb{E}_{s \sim \mu} V(s) + \lambda D_{TV}(\mu||\mu^0)\big] = \mathbb{E}_{s \sim \mu^0}[V(s)]_{V_{\min}^{\mu^0}+\lambda}. \tag{3.7}$$

---

[3]Although this phenomenon was first observed by Tang et al. (2025), their analysis implicitly assumes that $\mu^0$ has full support on $\mathcal{S}$. Here we provide a more refined analysis.

It turns out the *value-clipping operator* emerges naturally from the strong duality (3.7) of the TV-regularized optimization involved in the robust Bellman optimality equation (3.5). Next, we further relax the absolute continuous condition in Definition 3.1 to allow support shifts, and present the corresponding strong duality result. The proof for Proposition 3.2 and Proposition 3.4 is provided in Appendix A.1.

**Definition 3.3.** Define the general TV-distance: for any $P, Q \in \Delta(\mathcal{S})$, $D_{\text{TV}}(P||Q) = \frac{1}{2} \sum_{s \in \mathcal{S}} |P(s) - Q(s)|$.

**Proposition 3.4.** *Given any probability measure $\mu^0 \in \Delta(\mathcal{S})$, scalar $\lambda > 0$ and value function $V : \mathcal{S} \to [0, H]$, define the minimal value of $V$ over domain $\mathcal{S}$ as $V_{\min} := \min_{s \in \mathcal{S}} V(s)$. Set $D(\cdot||\cdot)$ as in Definition 3.3, the dual formulation of the regularized optimization problem is:*

$$\inf_{\mu \in \Delta(\mathcal{S})} \left[ \mathbb{E}_{s \sim \mu} V(s) + \lambda D_{TV}(\mu||\mu^0) \right] = \mathbb{E}_{s \sim \mu^0} [V(s)]_{V_{\min} + \lambda}.$$
(3.8)

Proposition 3.4 implies that the robust Bellman equation remains linear in the nominal transition kernel $P^0$. This enables unbiased one-sample estimation, which is crucial for Q-learning updates. Replacing $V(s')$ by $\max_{a'} Q(s', a')$ and $\mu^0$ by $P^0$, we conclude from Proposition 3.4 that the clipped Bellman backup $r + [\max_{a'} Q(s', a')]_{\lambda'}$, where $\lambda' = V_{\min} + \lambda$, is an unbiased one-sample estimator of the robust Bellman optimality operator. *Thus, Clipped Q-Learning aims to learn the optimal regularized Q-function $Q^{\star,\lambda}$ and its corresponding greedy policy $\pi^{\star,\lambda}$.*

*Remark* 3.5. We note that there is another line of works (Iyengar, 2005; Nilim & El Ghaoui, 2005; Panaganti & Kalathil, 2022; Shi et al., 2024; Liu & Xu, 2024a) studying distributionally robust RL, where the adversary is strictly restricted to picking a worst-case transition kernel from a TV-distance defined uncertainty set $\mathcal{U}^\rho(\mu^0) := \{\mu \in \Delta(\mathcal{S}) : D_{\text{TV}}(\mu||\mu^0) \leq \rho\}$. This formulation replaces the soft regularization term in our framework by a hard uncertainty set constraint, leading to the following constrained optimization problem with strong duality: $\inf_{\mu \in \mathcal{U}^\rho(\mu^0)} \mathbb{E}_{s \sim \mu} V(s) = \max_{\alpha \in [0,H]} \{ \mathbb{E}_{s \sim \mu^0} [V(s)]_\alpha - \rho(\alpha - \min_{s' \in \mathcal{S}} [V(s')]_\alpha) \}$. Compared to (3.8), the dual formulation on the right-hand side involves an additional optimization problem over a scalar variable $\alpha$, which leads to a biased finite-sample estimation due to the outer maximization operator. This observation highlights the simplicity and unique advantage of RRMDPs.

## 3.2. What Does Clipped Q-Learning Optimize?

By definition of the regularized Q-function in (3.2), Clipped Q-Learning learns a policy that optimizes performance in the worst-case target environment. To further elucidate its robustness properties, we next characterize the worst-case transition kernel selected by the adversary. Beyond the choice of divergence $D(\cdot||\cdot)$, the regularizer $\lambda$ also

governs the structure of the worst-case transition kernel. Building on Proposition 3.4, we derive its explicit form in Proposition 3.6, and the proof is postponed to Appendix A.2.

**Proposition 3.6.** *Given a probability measure $\mu^0 \in \Delta(\mathcal{S})$, scalar $\lambda > 0$ and value function $V : \mathcal{S} \to [0, H]$, define*

$$\mathcal{S}_\lambda := \{s \in \mathcal{S} \mid V(s) - V_{\min} > \lambda\}$$

*as the subset of states whose values exceed the threshold $\lambda + V_{\min}$. Let $s_{\min} \in \arg\min_{s \in \mathcal{S}} V(s)$ denote a minimizer of the value function. Define the probability measure*

$$\mu^\dagger(s) = \begin{cases} 0 & s \in \mathcal{S}_\lambda \\ \mu^0(s) & s \in \mathcal{S}_\lambda^c / \{s_{\min}\} \\ \mu^0(s_{\min}) + \mu^0(\mathcal{S}_\lambda) & s = s_{\min} \end{cases}.$$

*Then, $\mu^\dagger(\cdot)$ is a minimizer (worst-case probability measure) of the regularized optimization problem,*

$$\mu^\dagger(\cdot) \in \arg\inf_{\mu \in \Delta(\mathcal{S})} \left[ \mathbb{E}_{s \sim \mu} V(s) + \lambda D_{TV}(\mu||\mu^0) \right].$$

Proposition 3.6 shows that the worst-case transition kernel is fully determined by the value function $V$ and the regularization parameter $\lambda$, independent of the explicit uncertainty set constraints typically imposed in distributionally robust RL (cf. Remark 3.5). In particular, *the adversary reallocates probability mass from high-value states—whose values exceed a $\lambda$-dependent threshold—to states attaining the minimal value.* Consequently, RRMDPs with TV-distance regularization hedge against a specific class of adversarial attacks that target high-value states.

Next, we use a toy example to illustrate optimizing the regularized value function (3.1) under the worst-case transition kernel yields a targeted robust policy that differs from the optimal policy for the nominal environment. We further relate this toy example to a motivating healthcare scenario, thereby providing intuition for the induced robust behavior.

**Example 3.7.** Consider an episodic MDP with horizon $H = 3$, state space $\{s_1, s_2, s_{\text{good}}, s_{\text{moderate}}, s_{\text{bad}}\}$, and action space $\{a_1, a_2\}$. The initial state is always $s_1$. At stage $h = 1$, no matter what actions taken, it will always transition to $s_2$. From state $s_2$ it will transition to $s_{\text{good}}, s_{\text{moderate}}$ or $s_{\text{bad}}$ with the nominal transition kernel as:

$$P_2^0(s_{\text{good}}|s_2, a) = \mathbb{1}\{a = a_1\} \cdot \frac{1}{2},$$

$$P_2^0(s_{\text{moderate}}|s_2, a) = \mathbb{1}\{a = a_2\},$$

$$P_2^0(s_{\text{bad}}|s_2, a) = \mathbb{1}\{a = a_1\} \cdot \frac{1}{2}.$$

Rewards are only obtained at the terminal states, with $r(s_{\text{good}}, a) = 1$ and $r(s_{\text{moderate}}, a) = 0.3$ for any $a \in \mathcal{A}$, and $r(s_{\text{bad}}, a) = 0$. It is straightforward to verify that the

nominal optimal policy takes action $a_1$ at state $s_2$, yielding $V_1^\star(s_1) = 0.5$.

Now set the regularizer $\lambda = 0.5$, and consider the TV-regularized MDP, $\mathcal{M}^r = (\mathcal{S}, \mathcal{A}, P^0, r, \lambda, D_{\mathrm{TV}}, H)$. At the last stage $h = 3$, the optimal robust value is the immediate reward, hence $V_3^{\star,\lambda}(s_{\mathrm{good}}) = 1$, $V_3^{\star,\lambda}(s_{\mathrm{moderate}}) = 0.3$, and $V_3^{\star,\lambda}(s_{\mathrm{bad}}) = 0$. At stage $h = 2$, Proposition 3.6 implies that under action $a_1$ the adversary shifts mass away from the high-value outcome $s_{\mathrm{good}}$ toward the minimum-value outcome $s_{\mathrm{bad}}$, resulting in

$$P_2^\dagger(s_{\mathrm{good}}|s_2, a_1) = 0, \quad P_2^\dagger(s_{\mathrm{bad}}|s_2, a_1) = 1.$$

Therefore, according to the robust Bellman equation (3.3), we can calculate the optimal robust Q-function as $Q_2^{\star,\lambda}(s_2, a_1) = 0.25$ and $Q_2^{\star,\lambda}(s_2, a_2) = 0.3$. And hence $\pi_2^{\star,\lambda}(s_2) = \operatorname{argmax}_{a \in \{a_1, a_2\}} Q_2^{\star,\lambda}(s_2, a) = a_2$.

In summary, the nominal optimal policy $\pi^\star$ takes action $a_1$ at the second stage leading to an equal probability transitioning to $s_{\mathrm{good}}$ and $s_{\mathrm{bad}}$, which achieves a high mean reward. The optimal robust policy $\pi^{\star,\lambda}$ takes action $a_2$ at the second stage that leads with probability one to state $s_{\mathrm{moderate}}$ with a moderate reward, which achieves lower variance and better worst-case performance.

**Healthcare interpretation.** This example admits a natural interpretation in healthcare decision-making. A patient arrives at the hospital in an initial condition $s_1$, undergoes diagnosis at state $s_2$, and then receives one of two possible treatments at the second stage. Action $a_1$ represents an *aggressive treatment* (e.g., high-dose chemotherapy). Under the nominal model, this treatment either cures the patient ($s_{\mathrm{good}}$ with reward 1) or causes fatal complications ($s_{\mathrm{bad}}$ with reward 0), each with probability $1/2$. Since the mean outcome is favorable, the nominal optimal policy $\pi^\star$ will select this treatment.

Action $a_2$ represents a *conservative treatment* (e.g., symptom-managing therapy). It does not fully cure the patient, but reliably leads to a stable and manageable outcome ($s_{\mathrm{moderate}}$ with reward 0.3) with probability one. In the TV-regularized MDP, we explicitly account for worst-case perturbations of the transition dynamics, capturing severe medical uncertainty, unanticipated side effects, or rare but fatal complications. Under such adversarial shifts, the favorable outcome of the aggressive treatment can be eliminated, making action $a_1$ catastrophic in the worst case. By contrast, action $a_2$ remains stable under all admissible perturbations. Therefore, the optimal robust policy $\pi^{\star,\lambda}$ selects the conservative treatment $a_2$. Although this choice sacrifices the possibility of a full cure, it avoids catastrophic failure in the worst case.

This example highlights a central principle of robustness: when facing uncertainty or adversarial conditions, it can be

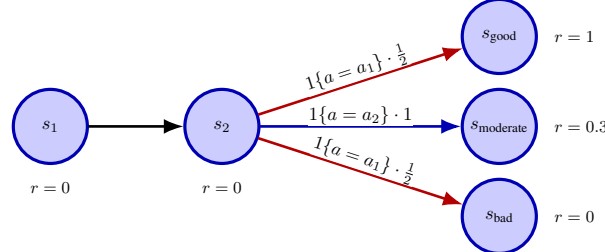

*Figure 2.* Illustration of the optimal robust policy.

optimal to favor reliable, low-variance actions over high-risk, high-reward alternatives, much like choosing a safer treatment plan in healthcare when worst-case outcomes must be guarded against.

## 4. Algorithm Design and Main Results

We present two variants of Clipped Q-Learning that incorporate UCB-style exploration. Our analysis begins with an assumption that is widely adopted in the literature (Panaganti et al., 2022; Liu & Xu, 2024a; Gu et al., 2025), which will be justified later in Section 4.1.

**Assumption 4.1** (Fail-state). There is a state $s_f \in \mathcal{S}$ such that $r_h(s_f, a) = 0$ and $P_h^0(s_f|s_f, a) = 1, \forall (h, a) \in [H] \times \mathcal{A}$. With the TV distance in Definition 3.1, we additionally assume $s_f \in \operatorname{supp}(P^0(\cdot|s, a)), \forall (s, a) \in \mathcal{S} \times \mathcal{A}$.

With Assumption 4.1, we have $V_{\min}^{\pi,\lambda} = 0$ for any $\pi$. Thus, according to Proposition 3.2 and Proposition 3.4, we have the following simplified Bellman optimality equations

$$\begin{cases} V_h^{\star,\lambda}(s) = \max_{a \in \mathcal{A}} Q_h^{\star,\lambda}(s, a) \\ Q_h^{\star,\lambda}(s, a) = \left(r_h + \mathbb{P}_h^0\left[V_{h+1}^{\star,\lambda}\right]_\lambda\right)(s, a) \\ V_{H+1}^{\star,\lambda}(s) = 0 \end{cases} \quad (4.1)$$

According to (4.1), given a transition tuple $(s, a, h, s')$ and a value function estimation $\hat{V}$, an unbiased one-sample estimation of the Bellman backup is $r_h(s, a) + [\hat{V}(s')]_\lambda$. We additionally incorporate an upper confidence bound (UCB) bonus term into the backup to facilitate exploration. We present our first algorithm, Clipped Q-Learning with UCB-Hoeffding, in Algorithm 1. In particular, Algorithm 1 modifies the standard Q-learning with UCB-Hoeffding algorithm of Jin et al. (2018) by incorporating the value-clipping module and a refined design of the UCB bonus, while leaving the remaining components unchanged. In Theorem 4.2, we show that the Clipped Q-Learning is statistically efficient, the proof of which is in Appendix B.1.

**Theorem 4.2.** *There exists an absolute constant $c$ such that, for any $p \in (0, 1)$, if we choose $b_t = c \min\{1 + \lambda, H\}\sqrt{H\iota/t}$, then with probability $1 - p$, the total regret of Algorithm 1 is at most $O\left(\min\{1 + \lambda, H\}\sqrt{SAH^3 K\iota}\right)$, where $\iota = \log(SAHK/p)$.*

**Algorithm 1** Clipped Q-Learning with UCB-Hoeffding

**Require:** regularizer $\lambda$, constant $c$
1: **Initialize** $Q_h^{1,\lambda}(s,a) \leftarrow \min\{1 + \lambda, H\}$ and $N_h(s,a) \leftarrow 0$ for all $(s,a,h) \in \mathcal{S} \times \mathcal{A} \times [H]$.
2: **for** episode $k = 1, \cdots, K$ **do**
3:    Receive $s_1^k$.
4:    **for** step $h = 1, \cdots, H$ **do**
5:      Take action $a_h^k \leftarrow \operatorname{argmax}_{a' \in \mathcal{A}} Q_h^k(s_h^k, a')$ and observe $s_{h+1}^k$.
6:      $N_h^k(s_h^k, a_h^k) = N_h(s_h^k, a_h^k) \leftarrow N_h(s_h^k, a_h^k) + 1$; $t \leftarrow N_h^k(s_h^k, a_h^k); b_t \leftarrow c \min\{1 + \lambda, H\}\sqrt{H\iota/t}$
7:      $Q_h^{k+1,\lambda}(s_h^k, a_h^k) \leftarrow (1 - \alpha_t)Q_h^{k,\lambda}(s_h^k, a_h^k) + \alpha_t[r_h(s_h^k, a_h^k) + [V_{h+1}^{k,\lambda}(s_{h+1}^k)]_\lambda + b_t]$
8:      $V_h^{k+1,\lambda}(s_h^k) \leftarrow [\max_{a' \in \mathcal{A}} Q_h^{k+1,\lambda}(s_h^k, a')]_{\min\{1+\lambda, H\}}$
9:    **end for**
10: **end for**

Compared to the regret bound for the standard Q-learning with UCB-Hoeffding (Jin et al., 2018, Theorem 1), Theorem 4.2 achieves an improvement of order $O(H/\min\{1 + \lambda, H\})$. This improvement stems from a range-shrinkage phenomenon: the regularized value function is upper bounded by $\lambda + 1$, rather than $H$, a fact that can be readily verified from the Bellman equations in (4.1). A general and formal characterization of this phenomenon is provided in Proposition B.1. Theorem 4.2 suggests that a smaller $\lambda$ leads to lower sample complexity. In the extreme case of $\lambda = 0$, no learning is required, since the worst-case transition kernel deterministically moves to the state with the smallest reward regardless of the action taken. However, such overly conservative behavior generally results in poor performance. Therefore, smaller $\lambda$ does not necessarily yield satisfying robust performances. In practice, one should prioritize selecting an appropriate $\lambda$ that balances robustness and performance, rather than sacrificing performance solely for faster convergence.

Moreover, a sharper regret bound can be obtained by incorporating a Bernstein-type UCB bonus. Due to space limits, we defer the description of Algorithm 2, Clipped Q-Learning with UCB-Bernstein, to Appendix B.2, while presenting its theoretical guarantee below.

**Theorem 4.3.** *For any $p \in (0,1)$, there exists absolute constants $c_1, c_2 > 0$ such that, for $b_t$ specified in Algorithm 2, with probability at least $1 - p$, the total regret of Algorithm 2 is at most $O(\sqrt{\min\{1 + \lambda, H\}H^3SAK\iota} + \sqrt{\min\{1 + \lambda, H\}^3 H^6 S^3 A^3 \iota^2})$.*

Theorem 4.3 further improves the dependence on $\min\{1 + \lambda, H\}$ by $O(\sqrt{\min\{1 + \lambda, H\}})$. Now we can conclude that Clipped Q-Learning is both computationally and statis-

tically efficient, validating that the value clipping module is theoretically grounded to be inserted into Q-learning.

### 4.1. Justification of the Fail State Assumption 4.1.

Next, we provide a justification of Assumption 4.1 through information theoretic lower bounds. Recall the online learning protocol of RRMDPs, in which an agent interacts with the nominal environment to collect data and, *based solely on this data*, attempts to infer the optimal behavior in the worst-case target environment. However, due to the distributional shift between the nominal and target transition kernels, the information obtained from the nominal environment may be fundamentally insufficient to support reliable decision making in the target environment, potentially leading to unavoidable failure. This phenomenon is known as the *information deficit issue* (Lu et al., 2024; He et al., 2025). We formally characterize this in the following two theorems.

**Theorem 4.4** (Hardness result due to the information deficit issue). *There exist two RRMDPs $\{\mathcal{M}_0, \mathcal{M}_1\}$ with TV-distance regularization defined in Definition 3.1, such that the minimax regret lower bound can be exponentially large in the size of the action space $|\mathcal{A}|$:*

$$\inf_{\mathcal{ALG}} \sup_{\theta \in \{0,1\}} \mathbb{E}[Regret(K; \mathcal{M}_\theta, \mathcal{ALG})] \geq 2^{|\mathcal{A}|}\sqrt{K}.$$

**Theorem 4.5** (Hardness result due to support shift). *There exist two RRMDPs $\{\mathcal{M}_0, \mathcal{M}_1\}$ with TV-distance regularization defined in Definition 3.3, such that the minimax regret lower bound is linear in $K$:*

$$\inf_{\mathcal{ALG}} \sup_{\theta \in \{0,1\}} \mathbb{E}[Regret(K; \mathcal{M}_\theta, \mathcal{ALG})] \geq \Omega(K).$$

The proofs of Theorem 4.4 and Theorem 4.5 are provided in Appendix B.3 and Appendix B.4. Both hardness instances are constructed around the presence of a state that is exponentially hard (Theorem 4.4)—or even impossible (Theorem 4.5)—to reach through interaction with the nominal environment, leading to severely limited information. However, distributional shift in the target environment substantially increases the visitation probability of this state. As a result, the agent is required to make decisions at this state despite having little or no prior experience, which leads to a large performance gap. In general, we call this phenomenon as the information deficit issue in RRMDPs. It is now clear that, without additional assumptions, statistically efficient learning in RRMDPs is in general impossible. Assumption 4.1 has been widely adopted in prior work on distributionally robust RL (Panaganti et al., 2022; Liu & Xu, 2024a; Liu et al., 2024; Gu et al., 2025). The intuition behind the fail-state assumption follows from Proposition 3.6: in the worst case, probability mass is transferred only from other states to the fail state. Moreover, the agent does not

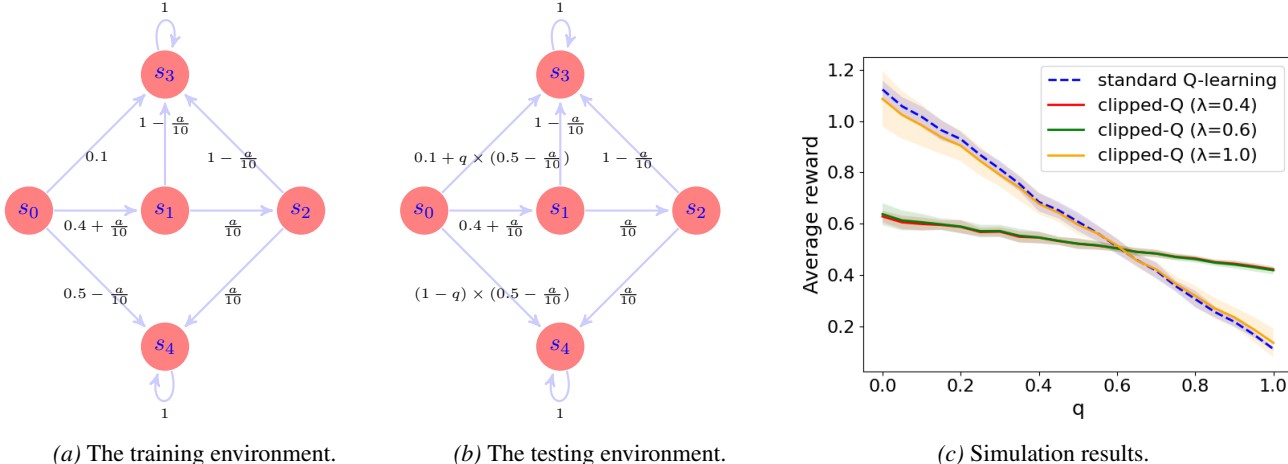

*(a)* The training environment.          *(b)* The testing environment.          *(c)* Simulation results.

*Figure 3.* Figure (a) and (b) are illustrations of the training and testing environment, respectively. Figure (c) is the simulation results.

require prior information to make decisions at the fail state, since all actions yield zero value. This effectively bypasses the information deficit issue discussed above and thereby identifies a tractable subclass of RRMDPs.

## 5. Proof-of-Concept Experiments

In this section, we first validate Algorithm 1 in a tabular MDP. We then test the value-clipping operator in a function-approximation setting by augmenting the Double DQN (DDQN; Van Hasselt et al. 2016) with our clipping module on the classical control task *Inverted Pendulum.*

### 5.1. A Simulated MDP

**Environment setup.** The nominal environment in Figure 3a is an MDP with five states $\{s_0, \cdots, s_4\}$, five actions $\{0, \cdots, 4\}$, and transition probabilities shown on the edges. At states $s_0, s_1, s_2$, taking action $a$ yields reward $a/20$. At states $s_4$ and $s_3$, the rewards are always 1 and 0, respectively. The target environment in Figure 3b is the same as the nominal MDP except that the transition probabilities from $s_0$ are perturbed; the perturbation level is controlled by $q$.

**Training and testing.** For each fixed $\lambda$, we train Algorithm 1 for 1000 episodes and evaluate the learned policy in target environments with $q \in [0, 1]$. For each target environment, we roll out 500 trajectories and report the average cumulative return. As a baseline, we set $\lambda = \infty$ (i.e., no clipping), in which case Algorithm 1 reduces to Q-learning with UCB-Hoeffding (Jin et al., 2018, Algorithm 1). All results are averaged over 20 independent repetitions.

**Results.** Figure 3c reports the evaluation curves. Recall that RRMDPs optimize worst-case performance, and we

can observe that moderate clipping (e.g., $\lambda \in \{0.4, 0.6\}$) improves performance under larger perturbations (larger $q$) compared to the unclipped baseline. This is consistent with our theoretical interpretation.

### 5.2. The Inverted Pendulum

Beyond Algorithm 1, the value clipping module can in principle be incorporated into any value-based RL algorithm. We test the value clipping with DDQN on the Inverted Pendulum task. We choose DDQN because it mitigates Q-function overestimation, providing a stable baseline method.

**Environment Setup.** As shown in Figure 4, the nominal (training) environment is the Inverted Pendulum simulator with pole length set to be 0.6 by default. We construct target environments by changing the pole length. The maximum steps in an episode is set to be 500. To apply DDQN, we discretize the action space into 11 discrete actions.

**Training and testing.** We set the discounted factor $\gamma = 0.99$, and thus the optimal Q-function is approximately 100. To test the robustness of DDQN across different clipping levels, we set $\lambda$ to be $250, 200, 100, 90$. We train each method for 3000 episodes. For evaluation, we calculate the average *cumulative reward* (instead of discounted reward) over 50 episodes. Experiments are repeated over 10 random seeds.

**Results.** Experiment results are shown in Figure 5. *Clipped DDQN consistently outperforms vanilla DDQN* in target environments with substantially longer or shorter pole length than the nominal one. Surprisingly, the performance enhancement in target environments does not sacrifice the nominal environment performance. One possible explanation for this is that multiple near-optimal policies exist in this nominal environment, but they differ in sensitivity

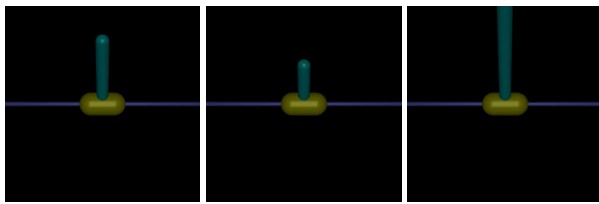

*(a)* Pole length 0.6  *(b)* Pole length 0.4  *(c)* Pole length 1.1

*Figure 4.* Illustration of the Inverted Pendulum task. Figure (a) is the nominal (training) environment with pole length 0.6; Figure (b) and (c) are testing environment with pole length 0.4 and 1.1.

with respect to the dynamics perturbation. We conducted abundant ablation study to further evaluate the clipping operator in Appendix D. The key takeaways are: (1) the value clipping operator highly relies on the base algorithm and robustness behavior only emerges under stable policy learning. If we replace DDQN with DQN, which is subject to the overestimation issue, then value clipping does not guarantee obvious robustness; and (2) value clipping often accelerates convergence, aligning well with our theoretical findings.

To the best of our knowledge, we are the first to empirically observe that a simple value-clipping scheme can improve robustness under adversarial dynamics perturbations.

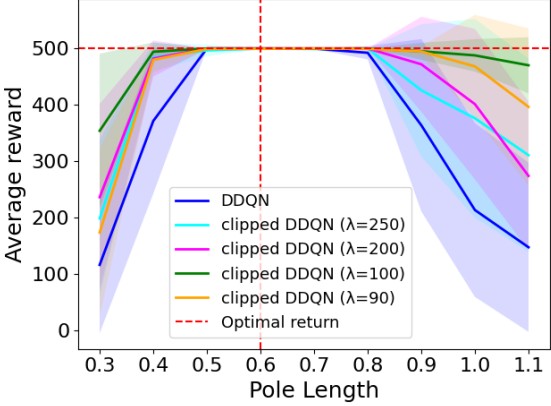

*Figure 5.* The Optimal return represents the largest cumulative reward that can be achieved at the test time. The vertical red line represents the pole length in the nominal environment. Shaded area represents the standard deviation.

## 6. Conclusions

We introduced a simple yet interesting modification to classical Q-learning that clips the value estimate in the Bellman backup updates to a threshold $\lambda$. We showed that this operation corresponds to an unbiased one-sample estimator of a robust Bellman operator induced by a recently proposed transition-regularized MDP framework. The resulting Clipped Q-Learning algorithm can be interpreted as tracking the fixed point of this robust Bellman equation, thereby

learning robust policies.

The value-clipping mechanism is modular and can, in principle, be incorporated into any value-based RL algorithms, such as Soft Actor-Critic (Haarnoja et al., 2018) and all DQN variants. As a proof of concept, we evaluated a clipped Double DQN variant on the Inverted Pendulum task and observed behaviors supporting our theoretical findings. An important direction for future work is to investigate the applicability and limitations of value clipping across broader classes of RL algorithms and tasks.

While our work provides a principled connection between value clipping and TV-regularized robust MDPs, several limitations remain. First, our finite-sample analysis relies on a fail-state assumption to control the range of the value function and enable tractable regret bounds. While this assumption is standard in theoretical analyses, it may not hold or be easily verifiable in more general environments. Second, the robustness induced by the TV-RRMDP framework is tailored to a specific class of adversarial perturbations. In particular, it primarily hedges against "large-value" attacks, where probability mass is shifted toward low-value states. As a result, it does not capture more general forms of model misspecification or structured perturbations that may arise in practice. Third, although we provide preliminary empirical evidence in a simple control setting, it remains unclear when and how the clipping module behaves in more complex environments. In particular, the interaction between clipping and function approximation in deep RL is not yet well understood, and further empirical and theoretical investigation is needed to assess its effectiveness in high-dimensional settings. Fourth, our analysis is restricted to tabular settings; extending the regret guarantees to function approximation remains open.

## Impact Statement

This paper presents work whose goal is to advance the field of Machine Learning. There are many potential societal consequences of our work, none which we feel must be specifically highlighted here.

## Acknowledgments

ZL and PX are supported in part by the National Science Foundation (DMS-2323112) and the Whitehead Scholars Program at the Duke University School of Medicine. We would like to thank Ronald Parr for his valuable feedback and discussion during the development of this work. We also thank the anonymous reviewers for their constructive feedback on this document. The views and conclusions contained in this paper are those of the authors and should not be interpreted as representing any funding agencies.

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

# A. Proof of Results in Section 3

## A.1. Proof of Proposition 3.2 and Proposition 3.4

Without loss of generality, we denote the support of $\mu^0$ as $\mathcal{X}$, i.e., $\mu^0(s) > 0, \forall s \in \mathcal{X}$.

*Proof of Proposition 3.2.* We note that the TV-distance is an instance of the general $\varphi$-divergence with $\varphi(t) = |t - 1|/2$, and it is easy to verify that its conjugate function has the following form

$$\phi^\star(s) = \begin{cases} -\frac{1}{2} & s \leq -\frac{1}{2}, \\ s & s \in [-\frac{1}{2}, \frac{1}{2}], \\ +\infty & s > \frac{1}{2}. \end{cases} \tag{A.1}$$

Thus, by Definition 3.1 and Lemma C.1, we have

$$\inf_{\mu \in \Delta(\mathcal{S})} \mathbb{E}_{s \sim \mu}[V(s)] + \lambda D_{\text{TV}}(\mu || \mu^0) = \inf_{\mu \in \Delta(\mathcal{X})} \mathbb{E}_{s \sim \mu}[V(s)] + \lambda D_{\text{TV}}(\mu || \mu^0) \tag{A.2}$$

$$= - \sup_{\mu \in \Delta(\mathcal{X})} \mathbb{E}_{s \sim \mu^0}[-V(s)] - \lambda D_{\text{TV}}(\mu || \mu^0)$$

$$= - \inf_{\eta' \in \mathbb{R}} \lambda \mathbb{E}_{s \sim \mu^0}\left[\varphi^\star\left(\frac{-\eta' - V(s)}{\lambda}\right)\right] + \eta'$$

$$= - \inf_{\eta \in \mathbb{R}} \lambda \mathbb{E}_{s \sim \mu^0}\left[\varphi^\star\left(\frac{\eta - V(s)}{\lambda}\right)\right] - \eta. \tag{A.3}$$

We substitute the conjugate function (A.1) into the RHS of the above equation and get

$$\inf_{\eta \in \mathbb{R}} \lambda \mathbb{E}_{s \sim \mu^0}\left[\varphi^\star\left(\frac{\eta - V(s)}{\lambda}\right)\right] - \eta$$

$$= \inf_{\frac{\eta - \min_{s \in \mathcal{X}} V(s)}{\lambda} \leq \frac{1}{2}} \lambda \mathbb{E}_{s \sim \mu^0}\left[\max\left\{\frac{\eta - V(s)}{\lambda}, -\frac{1}{2}\right\}\right] - \eta$$

$$= \inf_{\frac{\eta - \min_{s \in \mathcal{X}} V(s)}{\lambda} \leq \frac{1}{2}} \mathbb{E}_{s \sim \mu^0}\left[\max\left\{\eta - V(s), -\frac{\lambda}{2}\right\}\right] - \eta$$

$$= \inf_{\frac{\eta - \min_{s \in \mathcal{X}} V(s)}{\lambda} \leq \frac{1}{2}} \mathbb{E}_{s \sim \mu^0}\left[\max\left\{\eta - V(s) + \frac{\lambda}{2}, 0\right\}\right] - \eta - \frac{\lambda}{2}$$

$$= \inf_{\eta - \min_{s \in \mathcal{X}} V(s) \leq \lambda/2} \mathbb{E}_{s \sim \mu^0}\left\{\eta - V(s) + \frac{\lambda}{2}\right\}_+ - \eta - \frac{\lambda}{2}$$

$$= \inf_{\eta' - \min_{s \in \mathcal{X}} V(s) \leq \lambda} \mathbb{E}_{s \sim \mu^0}\left\{\eta' - V(s)\right\}_+ - \eta'$$

Since $V(s) \geq 0$, then when $\eta' \leq 0$, we have $\mathbb{E}_{s \sim \mu^0}\left[\{\eta' - V(s)\}_+\right] - \eta' = -\eta' \geq 0$. Thus, we can restrict the range of the inf operator as follows

$$\inf_{\eta \in \mathbb{R}} \lambda \mathbb{E}_{s \sim \mu^0}\left[\varphi^\star\left(\frac{\eta - V(s)}{\lambda}\right)\right] - \eta$$

$$= \inf_{0 \leq \eta' \leq \min_{s \in \mathcal{X}} V(s) + \lambda} \mathbb{E}_{s \sim \mu^0} \max\left\{\eta' - V(s)\right\}_+ - \eta'$$

$$= \inf_{0 \leq \eta' \leq \min_{s \in \mathcal{X}} V(s) + \lambda} -\mathbb{E}_{s \sim \mu^0}\left[V(s)\right]_{\eta'}$$

$$= - \sup_{0 \leq \eta' \leq \min_{s \in \mathcal{X}} V(s) + \lambda} \mathbb{E}_{s \sim \mu^0}\left[V(s)\right]_{\eta'}$$

$$= -\mathbb{E}_{s \sim \mu^0}\left[V(s)\right]_{\min_{s \in \mathcal{X}} V(s) + \lambda} \tag{A.4}$$

Finally, we substitute (A.4) back into (A.3) and get

$$\inf_{\mu \in \Delta(\mathcal{X})} \mathbb{E}_{s \sim \mu}[V(s)] + \lambda D_{\text{TV}}(\mu || \mu^0) = \mathbb{E}_{s \sim \mu^0}\left[V(s)\right]_{\min_{s \in \mathcal{X}} V(s) + \lambda}.$$

This completes the proof. □

Now we are ready to prove Proposition 3.4. The proof is inspired by the proof of Proposition 2.5 in (Lu et al., 2024).

*Proof of Proposition 3.4.* Note that when $\mu^0$ satisfies $\mu^0(s) > 0$ for any $s \in \mathcal{S}$, then any distribution $\mu \in \Delta(\mathcal{S})$ is absolute continuous w.r.t $\mu^0$. By Proposition 3.2 we know

$$\inf_{\mu \in \Delta(\mathcal{S})} \mathbb{E}_{s \sim \mu}[V(s)] + \lambda D_{\text{TV}}(\mu \| \mu^0) = \mathbb{E}_{s \sim \mu^0}[V(s)]_{V_{\min} + \lambda}.$$

Now for any $\mu^0$, we prove the same result by averaging $\mu^0$ with a uniform distribution and taking the limit. Specifically, denote $U \in \Delta(\mathcal{S})$ as the uniform distribution on $\mathcal{S}$. Consider the following regularized optimization problem, for any $\epsilon \in (0, 1]$,

$$P(\epsilon) := \inf_{\mu \in \Delta(\mathcal{S})} \mathbb{E}_{s \sim \mu} V(s) + \lambda D_{\text{TV}}(\mu \| (1 - \epsilon)\mu^0 + \epsilon U).$$

Consider the distribution $(1 - \epsilon)\mu^0 + \epsilon U$ with $\epsilon > 0$, it has full support on $\mathcal{S}$, i.e. $(1 - \epsilon)\mu^0(s) + \epsilon U(s) > 0$. Define

$$D(\epsilon) := (1 - \epsilon)\mathbb{E}_{s \sim \mu^0}[V(s)]_{V_{\min} + \lambda} + \epsilon \mathbb{E}_{s \sim U}[V(s)]_{V_{\min} + \lambda}.$$

Then we have $P(\epsilon) = D(\epsilon)$, for any $\epsilon \in (0, 1]$. Next we want to show $P(0) = D(0)$. To this end, we only need to prove that

$$P(0) = \lim_{\epsilon \to 0+} P(\epsilon) = \lim_{\epsilon \to 0+} D(\epsilon) = D(0).$$

**1.** $\lim_{\epsilon \to 0+} D(\epsilon)$ **exists and** $\lim_{\epsilon \to 0+} D(\epsilon) = D(0)$. $D(\epsilon)$ is continuous w.r.t to $\epsilon$ and thus the right limit exists and equals to $D(0)$. According to $P(\epsilon) = D(\epsilon)$, for any $\epsilon \in (0, 1]$, this also implies that $\lim_{\epsilon \to 0+} P(\epsilon)$ exists.

**2.** $\lim_{\epsilon \to 0+} P(\epsilon) = P(0)$. To prove this, we show that $\lim_{\epsilon \to 0+} P(\epsilon) \leq P(0)$ and $\lim_{\epsilon \to 0+} P(\epsilon) \geq P(0)$. Denote

$$\mu_0^\dagger = \arg\inf_{\mu \in \Delta(\mathcal{S})} \mathbb{E}_{s \sim \mu}[V(s)]_{V_{\min} + \lambda} + \lambda D_{\text{TV}}(\mu \| \mu^0)$$

By the definition of $P(\epsilon)$, we have

$$P(\epsilon) \leq \mathbb{E}_{s \sim \mu^\dagger}[V(s)]_{V_{\min} + \lambda} + \lambda D_{\text{TV}}(\mu_0^\dagger \| (1 - \epsilon)\mu^0 + \epsilon U)$$

On the one hand, note that the TV-distance is continuous w.r.t its second entry in terms of $\| \cdot \|_2$, taking limit on both side, we have

$$\lim_{\epsilon \to 0+} P(\epsilon) \leq \mathbb{E}_{s \sim \mu^\dagger}[V(s)]_{V_{\min} + \lambda} + \lambda D_{\text{TV}}(\mu \| \mu^0) = P(0).$$

On the other hand, consider a sequence $\{\epsilon_i\}_{i=1}^\infty$ converging to 0. Denote

$$\mu_\epsilon^\dagger = \arg\inf_{\mu \in \Delta(\mathcal{S})} \mathbb{E}_{s \sim \mu}[V(s)]_{V_{\min} + \lambda} + \lambda D_{\text{TV}}(\mu \| (1 - \epsilon)\mu^0 + \epsilon U).$$

Then $\{\epsilon_i\}_{i=1}^\infty$ defines a sequence of distributions $\{\mu_{\epsilon_i}^\dagger\}_{i=1}^\infty$, and this sequence of distributions is contained in the compact probability simplex subspace of $\mathbb{R}^{|\mathcal{S}|}$. It has a converging (w.r.t $\| \cdot \|_2$) subsequence denoted by $\{Q_{\epsilon_{i_k}}^\dagger\}_{k=1}^\infty$ whose limit is denoted as $\mu^\dagger$. Then, we have

$$\begin{aligned}
\lim_{\epsilon \to 0+} P(\epsilon) &= \lim_{\epsilon \to 0+} \mathbb{E}_{s \sim \mu_\epsilon^\dagger}[V(s)] + \lambda D_{\text{TV}}(\mu_\epsilon^\dagger \| (1 - \epsilon)\mu^0 + \epsilon U) \\
&= \lim_{k \to \infty} \mathbb{E}_{s \sim \mu_{\epsilon_{i_k}}^\dagger}[V(s)] + \lambda D_{\text{TV}}(\mu_{\epsilon_{i_k}}^\dagger \| (1 - \epsilon_{i_k})\mu^0 + \epsilon U) \\
&= \mathbb{E}_{s \sim \mu^\dagger}[V(s)] + \lambda D_{\text{TV}}(\mu^\dagger \| \mu^0) \\
&\geq \inf_{\mu \in \Delta(\mathcal{S})} \mathbb{E}_{s \sim \mu}[V(s)] + \lambda D_{\text{TV}}(\mu \| \mu^0) \\
&= P(0).
\end{aligned}$$

Thus, we conclude that $\lim_{\epsilon \to 0+} P(\epsilon) = P(0)$.

Now we have $P(0) = D(0)$, i.e.,

$$\inf_{\mu \in \Delta(\mathcal{S})} \mathbb{E}_{s \sim \mu}[V(s)] + \lambda D_{\text{TV}}(\mu \| \mu^0) = \mathbb{E}_{s \sim \mu^0}[V(s)]_{V_{\min} + \lambda},$$

where $\mu^0$ can be any distribution in $\Delta(\mathcal{S})$. This finishes the proof. $\square$

### A.2. Proof of Proposition 3.6

*Proof.* Recall the objective of interest $\mathbb{E}_{s \sim \mu} V(s) + \lambda D_{\mathrm{TV}}(\mu \| \mu^0)$, which is continuous in $\mu$ (with respect to $\| \cdot \|_2$) and the probability simplex subspace in $\mathbb{R}^{|S|}$ is compact. By the Weierstrass Theorem, the minimizer of the regularized optimization problem exists, and the infimum is attained at the minimizer. According to Proposition 3.4, the infimum of the regularized optimization problem equals to $\mathbb{E}_{s \sim \mu^0} \big[ V(s) \big]_{V_{\min} + \lambda}$. To prove the proposition, we only need to show that it is achieved at $\mu^\dagger$, i.e.,

$$\mathbb{E}_{s \sim \mu^\dagger} V(s) + \lambda D_{\mathrm{TV}}(\mu^\dagger \| \mu^0) = \mathbb{E}_{s \sim \mu^0} \big[ V(s) \big]_{V_{\min} + \lambda}.$$

Recall the definition of $\mu^\dagger$,

$$\mu^\dagger(\cdot) = \mathbb{1}\{\cdot \in \mathcal{S}_\lambda\} \cdot 0 + \mathbb{1}\{\cdot \in \mathcal{S}_\lambda^c / \{s_{\min}\}\} \cdot \mu^0(\cdot) + \mathbb{1}\{s = s_{\min}\} \cdot (\mu^0(s_{\min}) + \mu^0(\mathcal{S}_\lambda)),$$

we have $\lambda D_{\mathrm{TV}}(\mu^\dagger \| \mu^0) = \lambda \mu^0(\mathcal{S}_\lambda)$, and thus

$$
\begin{aligned}
\mathbb{E}_{s \sim \mu^\dagger} V(s) + \lambda D_{\mathrm{TV}}(\mu^\dagger \| \mu^0) &= V_{\min} \cdot \mu^0(\mathcal{S}_\lambda) + \sum_{s \in \mathcal{S}_\lambda^c} V(s) \cdot \mu^0(s) + \lambda \mu^0(\mathcal{S}_\lambda) \\
&= \sum_{s \in \mathcal{S}_\lambda} V_{\min} + \sum_{s \in \mathcal{S}_\lambda^c} V(s) \cdot \mu^0(s) + \lambda \mu^0(\mathcal{S}_\lambda) \\
&= \sum_{s \in \mathcal{S}_\lambda} (V_{\min} + \lambda) + \sum_{s \in \mathcal{S}_\lambda^c} V(s) \cdot \mu^0(s) \\
&= \mathbb{E}_{s \sim \mu^0} \big[ V(s) \big]_{V_{\min} + \lambda},
\end{aligned}
$$

where the last equation holds by the definition of $\mathcal{S}_\lambda$. This finishes the proof. $\qquad \square$

## B. Proof of results in Section 4

### B.1. Proof of Theorem 4.2

In standard MDPs, if we assume the reward is bounded in $[0, 1]$, then the standard value function $V_h^\pi(s)$ is bounded in $[0, H]$ by definition. However, in the robust regularized MDP, (4.1) suggests that for any $(\pi, s, h)$, the robust value function $V_h^{\pi, \lambda}(s)$ is at most $\lambda + 1$, and thus is bounded in $[0, \lambda + 1]$. A similar range shrinkage phenomenon is also observed in constrained robust MDPs with TV-distance defined uncertainty set Lu et al. (2024, Proposition 2.6). We formulate this observation in the following proposition.

**Proposition B.1.** *For any $(\lambda, \pi, h) \in \mathbb{R}_+ \times \Pi \times [H]$, we have*

$$\max_{s \in \mathcal{S}} V_h^{\pi, \lambda}(s) - \min_{s \in \mathcal{S}} V_h^{\pi, \lambda}(s) \le \min\{1 + \lambda, H\}.$$

*Further, under Assumption 4.1, we have $\max_{s \in \mathcal{S}} V_h^{\pi, \lambda}(s) \le \min\{1 + \lambda, H\}$.*

*Proof of Proposition B.1.* By the robust Bellman equation (3.4) and Proposition 3.4, for any $(\pi, s, h)$

$$
\begin{aligned}
V_h^{\pi, \lambda}(s) &= \mathbb{E}_{a \sim \pi(\cdot|s)} \Big[ r_h(s, a) + \inf_{P_h(\cdot|s, a) \in \Delta(\mathcal{S})} \big\{ \lambda \cdot D_{\mathrm{TV}}(P_h(\cdot|s, a) \| P_h^0(\cdot|s, a)) \big\} \Big] \\
&= \mathbb{E}_{a \sim \pi(\cdot|s)} \Big[ r_h(s, a) + \big[ \mathbb{P}_h^0 \big[ V_{h+1}^{\pi, \lambda} \big]_{\min_{s'} V_{h+1}^{\pi, \lambda}(s') + \lambda} \big](s, a) \Big].
\end{aligned}
$$

Then we have

$$\max_{s \in \mathcal{S}} V_h^{\pi, \lambda}(s) \le 1 + \min_{s' \in \mathcal{S}} V_{h+1}^{\pi, \lambda}(s') + \lambda \text{ and } \min_{s \in \mathcal{S}} V_h^{\pi, \lambda}(s) \ge \min_{s' \in \mathcal{S}} V_{h+1}^{\pi, \lambda}(s').$$

By definition, we also have $V_h^{\pi, \lambda}(s) \le H, \forall s \in \mathcal{S}$. Thus,

$$\max_{s \in \mathcal{S}} V_h^{\pi, \lambda}(s) - \min_{s \in \mathcal{S}} V_h^{\pi, \lambda}(s) \le \min\{1 + \lambda, H\}.$$

Under Assumption 4.1, the minimal value of any value function is always 0, $\min_{s \in \mathcal{S}} V_h^{\pi, \lambda}(s) = 0$. Then we immediately have $0 \le V_h^{\pi, \lambda}(s) \le \min\{1 + \lambda, H\}$. $\qquad \square$

The logic to analyze Algorithm 1 largely follows (Jin et al., 2018), but we need careful treatments of the regret decomposition and estimation error bounds specifically tailored to the robust regularized MDP setting. Recall the learning rate is selected as $\alpha_t = \frac{H+1}{H+t}$. Accordingly, define the following quantities

$$\alpha_t^0 = \prod_{j=1}^t (1 - \alpha_j), \qquad \alpha_t^i = \alpha_i \prod_{j=i+1}^t (1 - \alpha_j). \tag{B.1}$$

(Jin et al., 2018) show that (1) $\sum_{i=1}^t \alpha_t^i = 1$ and $\alpha_t^0 = 0$ for $t \geq 1$; (2) $\sum_{i=1}^t \alpha_t^i = 0$ and $\alpha_t^0 = 1$ for $t = 0$.

For any $h \in [H]$, the robust regularized Q-function update can be summarized as follows:

$$Q_h^{k+1,\lambda}(s,a) = \begin{cases} (1-\alpha_t)Q_h^{k,\lambda}(s,a) + \alpha_t\big[r_h(s,a) + \big[V_{h+1}^{k,\lambda}(x_{h+1}^k)\big]_\lambda + b_t\big], & \text{if } (s,a) = (x_h^k, a_h^k) \\ Q_h^{k,\lambda}(s,a), & \text{otherwise.} \end{cases} \tag{B.2}$$

And the value function is updated as $V_h^{k,\lambda}(s) \leftarrow \big[\max_{a' \in \mathcal{A}} Q_h^k(s,a')\big]_{\min\{1+\lambda, H\}}, \forall s \in \mathcal{S}$.

At any $(s,a,h,k) \in \mathcal{S} \times \mathcal{A} \times [H] \times [K]$, denote the random variable $t = N_h^k(s,a)$, the number of times the state-action pair $(s,a)$ is taken until episode $k$ (including). Suppose $(s,a)$ was previously taken at step $h$ of episodes $k_1, \cdots, k_t \leq k$. By the update equation (B.2) and the definition of $\alpha_t^i$ in (B.1), we have

$$Q_h^{k,\lambda}(s,a) = \alpha_t^0 \min\{1+\lambda, H\} + \sum_{i=1}^t \alpha_t^i\big[r_h(s,a) + \big[V_{h+1}^{k_i,\lambda}(x_{h+1}^{k_i})\big]_\lambda + b_i\big]. \tag{B.3}$$

Recall that $[\mathbb{P}_h V_{h+1}](s,a) := \mathbb{E}_{s' \sim \mathbb{P}_h(\cdot|s,a)} V_{h+1}(s')$. Denote its empirical counterpart of episode $k$ as $[\hat{\mathbb{P}}_h^k V_{h+1}](s,a) := V_{h+1}(x_{h+1}^k)$, which is defined only for $(s,a) = (s_h^k, a_h^k)$.

**Lemma B.2.** *For any $(s,a,h) \in \mathcal{S} \times \mathcal{A} \times [H]$ and episode $k \in [K]$, let $t = N_h^k(s,a)$ and suppose $(s,a)$ was previously taken at step $h$ of episodes $k_1, \cdots, k_t \leq k$, then we have*

$$(Q_h^{k,\lambda} - Q_h^{\star,\lambda})(s,a) = \alpha_t^0(\min\{1+\lambda, H\} - Q_h^{\star,\lambda}(s,a)) + \sum_{i=1}^t \alpha_t^i\Big[\big(\hat{\mathbb{P}}_h^{k_i} - \mathbb{P}_h\big)\big[V_{h+1}^{\star,\lambda}\big]_\lambda(s,a)$$
$$+ \big(\big[V_{h+1}^{k_i,\lambda}\big]_\lambda - \big[V_{h+1}^{\star,\lambda}\big]_\lambda\big)(x_{h+1}^{k_i}) + b_i\Big].$$

*Proof.* From the robust Bellman optimality equation, $Q_h^{\star,\lambda}(s,a) = (r_h + \mathbb{P}_h[V_{h+1}^{\star,\lambda}]_\lambda)(s,a)$ and the fact that $\sum_{i=0}^t \alpha_t^i = 1$, we have

$$Q_H^{\star,\lambda}(s,a) = \alpha_t^0 Q_h^{\star,\lambda}(s,a) + \sum_{i=1}^t \alpha_t^i\big[r_h(s,a) + \big(\mathbb{P}_h - \hat{\mathbb{P}}_h^{k_i}\big)\big[V_{h+1}^{\star,\lambda}\big]_\lambda(s,a) + \big[V_{h+1}^{\star,\lambda}(x_{h+1}^{k_i})\big]_\lambda\big].$$

Subtracting the above equation from (B.3), we finish the proof. $\square$

**Lemma B.3.** *There exists an absolute constant $c > 0$ such that, for any $p \in (0,1)$, letting $b_t = c\min\{1+\lambda, H\}\sqrt{H\iota/t}$, we have $\beta_t = 2\sum_{i=1}^t \alpha_t^i b_i \leq 4c\min\{1+\lambda, H\}\sqrt{H\iota/t}$ and, with probability at least $1-p$, the following holds simultaneously for all $(s,a,h,k) \in \mathcal{S} \times \mathcal{A} \times [H] \times [K]$:*

$$0 \leq \big(Q_h^{k,\lambda} - Q_h^{\star,\lambda}\big)(s,a) \leq \alpha_t^0 \min\{1+\lambda, H\} + \sum_{i=1}^t \alpha_t^i\big(\big[V_{h+1}^{k_i,\lambda}\big]_\lambda - \big[V_{h+1}^{\star,\lambda}\big]_\lambda\big)(s_{h+1}^{k_i}) + \beta_t, \tag{B.4}$$

*where $t = N_h^k(s,a)$ and $k_1, \cdots, k_t \leq k$ are the episodes where $(s,a)$ was taken at step $h$.*

*Proof.* For each $(s,a,h) \in \mathcal{S} \times \mathcal{A} \times [H]$, denote $k_0 = 0$, and denote the episode of which $(s,a)$ was taken at step $h$ for the $i$th time:

$$k_i = \min\big(\{k \in [K]|k > k_{i-1} \text{ and } (s_h^k, a_h^k) = (s,a)\} \cup \{K+1\}\big). \tag{B.5}$$

Note that $k_i = K + 1$ if it is taken for fewer than $i$ times. Then the random variable $k_i$ is a stopping time. Define $\mathcal{F}_i$ be the $\sigma$-field generated by all the random variables until episode $k_i$, step $h$. Then $\{\mathbb{1}[k_i \leq K] \cdot [(\hat{\mathbb{P}}_h^{k_i} - \mathbb{P}_h)V_{h+1}^{\star,\lambda}](s,a)\}_{i=1}^{\tau}$ is a martingale difference sequence w.r.t the filtration $\{\mathcal{F}_i\}_{i \geq 0}$. Then By Azuma-Hoeffding inequality and the union bound argument, we have that with probability at least $1 - p/(SAH)$, there exist an absolute constant $c$, such that for any $\tau \in [K]$:

$$\left| \sum_{i=1}^{\tau} \alpha_{\tau}^i \cdot \mathbb{1}[k_i \leq K] \cdot \left[ (\hat{\mathbb{P}}_h^{k_i} - \mathbb{P}_h)V_{h+1}^{\star,\lambda} \right](s,a) \right| \leq \frac{c \min\{1 + \lambda, H\}}{2} \sqrt{\sum_{i=1}^{\tau} (\alpha_{\tau}^i)^2 \cdot \iota}$$

$$\leq \frac{c \min\{1 + \lambda, H\}}{2} \sqrt{\frac{H\iota}{\tau}}, \tag{B.6}$$

where the last inequality is by property (b) of Lemma C.2. Since (B.6) holds for all fixed $\tau \in [K]$ uniformly, it also holds for $\tau = t = N_h^k(s,a)$. By the standard union bound argument, for all $(s,a,h,k) \in \mathcal{S} \times \mathcal{A} \times [H] \times [K]$, with probability at least $1 - p$, we have

$$\left| \sum_{i=1}^{t} \alpha_t^i \left[ (\hat{\mathbb{P}}_h^{k_i} - \mathbb{P}_h)V_{h+1}^{\star,\lambda} \right](s,a) \right| \leq \frac{c \min\{1 + \lambda, H\}}{2} \sqrt{\frac{H\iota}{t}}, \tag{B.7}$$

where $t = N_h^k(s,a)$. If we choose $b_t = c \min\{1 + \lambda, H\}\sqrt{H\iota/t}$ for the same constant as in (B.6), then according to property (a) of Lemma C.2, we have

$$\frac{\beta_t}{2} = \sum_{i=1}^{t} \alpha_t^i b_i \in \left[ c \min\{1 + \lambda, H\}\sqrt{H\iota/t}, 2c \min\{1 + \lambda, H\}\sqrt{H\iota/t} \right]. \tag{B.8}$$

Recall that by Lemma B.2, we have

$$(Q_h^{k,\lambda} - Q_h^{\star,\lambda})(s,a) = \alpha_t^0(\min\{1 + \lambda, H\} - Q_h^{\star,\lambda}(s,a)) + \sum_{i=1}^{t} \alpha_t^i \left[ (\hat{\mathbb{P}}_h^{k_i} - \mathbb{P}_h) \left[ V_{h+1}^{\star,\lambda} \right]_{\lambda}(s,a) + \right.$$

$$\left. \left( \left[ V_{h+1}^{k_i,\lambda} \right]_{\lambda} - \left[ V_{h+1}^{\star,\lambda} \right]_{\lambda} \right)(x_{h+1}^{k_i}) + b_i \right].$$

Then the right hand side of (B.4) follows immediately from (B.7), (B.8) and the above equation. For the left hand side of (B.4), we prove by induction. At the last stage $H$, it is trivial to verify that for all $(s,a) \in \mathcal{S} \times \mathcal{A}$, we have $Q_H^{k,\lambda}(s,a) - Q_H^{\star,\lambda}(s,a) \geq 0$. Thus, by definition of $V_H^{k,\lambda}(s), V_H^{\star,\lambda}(s), a_H^k$ and Proposition B.1, we know

$$V_H^{k,\lambda}(s) - V_H^{\star,\lambda}(s) = \left[ Q_H^{k,\lambda}(s, a_H^k) \right]_{\min\{1+\lambda,H\}} - \left[ Q_H^{\star,\lambda}(s, \pi_h^{\star}(s)) \right]_{\min\{1+\lambda,H\}}$$

$$\geq \left[ Q_H^{k,\lambda}(s, \pi_h^{\star}(s)) \right]_{\min\{1+\lambda,H\}} - \left[ Q_H^{\star,\lambda}(s, \pi_h^{\star}(s)) \right]_{\min\{1+\lambda,H\}}$$

$$\geq 0,$$

and $[V_H^{k,\lambda}(s)]_{\lambda} - [V_H^{\star,\lambda}(s)]_{\lambda} \geq 0$. Then by induction on $h = H - 1, \cdots, 1$ and (B.7), the left hand side of (B.4) follows. This finishes the proof.

$\square$

Now we are ready to prove Theorem 4.2.

*Proof of Theorem 4.2.* First, we decompose the robust version of regret in a recursive form.

$$\text{Regret}(K) = \sum_{k=1}^{K} \left( V_1^{\star,\lambda} - V_1^{\pi_k,\lambda} \right)(s_1^k) \leq \sum_{k=1}^{K} \left( V_1^{k,\lambda} - V_1^{\pi_k,\lambda} \right)(s_1^k) = \sum_{k=1}^{K} \delta_1^k, \tag{B.9}$$

where $\delta_h^{k,\lambda} := \left( V_h^{k,\lambda} - V_h^{\pi_k,\lambda} \right)(s_h^k), \forall h \in [H]$.

Second, we upper bound $\sum_{k=1}^{K} \delta_h^{k,\lambda}$ by the next step $\sum_{k=1}^{K} \delta_{h+1}^{k,\lambda}$, thus inducing a recursive formula to upper bound the total regret. Specifically, we additionally define $\phi_h^k := \left(V_h^{k,\lambda} - V_h^{\star,\lambda}\right)(s_h^k)$. For any $(k,h) \in [K] \times [H]$, let $t = N_h^k(s_h^k, a_h^k)$, and suppose $(s_h^k, a_h^k)$ was previously taken at step $h$ of episodes $k_1 < \cdots < k_t \leq k$. Then by the definition of value function $V_h^{k,\lambda}(s_h^k)$, we have

$$\delta_h^k = \left(V_h^{k,\lambda} - V_h^{\pi_k,\lambda}\right)(s_h^k) \leq Q_h^{k,\lambda}(s_h^k, a_h^k) - Q_h^{\pi_k,\lambda}(s_h^k, a_h^k).$$

To continue, we decompose the difference as follows

$$\begin{aligned}
\delta_h^k &\leq \left(Q_h^{k,\lambda} - Q_h^{\pi_k,\lambda}\right)(s_h^k, a_h^k) \\
&= \left(Q_h^{k,\lambda} - Q_h^{\star,\lambda}\right)(s_h^k, a_h^k) + \left(Q_h^{\star,\lambda} - Q_h^{\pi_k,\lambda}\right)(s_h^k, a_h^k) \\
&\leq \alpha_t^0 \min\{1 + \lambda, H\} + \sum_{i=1}^{t} \alpha_t^i \left(\left[V_{h+1}^{k_i,\lambda}\right]_\lambda - \left[V_{h+1}^{\star,\lambda}\right]_\lambda\right)(s_{h+1}^{k_i}) + \beta_t + \left[\mathbb{P}_h\left(\left[V_{h+1}^{\star,\lambda}\right]_\lambda - \left[V_{h+1}^{\pi_k,\lambda}\right]_\lambda\right)\right](s_h^k, a_h^k).
\end{aligned}$$

The last inequality follows from Lemma B.2 and the robust Bellman equation (4.1). For any $(s,h) \in \mathcal{S} \times [H]$, we know

$$\begin{aligned}
V_{h+1}^{k_i,\lambda}(s) &= \left[Q_{h+1}^{k,\lambda}(s, a_{h+1}^k)\right]_{\min\{1+\lambda, H\}} \\
&\geq \left[Q_{h+1}^{k,\lambda}(s, \pi_{h+1}^\star(s))\right]_{\min\{1+\lambda, H\}} \\
&\geq \left[Q_{h+1}^{\star,\lambda}(s, \pi_{h+1}^\star(s))\right]_{\min\{1+\lambda, H\}} = V_{h+1}^{\star,\lambda}(s),
\end{aligned}$$

where the last inequality holds by Lemma B.2 and the last equation holds by Proposition B.1. Then we can upper bound the differences between those clipped value functions by the differences between original value functions,

$$\begin{aligned}
\delta_h^k &\leq \alpha_t^0 \min\{1 + \lambda, H\} + \sum_{i=1}^{t} \alpha_t^i \left(V_{h+1}^{k_i,\lambda} - V_{h+1}^{\star,\lambda}\right)(s_{h+1}^{k_i}) + \beta_t + \left[\mathbb{P}_h\left(V_{h+1}^{\star,\lambda} - V_{h+1}^{\pi_k,\lambda}\right)\right](s_h^k, a_h^k) \\
&= \alpha_t^0 \min\{1 + \lambda, H\} + \sum_{i=1}^{t} \alpha_t^i \phi_{h+1}^{k_i,\lambda} + \beta_t - \phi_{h+1}^{k,\lambda} + \delta_{h+1}^{k,\lambda} + \xi_{h+1}^{k,\lambda},
\end{aligned} \tag{B.10}$$

where $\phi_h^{k,\lambda} := \left(V_h^{k,\lambda} - V_h^{\star,\lambda}\right)(s_h^k)$ and $\xi_{h+1}^{k,\lambda} := \left[\left(\mathbb{P}_h - \hat{\mathbb{P}}_h^k\right)\left(V_{h+1}^{\star,\lambda} - V_{h+1}^{k,\lambda}\right)\right](s_h^k, a_h^k)$ is a martingale difference sequence. The last equation holds due to the observation $\delta_{h+1}^{k,\lambda} - \phi_{h+1}^{k,\lambda} = \left(V_{h+1}^{\star,\lambda} - V_{h+1}^{\pi_k,\lambda}\right)(s_{h+1}^k)$. Next, we calculate the summation $\sum_{k=1}^{K} \delta_h^k$. Denote $n_h^k = N_h^k(s_h^k, a_h^k)$, we have

$$\sum_{k=1}^{K} \alpha_{n_h^k}^0 \min\{1 + \lambda, H\} = \sum_{k=1}^{K} H \cdot \mathbb{1}[n_h^k = 0] \leq SA \min\{1 + \lambda, H\}.$$

For the term $\sum_{k=1}^{K} \sum_{i=1}^{n_h^k} \alpha_{n_h^k}^i \phi_{h+1}^{k_i(s_h^k, a_h^k)}$, the same regrouping argument in the proof of Theorem 1 in (Jin et al., 2018) directly leads to the following inequality

$$\sum_{k=1}^{K} \sum_{i=1}^{n_h^k} \alpha_{n_h^k}^i \phi_{h+1}^{k_i(s_h^k, a_h^k)} \leq \sum_{k'=1}^{K} \phi_{h+1}^{k'} \sum_{t=n_h^{k'}+1}^{\infty} \alpha_t^{n_h^{k'}} \leq \left(1 + \frac{1}{H}\right) \sum_{k=1}^{K} \phi_{h+1}^k.$$

Plugging these back into (B.10), we have:

$$\begin{aligned}
\sum_{k=1}^{K} \delta_h^k &\leq SA \min\{1 + \lambda, H\} + \left(1 + \frac{1}{H}\right) \sum_{k=1}^{K} \phi_{h+1}^k + \sum_{k=1}^{K} \left(\beta_{n_h^k} + \xi_{h+1}^k\right) + \sum_{k=1}^{K} \delta_{h+1}^k - \sum_{k=1}^{K} \phi_{h+1}^k \\
&\leq SA \min\{1 + \lambda, H\} + \left(1 + \frac{1}{H}\right) \sum_{k=1}^{K} \delta_{h+1}^k + \sum_{k=1}^{K} \left(\beta_{n_h^k} + \xi_{h+1}^k\right),
\end{aligned} \tag{B.11}$$

where the final inequality holds due to the fact that $V_h^{\star,\lambda} \geq V_h^{\pi_k,\lambda}$ and thus $\phi_{h+1}^k \leq \delta_{h+1}^k$. Recursing the result for $h = 1, 2, \cdots, H$ and using the fact that $\delta_{H+1}^K \equiv 0$, we have

$$\sum_{k=1}^K \delta_1^k \leq O\left( H\min\{1+\lambda, H\}SA + \sum_{h=1}^H \sum_{k=1}^K \left(\beta_{n_h^k} + \xi_{h+1}^k\right)\right).$$

Finally, by the pigeonhole principle, for any $h \in [H]$:

$$\sum_{k=1}^K \beta_{n_h^k} \leq O(1) \cdot \sum_{k=1}^K \min\{1+\lambda, H\}\sqrt{\frac{H\iota}{n_h^k}}$$

$$= O(\min\{1+\lambda, H\}) \cdot \sum_{s,a} \sum_{n=1}^{N_h^k(s,a)} \sqrt{\frac{H\iota}{n}}$$

$$\leq O(\min\{1+\lambda, H\}\sqrt{SAHK\iota}) \tag{B.12}$$

where the last inequality is true because $\sum_{s,a} N_h^K(s,a) = K$ and the left hand side of the last inequality is maximized when $N_h^K(s,a) = K/SA$ for all $s, a$. Also, by the Azuma-Hoeffding inequality, with probability at least $1 - p$, we have

$$\left| \sum_{h=1}^H \sum_{k=1}^K \xi_{h+1}^k \right| = \left| \sum_{h=1}^H \sum_{k=1}^K \left[ \left(\mathbb{P}_h - \hat{\mathbb{P}}_h^k\right)\left(V_h^{\star,\lambda} - V_{h+1}^{k,\lambda}\right)\right](s_h^k, a_h^k) \right| \leq c\min\{1+\lambda, H\}\sqrt{HK\iota}.$$

This establishes $\sum_{k=1}^K \delta_1^k \leq O\big(\min\{1+\lambda, H\}HSA + \min\{1+\lambda, H\}\sqrt{SAH^3K\iota}\big)$. Note that when $HK \geq \min\{1+\lambda, H\}\sqrt{SAH^3K\iota}$, we have $\min\{1+\lambda, H\}\sqrt{SAH^3K\iota} \geq \min\{1+\lambda, H\}HSA$; and when $HK \leq \min\{1+\lambda, H\}\sqrt{SAH^3K\iota}$, we have $\sum_{k=1}^K \delta_1^k \leq \min\{1+\lambda, H\}K \leq HK \leq \{1+\lambda, H\}\sqrt{SAH^3K\iota}$. Therefore, we can remove the $\min\{1+\lambda, H\}HSA$ term in the regret upper bound. Then with probability at least $1 - 2p$, we have $\sum_{k=1}^K \delta_1^k \leq O\big(\min\{1+\lambda, H\}\sqrt{SAH^3K\iota}\big)$. We finish the proof by rescaling $p$ to $p/2$. $\qquad\square$

### B.2. Proof of Theorem 4.3

---

**Algorithm 2** Clipped Q-Learning with UCB-Bernstein

---

**Require:** regularizer $\lambda$, constant $c_1, c_2$
1: **Initialize** $Q_h^{1,\lambda}(s,a) \leftarrow H, N_h(s,a) \leftarrow 0, \mu_h(s,a) \leftarrow 0, \sigma_h(s,a) \leftarrow 0, \beta_0(s,a,h) \leftarrow 0$ for all $(s,a,h) \in \mathcal{S} \times \mathcal{A} \times [H]$.
2: **for** episode $k = 1, \cdots, K$ **do**
3:     Receive $s_1^k$.
4:     **for** step $h = 1, \cdots, H$ **do**
5:         Take action $a_h^k \leftarrow \operatorname{argmax}_{a' \in \mathcal{A}} Q_h^k(s_h, a')$ and observe $s_{h+1}^k$.
6:         $N_h^k(s_h^k, a_h^k) = N_h(s_h^k, a_h^k) \leftarrow N_h(s_h^k, a_h^k) + 1; t \leftarrow N_h^k(s_h^k, a_h^k)$
7:         $\mu_h(s_h^k, a_h^k) \leftarrow \mu_h(s_h^k, a_h^k) + V_{h+1}^k(s_{h+1}^k)$
8:         $\sigma_h(s_h^k, a_h^k) \leftarrow \sigma_h(s_h^k, a_h^k) + (V_{h+1}^k(s_{h+1}^k))^2$
9:         $\beta_t(s_h^k, a_h^k, h) \qquad \leftarrow \qquad \min\Big\{ c_1\Big(\sqrt{\frac{H}{t} \cdot \frac{\sigma_h(s_h^k, a_h^k) - (\mu_h(s_h^k, a_h^k))^2}{t}} + \min\{1+\lambda, H\}\Big)\iota \quad + \quad \min\{1 \quad +$
        $\lambda, H\}^{3/2}\frac{\sqrt{SAH^4}}{t}\iota\Big), c_2 \min\{1+\lambda, H\}\sqrt{\frac{H\iota}{t}}\Big\}$
10:       $b_t(s_h^k, a_h^k, h) \leftarrow \frac{\beta_t(s_h^k, a_h^k, h) - (1-\alpha_t)\beta_{t-1}(s_h^k, a_h^k, h)}{2\alpha_t}$
11:       $Q_h^{k+1,\lambda}(s_h^k, a_h^k) \leftarrow (1-\alpha_t)Q_h^{k,\lambda}(s_h^k, a_h^k) + \alpha_t\big[r_h(s_h^k, a_h^k) + \big[V_{h+1}^{k,\lambda}(s_{h+1}^k)\big]_\lambda + b_t\big]$
12:       $V_h^{k+1,\lambda}(s_h^k) \leftarrow \big[\max_{a' \in \mathcal{A}} Q_h^{k+1,\lambda}(s_h, a')\big]_{\min\{1+\lambda, H\}}$
13:     **end for**
14: **end for**

---

Recall the variance operator $[\mathrm{Var}_h V](s,a) = \mathrm{Var}_{s \sim \mathbb{P}_h(\cdot|s,a)}[V(s)] = [\mathbb{P}_h V^2](s,a) - ([\mathbb{P}_h V](s,a))^2$. The empirical version of this variance operator that is calculated by Algorithm 2 is defined as follows: when $(s,a)$ was taken at step $h$ for $t$ times

at $k_1, , k_t$ episodes respectively

$$W_t(s,a) := \frac{1}{t} \sum_{i=1}^{t} \left[ V_{h+1}^{k_i}(s_{h+1}^{k_i}) - \frac{1}{t} \sum_{j=1}^{t} V_{h+1}^{k_j}(x_{h+1}^{k_j}) \right]^2$$

For some absolute constants $c_1, c_2 > 0$, we define

$$\beta_t(s,a,h) := \min \left\{ c_1 \left( \sqrt{\frac{H}{t} \cdot (W_t(s,a,h) + \min\{1+\lambda, H\})\iota} + \min\{1+\lambda, H\}^{3/2} \frac{\sqrt{SAH^4}}{t}\iota \right), \right.$$

$$\left. c_2 \min\{1+\lambda, H\} \sqrt{\frac{H\iota}{t}} \right\}, \tag{B.13}$$

and accordingly,

$$b_1(s,a,h) := \frac{\beta_1(s,a,h)}{2}, \qquad b_t(s,a,h) := \frac{\beta_t(s,a,h) - (1-\alpha_t)\beta_{t-1}(s,a,h)}{2\alpha_t}.$$

It is easy to verify $\beta_t = 2 \sum_{i=1}^{t} \alpha_t^i b_i$ for every $t \geq 1$.

We first note that the following lemma, obtained from Lemma B.2, still holds here.

**Lemma B.4.** *For any $(s,a,h) \in \mathcal{S} \times \mathcal{A} \times [H]$ and episode $k \in [K]$, let $t = N_h^k(s,a)$ and suppose $(s,a)$ was previously taken at step $h$ of episodes $k_1, \cdots, k_t \leq k$, then we have*

$$(Q_h^{k,\lambda} - Q_h^{\star,\lambda})(s,a) = \alpha_t^0 (\min\{1+\lambda, H\} - Q_h^{\star,\lambda}(s,a)) + \sum_{i=1}^{t} \alpha_t^i \left[ (\hat{\mathbb{P}}_h^{k_i} - \mathbb{P}_h) [V_{h+1}^{\star,\lambda}]_\lambda (s,a) \right.$$

$$\left. + ([V_{h+1}^{k_i,\lambda}]_\lambda - [V_{h+1}^{\star,\lambda}]_\lambda)(x_{h+1}^{k_i}) + b_i(s,a,h) \right].$$

By Lemma B.4 and proof of Lemma B.3, we immediately have the following coarse bound on $Q_h^{k,\lambda} - Q_h^{\star,\lambda}$.

**Lemma B.5.** *There exists an absolute constant $c_2 > 0$ such that, if $\beta_t(s,a,h) \leq c_2 \min\{1+\lambda, H\}\sqrt{H\iota/t}$, then with probability at least $1-p$, the following holds for all $(s,a,h,k) \in \mathcal{S} \times \mathcal{A} \times [H] \times [K]$*

$$(Q_h^{k,\lambda} - Q_h^{\star,\lambda})(s,a)$$

$$\leq \alpha_t^0 \min\{1+\lambda, H\} + \sum_{i=1}^{t} \alpha_t^i ([V_{h+1}^{k_i,\lambda}]_\lambda - [V_{h+1}^{\star,\lambda}]_\lambda)(s_{h+1}^{k_i}) + 4c_2 \min\{1+\lambda, H\}\sqrt{\frac{H\iota}{t}}, \tag{B.14}$$

*where $t = N_h^k(s,a)$ and $k_1, \cdots, k_t \leq k$ are episodes in which $(s,a)$ was taken at step $h$.*

We compute the empirical variance of $V_h^{k,\lambda}$ to estimate the variance of $V_h^{\star,\lambda}$, which is unknown. The next lemma shows that, if $Q_h^{k',\lambda} - Q_h^{\star,\lambda}$ is nonnegative for all episodes $k' \leq k$, the variance of $V_h^{\star,\lambda}$ and the empirical variance of $V_h^{k,\lambda}$ are sufficiently close.

**Lemma B.6.** *There exists an absolute constant $c > 0$ such that for any $p \in (0,1)$ and $k \in [K]$, with probability at least $1 - p/K$, if (B.14) holds and $Q_h^{k',\lambda} - Q_h^{\star,\lambda} \geq 0$ for all $k' \leq k$, then for all $(s,a,h) \in \mathcal{S} \times \mathcal{A} \times [H]$:*

$$\left| \mathbb{V}_h V_{h+1}^{\star,\lambda} - W_t(s,a,h) \right| \leq c \left( \frac{SA \min\{1+\lambda, H\}^2 \sqrt{H^3 \iota}}{t} + \min\{1+\lambda, H\}^2 \sqrt{\frac{SAH^3 \iota}{t}} \right),$$

*where $t = N_h^k(s,a)$.*

*Proof.* For each $(s,a,h) \in \mathcal{S} \times \mathcal{A} \times [H]$, denote $k_0 = 0$, and denote the episode of which $(s,a)$ was taken at step $h$ for the $i$th time:

$$k_i = \min \left( \{k \in [K]|k > k_{i-1} \text{ and } (s_h^k, a_h^k) = (s,a)\} \cup \{K+1\} \right).$$

Note that $k_i = K + 1$ if it is taken for fewer than $i$ times. Then the random variable $k_i$ is a stopping time. Define $\mathcal{F}_i$ be the $\sigma$-field generated by all the random variables until episode $k_i$, step $h$.

To bridge the gap between $\mathbb{V}_h V_{h+1}^\star(s, a)$ and $W_t(s, a, h)$, we consider following four quantities:

$$\mathbb{V}_h V_{h+1}^{\star,\lambda}(s, a) = \mathbb{E}_{s' \sim \mathbb{P}_h(\cdot|s,a)} \left[ V_{h+1}^{\star,\lambda}(s') - \left[ \mathbb{P}_h V_{h+1}^{\star,\lambda} \right](s, a) \right]^2 \tag{B.15}$$

$$\frac{1}{t} \sum_{i=1}^{t} \left[ V_{h+1}^{\star,\lambda}(s_{h+1}^{k_i}) - \left[ \mathbb{P}_h V_{h+1}^{\star,\lambda} \right](s, a) \right]^2 \tag{B.16}$$

$$\frac{1}{t} \sum_{i=1}^{t} \left[ V_{h+1}^{\star,\lambda}(s_{h+1}^{k_i}) - \frac{1}{t} \sum_{j=1}^{t} V_{h+1}^{\star,\lambda}(s_{h+1}^{k_j}) \right]^2 \tag{B.17}$$

$$W_t(s, a, h) = \frac{1}{t} \sum_{i=1}^{t} \left[ V_{h+1}^{k_i,\lambda}(s_{h+1}^{k_i}) - \frac{1}{t} \sum_{j=1}^{t} V_{h+1}^{k_j,\lambda}(s_{h+1}^{k_j}) \right]^2 \tag{B.18}$$

Then we shall bound $|\mathbb{V}_h V_{h+1}^\star(s, a) - W_t(s, a, h)|$ by $|(\text{B.15}) - (\text{B.16})| + |(\text{B.16}) - (\text{B.17})| + |(\text{B.17}) - (\text{B.18})|$ via the triangle inequality.

**Bound $|(\text{B.15}) - (\text{B.16})|$:** For any fixed $\tau \in [k]$, by the Azuma-Hoeffding inequality, there exist a sufficiently large constant $c > 0$ such that, with probability at least $1 - p/(2SAHK)$:

$$\left| \frac{1}{\tau} \sum_{i=1}^{\tau} \mathbb{1}[k_i \leq k] \cdot \left[ \left( V_{h+1}^\star(s_{h+1}^{k_i}) - \left[ \mathbb{P}_h V_{h+1}^\star \right](s, a) \right)^2 - \left[ \mathbb{V}_h V_{h+1} \right](s, a) \right] \right| \leq c \min\{1 + \lambda, H\}^2 \sqrt{\iota/\tau}, \tag{B.19}$$

since LHS is a martingale sequence with respect to the filtration $\{\mathcal{F}_i\}$. Because (B.19) holds for all fixed $\tau \in [k]$ uniformly, it also holds for $\tau = t = N_h^k(s, a) \leq k$ which is a random variable. Also note $\mathbb{1}[k_i \leq k] = 1$ for all $i \leq N_h^k(s, a)$. Therefore, we can conclude —(B.15) - (B.16)— $\leq c \min\{1 + \lambda, H\}^2 \sqrt{\iota/t}$.

**Bound $|(\text{B.16}) - (\text{B.17})|$:** We calculate

$$\left| \frac{1}{t} \sum_{i=1}^{t} \left[ V_{h+1}^{\star,\lambda}(s_{h+1}^{k_i}) - \left[ \mathbb{P}_h V_{h+1}^{\star,\lambda} \right](s, a) \right]^2 - \frac{1}{t} \sum_{i=1}^{t} \left[ V_{h+1}^{\star,\lambda}(s_{h+1}^{k_i}) - \frac{1}{t} \sum_{j=1}^{t} V_{h+1}^{\star,\lambda}(s_{h+1}^{k_j}) \right]^2 \right|$$

$$\leq \frac{2 \min\{1 + \lambda, H\}}{t} \sum_{i=1}^{t} \left| \left[ \mathbb{P}_h V_{h+1}^{\star,\lambda} \right](s, a) - \frac{1}{t} \sum_{j=1}^{t} V_{h+1}^{\star,\lambda}(s_{h+1}^{k_j}) \right|$$

$$= 2 \min\{1 + \lambda, H\} \left| \left[ \mathbb{P}_h V_{h+1}^{\star,\lambda} \right](s, a) - \frac{1}{t} \sum_{j=1}^{t} V_{h+1}^{\star,\lambda}(s_{h+1}^{k_j}) \right|.$$

Again, for any fixed $\tau \in [k]$, by the Azuma-Hoeffding inequality, with probability $1 - p/(2SAHK)$:

$$\left| \frac{1}{\tau} \sum_{i=1}^{\tau} \mathbb{1}\left[k_i \leq k\right] \cdot \left[ V_{h+1}^{\star,\lambda}(s_{h+1}^{k_i}) - \left[ \mathbb{P}_h V_{h+1}^{\star,\lambda} \right](s, a) \right] \right| \leq cH\sqrt{\iota/\tau}. \tag{B.20}$$

By the same argument as above, we also know that (B.20) holds for the random variable $\tau = t = N_h^k(s, a) \leq k$, which implies $|(\text{B.16}) - (\text{B.17})| \leq c \min\{1 + \lambda, H\} \sqrt{\iota/t}$.

**Bound $|$(B.17) - (B.18)$|$:** We calculate that

$$\frac{1}{t}\sum_{i=1}^{t}\left[V_{h+1}^{\star,\lambda}(s_{h+1}^{k_i}) - \frac{1}{t}\sum_{j=1}^{t}V_{h+1}^{\star,\lambda}(s_{h+1}^{k_j})\right]^2 - \frac{1}{t}\sum_{i=1}^{t}\left[V_{h+1}^{k_i,\lambda}(s_{h+1}^{k_i}) - \frac{1}{t}\sum_{j=1}^{t}V_{h+1}^{k_j,\lambda}(s_{h+1}^{k_j})\right]^2$$

$$\leq \frac{2\min\{1+\lambda,H\}}{t}\sum_{i=1}^{t}\left|V_{h+1}^{k_i,\lambda}(s_{h+1}^{k_i}) - V_{h+1}^{\star,\lambda}(s_{h+1}^{k_i}) - \frac{1}{t}\sum_{j=1}^{t}\left(V_{h+1}^{k_j}(s_{h+1}^{k_j}) - V_{h+1}^{\star}(s_{h+1}^{k_j})\right)\right|$$

$$\leq \frac{4\min\{1+\lambda,H\}}{t}\sum_{i=1}^{t}\left|V_{h+1}^{k_i,\lambda}(s_{h+1}^{k_i}) - V_{h+1}^{\star,\lambda}(s_{h+1}^{k_i})\right|$$

$$= \frac{4\min\{1+\lambda,H\}}{t}\sum_{i=1}^{t}\left(V_{h+1}^{k_i,\lambda}(s_{h+1}^{k_i}) - V_{h+1}^{\star,\lambda}(s_{h+1}^{k_i})\right),$$

where the last equation uses $V_{h+1}^{k_i,\lambda}(s) \geq V_{h+1}^{\star,\lambda}(s)$ for all $s \in \mathcal{S}$ and $k' \leq k$, which follows from our assumption that $Q_{h+1}^{k',\lambda}(s) \geq Q_{h+1}^{\star,\lambda}(s,a)$ for all $k' \leq k$.

Applying Lemma C.3 with a weight vector $w$ such that $w_k = 1/t$ for all $i \in [t]$, but $w_{k'} = 0$ for all $k' \notin \{k_1, \cdots, k_t\}$. This tells us that

$$\frac{4\min\{1+\lambda,H\}}{t}\sum_{i=1}^{t}\left(V_{h+1}^{k_i,\lambda}(s_{h+1}^{k_i}) - V_{h+1}^{\star,\lambda}(s_{h+1}^{k_i})\right)$$

$$\leq \frac{4\min\{1+\lambda,H\}}{t}\sum_{i=1}^{t}\left(V_{h+1}^{k_i,\lambda}(s_{h+1}^{k_i}) - V_{h+1}^{\star,\lambda}(s_{h+1}^{k_i})\right)$$

$$\leq O\left(\frac{SA\min\{1+\lambda,H\}^2\sqrt{H^3\iota}}{t} + \min\{1+\lambda,H\}^2\sqrt{\frac{SAH^3\iota}{t}}\right).$$

Finally, by the triangle inequality and a union bound over $(s,a,h) \in \mathcal{S} \times \mathcal{A} \times [H]$, we finish the proof. $\qquad\square$

Equipped with Lemma B.5 and Lemma B.6, we can use induction and Azuma-Bernstein concentration argument to prove that $Q_h^{k,\lambda} - Q_h^{\star,\lambda}$ is non-negative and upper bounded by $\beta$, an analog as Lemma B.3. We formally state it in the following Lemma.

**Lemma B.7.** *For every $p \in (0,1)$, there exists an absolute constant $c_1, c_2 > 0$ such that, under the choice of $\beta_t(s,a,h)$ in (B.13), with probability at least $1 - 2p$, the following holds simultaneously for all $(s,a,h,k) \in \mathcal{S} \times \mathcal{A} \times [H] \times [K]$:*

$$0 \leq (Q_h^{k,\lambda} - Q_h^{\star,\lambda})(s,a) \leq \alpha_t^0 \min\{1+\lambda,H\} + \sum_{i=1}^{t}\alpha_t^i(V_{h+1}^{k_i,\lambda} - V_{h+1}^{\star})(s_{h+1}^{k_i}) + \beta_t, \tag{B.21}$$

*where $t = N_h^k(s,a)$ and $k_1, \cdots, k_t \leq k$ are the episodes in which $(s,a)$ was taken at step $h$.*

*Proof.* We first choose $c_2 > 0$ large enough so that Lemma B.5 holds with probability at least $1 - p$. For each fixed $(s,a,h) \in \mathcal{S} \times \mathcal{A} \times [H]$, let us denote $k_0 = 0$, and

$$k_i = \min\left(\{k \in [K] | k > k_{i-1} \text{ and } (s_h^k, a_h^k) = (s,a)\} \cup \{K+1\}\right).$$

By the Azuma-Bernstein inequality, with probability at least $1 - p/(SAT)$, we have for all $\tau \in [K]$:

$$\left|\sum_{i=1}^{\tau}\alpha_\tau^i \mathbb{1}[k_i \leq K] \cdot \left[(\hat{\mathbb{P}}_h^{k_i} - \mathbb{P}_h)V_{h+1}^{\star,\lambda}\right](s,a)\right|$$

$$\leq O(1) \cdot \left[\sqrt{\sum_{i=1}^{\tau}(\alpha_\tau^i)^2\left[\mathbb{V}_h V_{h+1}^{\star,\lambda}\right](s,a)\iota} + \max_{i\in[\tau]}\alpha_\tau^i \cdot \min\{1+\lambda,H\}\iota\right]$$

$$\leq O(1) \cdot \left[\sqrt{\frac{H}{\tau}\left[\mathbb{V}_h V_{h+1}^{\star,\lambda}\right](s,a)\iota} + \frac{\min\{1+\lambda,H\}H}{\tau}\iota\right], \tag{B.22}$$

where the last inequality holds by (b) of Lemma C.2. Since the inequality Equation (B.22) holds for all fixed $\tau \in [K]$ uniformly, it also holds for the random variable $\tau = t = N_h^k(s, a) \le K$. By a union bound, with probability at least $1 - p$, we have that for all $(s, a, h, k) \in \mathcal{S} \times \mathcal{A} \times [H] \times [K]$,

$$\left| \sum_{i=1}^{t} \alpha_\tau^i \mathbb{1}[k_i \le K] \cdot \left[ (\hat{\mathbb{P}}_h^{k_i} - \mathbb{P}_h) V_{h+1}^{\star,\lambda} \right](s, a) \right| \le O(1) \cdot \left[ \sqrt{\frac{H}{\tau} \left[ \mathbb{V}_h V_{h+1}^{\star,\lambda} \right](s, a) \iota} + \frac{\min\{1 + \lambda, H\} H}{\tau} \iota \right], \quad \text{(B.23)}$$

where $t = N_h^k(s, a)$ and $k_1, \cdots, k_t \le k$ are episodes in which $(s, a)$ was taken at step $h$. Now, we prove (B.21) by induction. When $k = 1$, the statement clearly holds. In the rest of the proof we assume (B.21) holds for all $k' \le k$. We denote by $k_1, k_2, \cdots, k_t \le k$ all indices of previous episodes where $(s, a)$ is taken at step $h$. By Lemma B.6, with probability at least $1 - p/K$, we have for all $(s, a, h) \in \mathcal{S} \times \mathcal{A} \times [H]$:

$$\left| \left[ \mathbb{V}_h V_{h+1}^{\star,\lambda}(s, a) - W_t(s, a, h) \right] \right| \le O\left( \min\{1 + \lambda, H\}^2 \sqrt{\frac{SAH^3\iota}{t}} + \frac{SA\min\{1 + \lambda, H\}^2 \sqrt{H^3\iota}}{t} \right).$$

Therefore, putting this back into (B.22), we have

$$\left| \sum_{i=1}^{t} \alpha_t^i \left[ (\hat{\mathbb{P}}_h^{k_i} - \mathbb{P}_h) V_{h+1}^{\star,\lambda}(s, a) \right] \right|$$

$$\le O(1) \cdot \left[ \sqrt{\frac{H}{t} \left( W_t(s, a, h) + \min\{1 + \lambda, H\}^2 \sqrt{\frac{SAH^3\iota}{t}} + \min\{1 + \lambda, H\}^2 \frac{SA\sqrt{H^3\iota}}{t} \right) \iota} \right.$$

$$\left. + \frac{\min\{1 + \lambda, H\} H}{t} \iota \right]$$

$$\le O(1) \cdot \left[ \sqrt{\frac{H}{t} \left( W_t(s, a, h) + \min\{1 + \lambda, H\} + \min\{1 + \lambda, H\}^3 \frac{SAH^3\iota}{t} + \min\{1 + \lambda, H\}^2 \frac{SA\sqrt{H^3\iota}}{t} \right) \iota} \right. \quad \text{(B.24)}$$

$$\left. + \frac{\min\{1 + \lambda, H\} H}{t} \iota \right]$$

$$\le O(1) \cdot \left[ \sqrt{\frac{H}{t} \left( W_t(s, a, h) + \min\{1 + \lambda, H\} \right) \iota + \min\{1 + \lambda, H\}^3 \frac{SAH^4\iota^2}{t^2}} + \frac{\min\{1 + \lambda, H\} H}{t} \iota \right]$$

$$\le O(1) \cdot \left[ \sqrt{\frac{H}{t} \left( W_t(s, a, h) + \min\{1 + \lambda, H\} \right) \iota} + \min\{1 + \lambda, H\}^{3/2} \frac{\sqrt{SAH^4}}{t} \iota \right], \quad \text{(B.25)}$$

where the second inequality uses $\sqrt{\frac{\min\{1+\lambda, H\}^4 SAH^3\iota}{t}} \le \min\{1 + \lambda, H\} + \min\{1 + \lambda, H\}^3 H^3 SA\iota/t$, and the last inequality uses $\sqrt{a + b} \le \sqrt{a} + \sqrt{b}$ for any $a, b > 0$. According to (B.7), (B.25) and the choice of $\beta_t$, we have

$$\left| \sum_{i=1}^{t} \alpha_t^i \left[ (\hat{\mathbb{P}}_h^{k_i} - \mathbb{P}_h) V_{h+1}^{\star,\lambda}(s, a) \right] \right| \le \frac{\beta_t}{2}.$$

Finally, applying the above inequality to Lemma B.4, we have for all $(s, a, h) \in \mathcal{S} \times \mathcal{A} \times [H]$,

$$0 \le \left( Q_h^k - Q_h^\star \right)(s, a) - \alpha_t^0 \left( H - Q_h^\star(s, a) \right) - \sum_{i=1}^{t} \alpha_t^i \left[ \left( V_{h+1}^{k_i} - V_{h+1}^\star \right)(s_{h+1}^{k_i}) \right] \le \beta_t.$$

This proves that (B.21) holds for $k$ with probability at least $1 - p/K$. By induction, we know (B.21) holds for all $k \in [K]$ with probability at least $1 - p$. Combining this with the $1 - p$ probability event for (B.23), we finish the proof that Lemma B.7 holds with probability at least $1 - 2p$. □

In the following Lemma, we show that the total variance of the value function for an entire episode is at most $O(\min\{1 + \lambda, H\}H)$, and the total variance for all steps is at most $O(\min\{1 + \lambda, H\}HK)$.

**Lemma B.8.** *There exists an absolute constant c, such that with probability at least $1 - p$:*

$$\sum_{k=1}^{K}\sum_{h=1}^{H} \mathbb{V}_h V_{h+1}^{\pi_k,\lambda}(s_h^k, a_h^k) \le c\big( \min\{1+\lambda, H\}HK + H\min\{1+\lambda, H\}^3\iota\big).$$

*Proof.* Consider the following decomposition

$$\sum_{k=1}^{K}\sum_{h=1}^{H} \mathbb{V}_h V_{h+1}^{\pi_k,\lambda}(s_h^k, a_h^k)$$

$$= \underbrace{\sum_{k=1}^{K}\sum_{h=1}^{H} \mathbb{V}_h V_{h+1}^{\pi_k,\lambda}(s_h^k, a_h^k) - \mathbb{E}_{(s_h^k,a_h^k)\sim(P^0,\pi^k),\forall h\in[H]}\left[\sum_{h=1}^{H}\mathbb{V}_{s\sim P_h^0(\cdot|s_h^k,a_h^k)}\big[V_{h+1}^{\pi^k,\lambda}(s)\big]\right]}_{\text{Term (i): martingale difference term}}$$

$$+ \underbrace{\sum_{k=1}^{K} \mathbb{E}_{(s_h,a_h)\sim(P^0,\pi^k),\forall h\in[H]}\left[\sum_{h=1}^{H}\mathbb{V}_{s\sim P_h^0(\cdot|s_h^k,a_h^k)}\big[V_{h+1}^{\pi^k,\lambda}(s)\big]\right]}_{\text{total variance law}}. \tag{B.26}$$

**Term (i): martingale difference term.** Defined a filtration $\mathcal{G}_k = \sigma\big(\{s_h^\tau, a_h^\tau\}_{(h,\tau)\in[H]\times[k]}\big)$. Then Term (i) is a summation of martingale difference term with respect to $\mathcal{G}_k$. By Azuma-Hoeffding's inequality, there exists an absolute constant $c > 0$, with probability at least $1 - p$, we have

$$\text{Term (i)} \le c \cdot H \cdot \min\{1+\lambda, H\}^2\sqrt{K\iota}. \tag{B.27}$$

We have utilize Proposition B.1 to obtain the upper bound $H\min\{1+\lambda, H\}^2$ on each martingale difference term in the summation.

**Term (ii): total variance law.** For any deterministic policy $\pi$, consider the variance

$$\mathbb{V}_{s\sim P_h^0(\cdot|s_h,a_h)}\big[V_{h+1}^{\pi,\lambda}(s)\big] = \mathbb{E}_{s\sim P_h^0(\cdot|s_h,a_h)}\big[\big(V_{h+1}^{\pi,\lambda}(s)\big)^2\big] - \big(\mathbb{E}_{s\sim P_h^0(\cdot|s_h,a_h)}\big[V_{h+1}^{\pi,\lambda}(s)\big]\big)^2. \tag{B.28}$$

By the robust bellman equation Equation (4.1), we know

$$V_h^{\pi,\lambda}(s_h) = r_h(s_h, \pi_h(s_h)) + \mathbb{E}_{s\sim P_h^0(\cdot|s_h,a_h)}\big[\big[V_{h+1}^{\pi,\lambda}(s)\big]_\lambda\big] \le r_h(s_h, \pi_h(s_h)) + \mathbb{E}_{s\sim P_h^0(\cdot|s_h,a_h)}\big[V_{h+1}^{\pi,\lambda}(s)\big],$$

then we have

$$\mathbb{E}_{s\sim P^0(\cdot|s_h,a_h)}\big[V_{h+1}^{\pi,\lambda}(s)\big] \ge V_h^{\pi,\lambda}(s_h) - r_h(s_h, \pi_h(s_h)). \tag{B.29}$$

Substitute (B.29) into (B.28), we have

$$\mathbb{V}_{s\sim P_h^0(\cdot|s_h,a_h)}\big[V_{h+1}^{\pi,\lambda}(s)\big] \le \mathbb{E}_{s\sim P_h^0(\cdot|s_h,a_h)}\big[\big(V_{h+1}^{\pi,\lambda}(s)\big)^2\big] - \big(V_h^{\pi,\lambda}(s_h) - r_h(s_h, \pi_h(s_h))\big)^2.$$

Next, we bound the second term on the RHS of the above inequality.

$$-\big(V_h^{\pi,\lambda}(s_h) - r_h(s_h, \pi_h(s_h))\big)^2 = -\big(V_h^{\pi,\lambda}(s_h)\big)^2 + 2V_h^{\pi,\lambda}(s_h)r_h(s_h, \pi_h(s_h)) - r_h(s_h, \pi_h(s_h))^2$$
$$\le -\big(V_h^{\pi,\lambda}(s_h)\big)^2 + 2\min\{1+\lambda, H\},$$

where in the second inequality we use the fact that the reward is bound in $[0, 1]$ and Proposition B.1. Then we can upper bound the variance as

$$\mathbb{V}_{s\sim P_h^0(\cdot|s_h,a_h)}\big[V_{h+1}^{\pi,\lambda}(s)\big] \le \mathbb{E}_{s\sim P_h^0(\cdot|s_h,a_h)}\big[\big(V_{h+1}^{\pi,\lambda}(s)\big)^2\big] - \big(V_h^{\pi,\lambda}(s_h)\big)^2 + 2\min\{1+\lambda, H\}.$$

Taking expectation over $s_h, a_h$, we have

$$\mathbb{E}_{(s_h,a_h)\sim(P^0,\pi)}\big[\mathbb{V}_{s\sim P_h^0(\cdot|s_h,a_h)}\big[V_{h+1}^{\pi,\lambda}(s)\big]\big]$$

$$\leq \mathbb{E}_{s_h\sim(P^0,\pi)}\big[\mathbb{V}_{s\sim P_h^0(\cdot|s_h,\pi_h(s_h))}\big[V_{h+1}^{\pi,\lambda}(s)\big]\big]$$

$$\leq \mathbb{E}_{s_h\sim(P^0,\pi)}\big[\mathbb{E}_{s\sim P_h^0(\cdot|s_h,a_h)}\big[\big(V_{h+1}^{\pi,\lambda}(s)\big)^2\big] - \big(V_h^{\pi,\lambda}(s_h)\big)^2 + 2\min\{1+\lambda,H\}\big]$$

$$\leq \mathbb{E}_{s_{h+1}\sim(P^0,\pi)}\big[\big(V_{h+1}^{\pi,\lambda}(s)\big)^2\big] - \mathbb{E}_{s_h\sim(P^0,\pi)}\big[\big(V_h^{\pi,\lambda}(s_h)\big)^2\big] + 2\min\{1+\lambda,H\}.$$

Then we taking a summation over $h \in [H]$ and by telescoping we have

$$\mathbb{E}_{(s_h,a_h)\sim(P^0,\pi),\forall h\in[H]}\bigg[\sum_{h=1}^{H}\mathbb{V}_{s\sim P_h^0(\cdot|s_h,a_h)}\big[V_{h+1}^{\pi,\lambda}(s)\big]\bigg] \leq 2H\min\{1+\lambda,H\}.$$

Then substitute $\pi$ by $\pi^1, \cdots, \pi^K$, we have

$$\text{Term (ii)} = \sum_{k=1}^{K}\mathbb{E}_{(s_h,a_h)\sim(P^0,\pi^k),\forall h\in[H]}\bigg[\sum_{h=1}^{H}\mathbb{V}_{s\sim P_h^0(\cdot|s_h^k,a_h^k)}\big[V_{h+1}^{\pi^k,\lambda}(s)\big]\bigg] \leq 2H\min\{1+\lambda,H\}K. \tag{B.30}$$

Substitute (B.27) and (B.30) back into (B.26), we have

$$\sum_{k=1}^{K}\sum_{h=1}^{H}\mathbb{V}_h V_{h+1}^{\pi^k,\lambda}(s_h^k,a_h^k) \leq c\cdot H\cdot\min\{1+\lambda,H\}^2\cdot\sqrt{K\iota} + 2\cdot H\cdot\min\{1+\lambda,H\}\cdot K$$

$$\leq c'\cdot H\cdot\min\{1+\lambda,H\}\cdot K + c''\cdot H\cdot\min\{1+\lambda,H\}^3\cdot\iota,$$

where in the last inequality we use $\sqrt{ab} \leq a + b$ for any $a, b > 0$. This finishes the proof. $\qquad\square$

The last lemma shows that the empirical variance of $V^{k,\lambda}$, $W_t(s,a,h)$, is also upper bounded by the variance $\mathbb{V}_h V_{h+1}^{\pi^k,\lambda}(s,a)$ plus some small terms.

**Lemma B.9.** *There exists absolute constants $c_1, c_2 > 0$ such that, letting $(s,a) = (s_h^k, a_h^k)$ and $t = n_h^k = N_h^k(s,a)$, we have that for all $(k,h) \in [K] \times [H]$, with probability at least $1 - 4p$,*

$$W_t(s,a,h) \leq \mathbb{V}_h V_{h+1}^{\pi^k,\lambda}(s,a) + 2\min\{1+\lambda,H\}\big(\xi_{h+1}^k + \delta_{h+1}^k\big)$$

$$+ c\Big(\frac{SA\min\{1+\lambda,H\}^2\sqrt{H^3\iota}}{t} + \min\{1+\lambda,H\}^2\sqrt{\frac{H^3\iota}{t}}\Big),$$

*where $\xi_{h+1}^k := \big[\big(\mathbb{P}_h - \hat{\mathbb{P}}_h^k\big)\big(V_{h+1}^{\star,\lambda} - V_{h+1}^{k,\lambda}\big)\big](s_h^k, a_h^k)$ and $\delta_{h+1}^k := \big(V_{h+1}^{\star,\lambda} - V_{h+1}^{k,\lambda}\big)(s_{h+1}^k).$*

*Proof.* We first assume that Lemma B.7 holds with probability $1 - 2p$, and Lemma B.5 with probability $1 - p$. As a consequence, with probability at least $1 - p$, Lemma B.6 holds for all $k \in [K]$. By the triangle inequality, we have

$$W_t(s,a,h) - \mathbb{V}_h V_{h+1}^{\pi^k,\lambda}(s,a) \leq \big|\mathbb{V}_h V_{h+1}^{\star,\lambda}(s,a) - W_t(s,a,h)\big| + \big|\mathbb{V}_h V_{h+1}^{\star,\lambda}(s,a) - \mathbb{V}_h V_{h+1}^{\pi^k,\lambda}(s,a)\big|,$$

where the first term on the RHS is upper bounded by Lemma B.6. For the second term,

$$\big|\mathbb{V}_h V_{h+1}^{\star,\lambda}(s,a) - \mathbb{V}_h V_{h+1}^{\pi^k,\lambda}(s,a)\big|$$

$$\leq 2\min\{1+\lambda,H\}\big[\mathbb{P}_h\big(V_{h+1}^{\star,\lambda} - V_{h+1}^{\pi^k,\lambda}\big)\big](s_h^k, a_h^k) = 2\min\{1+\lambda,H\}\big(\xi_{h+1}^k + \delta_{h+1}^k\big).$$

$\qquad\square$

*Proof of Theorem 4.3.* We assume that Lemma B.8 holds with probability at least $1 - p$ and Lemma B.6 holds with probability at least $1 - 4p$. By the same argument in the proof of Theorem 4.2 (in particular (B.11)), we have

$$\sum_{k=1}^{K}\delta_h^k \leq SA\min\{1+\lambda,H\} + \Big(1 + \frac{1}{H}\Big)\sum_{k=1}^{K}\delta_{h+1}^k + \sum_{k=1}^{K}\big(\beta_{n_h^k}(s_h^k,a_h^k,h) + \xi_{h+1}^k\big), \tag{B.31}$$

where $\xi_{h+1}^k := \big[(\mathbb{P}_h - \hat{\mathbb{P}}_h^k)(V_{h+1}^{\star,\lambda} - V_{h+1}^{k,\lambda})\big](s_h^k, a_h^k)$ and $\delta_{h+1}^k := (V_{h+1}^{\star,\lambda} - V_{h+1}^{k,\lambda})(s_{h+1}^k)$. As a result, for any $h \in [H]$, by recursing the above formula for $h, h+1, \cdots, K$, we have

$$\sum_{k=1}^K \delta_h^k \leq SA \min\{1+\lambda, H\}H + \sum_{h'=h}^H \sum_{k=1}^K \big(\beta_{n_{h'}^k}(s_{h'}^k, a_{h'}^k, h') + \xi_{h+1}^k\big), \tag{B.32}$$

By the Azuma-Hoeffding inequality, with probability at least $1 - p$, we have:

$$\forall h \in [H] : \left| \sum_{h'=h}^H \sum_{k=1}^K \xi_{h'+1}^k \right| = \left| \sum_{h'=h}^H \sum_{k=1}^K \big[(\mathbb{P}_{h'} - \hat{\mathbb{P}}_{h'}^k)(V_{h'+1}^\star - V_{h'+1}^k)\big](s_{h'}^k, a_{h'}^k) \right|$$

$$\leq O(\min\{1+\lambda, H\}\sqrt{HK\iota}). \tag{B.33}$$

Also recall that by definition $\beta_t(s, a, h) \leq c\min\{1+\lambda, H\}\sqrt{H\iota/t}$, so according to (B.12),

$$\sum_{k=1}^K \beta_{n_h^k}(s_h^k, a_h^k, h) \leq O(\min\{1+\lambda, H\}\sqrt{SAHK\iota}).$$

Substituting these back into (B.31), we have

$$\sum_{k=1}^K \delta_h^k \leq O\big(SA \min\{1+\lambda, H\}H + \min\{1+\lambda, H\}\sqrt{SAH^3 K\iota}\big). \tag{B.34}$$

Note that when $HK \geq \min\{1+\lambda, H\}\sqrt{SAH^3 K\iota}$, we have $\min\{1+\lambda, H\}\sqrt{SAH^3 K\iota} \geq SA\min\{1+\lambda, H\}H$; when $HK \leq \min\{1+\lambda, H\}\sqrt{SAH^3 K\iota}$, we have $\sum_{k=1}^K \delta_h^k \leq HK \leq \min\{1+\lambda, H\}\sqrt{SAH^3 K\iota}$. Therefore, in conclusion we have

$$\sum_{k=1}^K \delta_h^k \leq O\big(\min\{1+\lambda, H\}\sqrt{SAH^3 K\iota}\big).$$

By our choice of $\beta_t$, we have

$$\sum_{k=1}^K \sum_{h=1}^H \beta_{n_h^k}(s_h^k, a_h^k, h)$$

$$\leq O(1) \cdot \sum_{k=1}^K \sum_{h=1}^H \left[ \sqrt{\frac{H}{n_h^k} \cdot \big(W_{n_h^k}(s_h^k, a_h^k, h) + \min\{1+\lambda, H\}\big)\iota} + \min\{1+\lambda, H\}^3 \frac{\sqrt{SAH^4}}{n_h^k}\iota \right]. \tag{B.35}$$

The summation of the second term in (B.35) is upper bounded by

$$\sum_{k=1}^K \sum_{h=1}^H \min\{1+\lambda, H\}^3 \frac{\sqrt{SAH^4}}{n_h^k}\iota \leq \min\{1+\lambda, H\}^3 \sqrt{S^3 A^3 H^6 \iota^4}, \tag{B.36}$$

where we use the fact that $1 + \frac{1}{2} + \frac{1}{3} + \cdots \leq \iota$ and $\sum_{k=1}^K \frac{1}{n_h^k} \leq SA\iota$. The summation of the first term in (B.35) can be upper bounded by

$$\sum_{k=1}^K \sum_{h=1}^H \sqrt{\frac{H}{n_h^k} \cdot \big(W_{n_h^k}(s_h^k, a_h^k, h) + \min\{1+\lambda, H\}\big)\iota} \tag{B.37}$$

$$\leq \sqrt{\left( \sum_{k=1}^K \sum_{h=1}^H \big(W_{n_h^k}(s_h^k, a_h^k, h) + \min\{1+\lambda, H\}\big) \right) \cdot \left( \sum_{k=1}^K \sum_{h=1}^H \frac{H}{n_h^k}\iota \right)}$$

$$\leq \sqrt{\sum_{k=1}^K \sum_{h=1}^H W_{n_h^k}(s_h^k, a_h^k, h) \cdot \sqrt{H^2 SA\iota^2}} + \sqrt{\min\{1+\lambda, H\}H^3 SAK\iota}. \tag{B.38}$$

Then it remains to bound the first term on the RHS of the above inequality. By Lemma B.6, we have

$$\sum_{k=1}^{K}\sum_{h=1}^{H} W_{n_h^k}(s_h^k, a_h^k, h) \le \sum_{k=1}^{K}\sum_{h=1}^{H}\left[\mathbb{V}_h V_{h+1}^{\pi^k,\lambda}(s_h^k, a_h^k) + 2\min\{1+\lambda, H\}\big(\xi_{h+1}^k + \delta_{h+1}^k\big)\right.$$
$$\left. + O\Big(\frac{SA\min\{1+\lambda, H\}^2\sqrt{H^3\iota}}{n_h^k} + \min\{1+\lambda, H\}^2\sqrt{\frac{H^3SA\iota}{n_h^k}}\Big)\right].$$

Using the fact that $\sum_{k=1}^{K}\frac{1}{n_h^k} \le SA\iota$ and $\sum_{k=1}^{K}\frac{1}{\sqrt{n_h^k}} \le \sqrt{SAK}$, we have

$$\sum_{k=1}^{K}\sum_{h=1}^{H} W_{n_h^k}(s_h^k, a_h^k, h) \le \sum_{k=1}^{K}\sum_{h=1}^{H}\left[\mathbb{V}_h V_{h+1}^{\pi^k,\lambda}(s_h^k, a_h^k) + 2\min\{1+\lambda, H\}\big(\xi_{h+1}^k + \delta_{h+1}^k\big)\right]$$
$$+ O\big(S^2A^2\min\{1+\lambda, H\}^2\sqrt{H^5\iota^3} + SA\min\{1+\lambda, H\}^2\sqrt{H^5K\iota}\big)$$

Next, by Lemma B.8, (B.33) and (B.34), we further have

$$\sum_{k=1}^{K}\sum_{h=1}^{H} W_{n_h^k}(s_h^k, a_h^k, h)$$
$$\le 2\min\{1+\lambda, H\}\sum_{k=1}^{K}\sum_{h=1}^{H}\big(\xi_{h+1}^k + \delta_{h+1}^k\big) + O\big(\min\{1+\lambda, H\}HK + H\min\{1+\lambda, H\}^3\iota$$
$$+ S^2A^2\min\{1+\lambda, H\}^2\sqrt{H^5\iota^3} + SA\min\{1+\lambda, H\}^2\sqrt{H^5K\iota}\big)$$
$$\le O\big(\min\{1+\lambda, H\}\sqrt{H^7SAK\iota} + \min\{1+\lambda, H\}HK + H\min\{1+\lambda, H\}^3\iota$$
$$+ S^2A^2\min\{1+\lambda, H\}^2\sqrt{H^5\iota^3} + SA\min\{1+\lambda, H\}^2\sqrt{H^5K\iota}\big)$$
$$\le O\big(\min\{1+\lambda, H\}HK + S^2A^2\min\{1+\lambda, H\}^2\sqrt{H^5\iota^3} + \min\{1+\lambda, H\}H^6S^2A^2\iota\big). \tag{B.39}$$

Substituting (B.36), (B.37) and (B.39) back into (B.35), we have

$$\sum_{k=1}^{K}\sum_{h=1}^{H}\beta_{n_h^k}(s_h^k, a_h^k, h) \le O\big(\sqrt{\min\{1+\lambda, H\}H^3KSA\iota^2} + \sqrt{\min\{1+\lambda, H\}^3H^6S^3A^3\iota^4}\big). \tag{B.40}$$

Substituting (B.40) into (B.32), then with probability at least $1 - 6p$, for any $h \in [H]$, we have

$$\sum_{k=1}^{K}\delta_h^k \le O\big(\sqrt{\min\{1+\lambda, H\}H^3KSA\iota^2} + \sqrt{\min\{1+\lambda, H\}^3H^6S^3A^3\iota^4}\big).$$

Recall (B.9), we have

$$\mathrm{Regret}(K) \le \sum_{k=1}^{K}\big(V_1^{k,\lambda} - V_1^{\pi_k,\lambda}(s_1^k)\big) = \sum_{k=1}^{K}\delta_1^k,$$

then we finish the proof by rescaling $p$ to $p/6$. $\qquad\square$

### B.3. Proof of Theorem 4.4

*Proof.* The proof of Theorem 4.4 mostly follows the proof of Theorem 5.12 in (He et al., 2025). According to (E.8) therein, we have for any algorithm $\xi$

$$2\max\big\{\mathbb{E}\big[\mathrm{Regret}_{\mathcal{M}_1}(\xi, K)\big], \mathbb{E}\big[\mathrm{Regret}_{\mathcal{M}_2}(\xi, K)\big]\big\} \ge \Omega\Big(\frac{e^{-\frac{1}{2}}\sqrt{\widetilde{p}}}{2}\sqrt{C_{vr}(|\mathcal{A}| - 1)K}\Big).$$

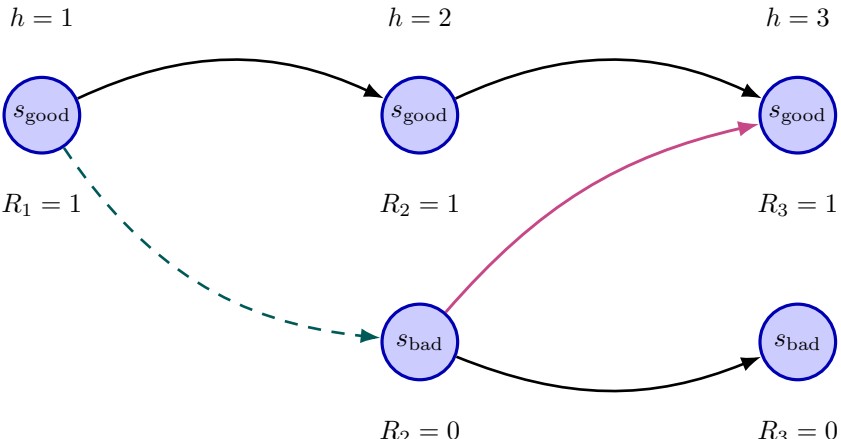

*Figure 6.* Illustration of the hard instance building on Lu et al. (2024, Example 3.1) adapted to the RRMDP setting. The solid lines represent possible transitions of the nominal transition kernel. The dashed line represents the possible transitions induced by the worst case transition kernel of the regularize objective function. By a proper choice of the regularizer $\lambda$, there is no dynamics shift in the worst case at $s_{\text{good}}$ in stage $h = 2$. The red solid line represents the transition where the two RRMDP instances differ in that different actions leads to higher transition probability from $s_{\text{bad}}$ to $s_{\text{good}}$. In the nominal environment, starting from $s_1 = s_{\text{good}}$, the nominal transition kernel keeps the agent at $s_{\text{good}}$ and no information at $s_{\text{bad}}$ is revealed.

By the argument in Appendix F.1.2, we have $\tilde{p} = 1$ and $C_{vr} = \tilde{p}/p$. To proceed, we set $K \geq 2^{|\mathcal{A}|+1}|\mathcal{A}|$ and $p = 2^{-(|\mathcal{A}|+1)}$, then we have

$$\inf_{\xi} \sup_{\theta \in \{0,1\}} \mathbb{E}\big[\text{Regret}_{\mathcal{M}_\theta}(\xi, K)\big] \geq \Omega(2^{|\mathcal{A}|}\sqrt{K}).$$

This finishes the proof. $\qquad\qquad\square$

### B.4. Proof of Theorem 4.5

In this section, we prove the hardness result in Theorem 4.5. In particular, we leverage the hard instances constructed in Example 3.1 of Lu et al. (2024) as illustrated in Figure 6, and establish the lower bound by carefully analyzing the regularized value function and the worst case transition kernel of the regularized optimization problem using Proposition 3.6.

**Hard Instances.** Consider two MDPs $\mathcal{M}_0$ and $\mathcal{M}_1$ with the state space $\mathcal{S} = \{s_{\text{good}}, s_{\text{bad}}\}$, the action space $\mathcal{A} = \{0, 1\}$, horizon length $H = 3$. The reward function is defined as

$$r_h(s, a) = \begin{cases} 1, & s = s_{\text{good}} \\ 0, & s = s_{\text{bad}} \end{cases}, \quad \forall(a, h) \in \mathcal{A} \times [H].$$

The good state $s_{\text{good}}$ always leads to a reward of 1 and the bad state $s_{\text{bad}}$ always leads to 0 reward. The only difference between $\mathcal{M}_0$ and $\mathcal{M}_1$ lies in their transition kernels, which are defined as follows

$$P_h^{0,\mathcal{M}_\theta}(s_{\text{good}}|s_{\text{good}}, a) = 1, \quad \forall(a, h, \theta) \in \mathcal{A} \times \{1, 2\} \times \{0, 1\},$$

$$P_2^{0,\mathcal{M}_\theta}(s_{\text{good}}|s_{\text{bad}}, a) = \begin{cases} p, & a = \theta \\ q, & a = 1 - \theta \end{cases}, \quad \forall \theta \in \{0, 1\},$$

where $0 < p < q < 1$ are two constants. So the good state $s_{\text{good}}$ is an absorbing state that only transits to itself. For the bad state $s_{\text{bad}}$, there is a chance to transit to the good state $s_{\text{good}}$ with the probability depending on the action taken. Thus, the optimal policy for $\mathcal{M}_\theta$, at the bad state $s_{\text{bad}}$, would take the corresponding action $\theta$ that leads to a larger probability of transitioning to the good state $s_{\text{good}}$. We set the initial state $s_1$ always be the good state $s_{\text{good}}$. Thus, in the nominal environment, the only trajectory that can be collected is $\{s_{\text{good}}, a_1, r_1, s_{\text{good}}, a_2, r_2, s_{\text{good}}, a_3, r_3\}$. In other words, the agent does not have access to any information over the bad state $s_{\text{bad}}$ by interacting with the nominal environment, thus it is not informed how to make decisions at bad state $s_{\text{bad}}$.

Based on the MDPs $\mathcal{M}_0$ and $\mathcal{M}_1$, we construct robust regularized MDPs with TV-distance defined regularization by setting $\lambda \in (1, 2)$. We are interested in lower bounding the minimax regret defined as follows

$$\inf_{\mathcal{ALG}} \sup_{\theta \in \{0,1\}} \mathbb{E}\big[\text{Regret}(K; \mathcal{M}_\theta, \mathcal{ALG}) \geq \Omega(K)\big],$$

where we recall the definition of regret

$$\text{Regret}(K; \mathcal{M}_\theta, \mathcal{ALG}) = \sum_{k=1}^{K} \big[V_1^{\star,\lambda}(s_1^k; \mathcal{M}_\theta) - V_1^{\pi^k,\lambda}(s_1^k; \mathcal{M}_\theta)\big],$$

and for any policy $\pi$ the robust regularized value function is defined as

$$V_h^{\pi,\lambda}(s; \mathcal{M}_\theta) = \inf_P \mathbb{E}^{\pi,P}\bigg[\sum_{t=h}^{H} r_t(s_t, a_t) + \lambda \cdot D(P_t(\cdot|s_t, a_t)||P_t^{0,\mathcal{M}_\theta}(\cdot|s_t, a_t))\Big|s_h = s\bigg].$$

Next, we present the performance difference lemma for robust regularized value functions that would be useful in the proof of Theorem 4.5.

**Lemma B.10.** *For the robust regularized MDPs with TV-distance defined regularizations, given any policy $\pi$, we have the following inequality*

$$V_1^{\pi^\star,\lambda}(s) - V_1^{\pi,\lambda}(s) \geq \mathbb{E}_{P^{\pi^\star,\lambda,\dagger}}^{\pi^\star}\bigg[\sum_{h=1}^{H}\sum_{a\in\mathcal{A}} \big(\pi_h^\star(a|s_h) - \pi_h(a|s_h)\big) \cdot Q_h^{\pi,\lambda}(s_h, a)\Big|s_1 = s\bigg], \tag{B.41}$$

*where the expectation is taken with respect to trajectories induced by the policy $\pi^\star$ and transition kernel $P^{\pi^\star,\lambda,\dagger}$. Here $P^{\pi^\star,\lambda,\dagger} = \big\{P_h^{\pi^\star,\lambda,\dagger}\big\}_{h=1}^{H}$ is the worst case transition kernel defined as*

$$P_h^{\pi^\star,\lambda,\dagger}(\cdot|s,a) = \arg\inf_{P(\cdot)\in\Delta(\mathcal{S})} \mathbb{E}_{s\sim P(\cdot)}\big[V_{h+1}^{\star,\lambda}(s) + \lambda D_{TV}\big(P(\cdot)||P_h^0(\cdot|s,a)\big)\big], \quad \forall(s,a,h)\in\mathcal{S}\times\mathcal{A}\times[H]. \tag{B.42}$$

*Proof.* For any step $h \in [H]$, by robust Bellman equation (3.3), we have

$$
\begin{aligned}
Q_h^{\pi^\star,\lambda}(s,a) - Q_h^{\pi,\lambda}(s,a) &= \inf_{P(\cdot)\in\Delta(\mathcal{S})} \mathbb{E}_{s\sim P(\cdot)}\big[V_{h+1}^{\pi^\star,\lambda}(s) + \lambda D_{TV}\big(P(\cdot)||P_h^0(\cdot|s,a)\big)\big] \\
&\quad - \inf_{P(\cdot)\in\Delta(\mathcal{S})} \mathbb{E}_{s\sim P(\cdot)}\big[V_{h+1}^{\pi,\lambda}(s) + \lambda D_{TV}\big(P(\cdot)||P_h^0(\cdot|s,a)\big)\big] \\
&\geq \mathbb{E}_{s\sim P_h^{\pi^\star,\lambda,\dagger}(\cdot|s,a)}\big[V_{h+1}^{\pi^\star,\lambda}(s)\big] - \mathbb{E}_{s\sim P_h^{\pi^\star,\lambda,\dagger}(\cdot|s,a)}\big[V_{h+1}^{\pi,\lambda}(s)\big] \\
&= \mathbb{E}_{s\sim P_h^{\pi^\star,\lambda,\dagger}(\cdot|s,a)}\big[V_{h+1}^{\pi^\star,\lambda}(s) - V_{h+1}^{\pi,\lambda}(s)\big], \tag{B.43}
\end{aligned}
$$

where the first inequality holds due by the definition (B.42).

By robust Bellman equation (3.4), we have

$$
\begin{aligned}
V_h^{\pi^\star,\lambda}(s) - V_h^{\pi,\lambda}(s) &= \mathbb{E}_{a\sim\pi_h^\star(\cdot|s)}\big[Q_h^{\pi^\star,\lambda}(s,a)\big] - \mathbb{E}_{a\sim\pi_h(\cdot|s)}\big[Q_h^{\pi,\lambda}(s,a)\big] \\
&= \mathbb{E}_{a\sim\pi_h^\star(\cdot|s)}\big[Q_h^{\pi,\lambda}(s,a)\big] - \mathbb{E}_{a\sim\pi_h(\cdot|s)}\big[Q_h^{\pi,\lambda}(s,a)\big] \\
&\quad + \mathbb{E}_{a\sim\pi_h^\star(\cdot|s)}\big[Q_h^{\pi^\star,\lambda}(s,a)\big] - \mathbb{E}_{a\sim\pi_h^\star(\cdot|s)}\big[Q_h^{\pi,\lambda}(s,a)\big] \\
&\geq \sum_{a\in\mathcal{A}} \big(\pi_h^\star(a|s) - \pi_h(a|s)\big)Q_h^{\pi,\lambda}(s,a) + \mathbb{E}_{a\sim\pi_h^\star(\cdot|s),s\sim P_h^{\pi^\star,\lambda,\dagger}(\cdot|s,a)}\big[V_{h+1}^{\pi^\star,\lambda}(s) - V_{h+1}^{\pi,\lambda}(s)\big]
\end{aligned}
$$

Recursively applying (B.43) over $h \in [H]$, we can conclude that

$$V_1^{\pi^\star,\lambda}(s) - V_1^{\pi,\lambda}(s) \geq \mathbb{E}_{P^{\pi^\star,\lambda,\dagger}}^{\pi^\star}\bigg[\sum_{h=1}^{H}\sum_{a\in\mathcal{A}} \big(\pi_h^\star(a|s_h) - \pi_h(a|s_h)\big) \cdot Q_h^{\pi,\lambda}(s_h, a)\Big|s_1 = s\bigg].$$

This completes the proof. $\square$

Now we are ready to prove Theorem 4.5.

*Proof of Theorem 4.5.* Fix a policy $\pi$, we write out the ground truth value functions at each step as follows. At the last step $h = 3$, the value function is just the immediate reward achieved, thus we have for any $\theta \in \{0, 1\}$,

$$Q_3^{\pi,\lambda}(s_{\text{good}}, a; \mathcal{M}_\theta) = V_3^{\pi,\lambda}(s_{\text{good}}; \mathcal{M}_\theta) = 1, \tag{B.44}$$
$$Q_3^{\pi,\lambda}(s_{\text{bad}}, a; \mathcal{M}_\theta) = V_3^{\pi,\lambda}(s_{\text{bad}}; \mathcal{M}_\theta) = 0.$$

At step $h = 2$, by the robust Bellman equation (3.3), Proposition 3.4 and our choice of regularizer $\lambda \in (1, 2)$, we have

$$
\begin{aligned}
Q_2^{\pi,\lambda}(s_{\text{good}}, a; \mathcal{M}_\theta) &= 1 + \inf_{P(\cdot) \in \Delta(\mathcal{S})} \mathbb{E}_{s \sim P(\cdot)} \big[ V_3^{\pi,\lambda}(s; \mathcal{M}_\theta) + \lambda D_{\text{TV}}(P(\cdot) \| P_2^{0,\mathcal{M}_\theta}(\cdot | s_{\text{good}}, a)) \big] \\
&= 1 + \mathbb{E}_{s \sim P_2^{0,\mathcal{M}_\theta}(\cdot | s_{\text{good}}, a)} \big[ V_3^{\pi,\lambda}(s; \mathcal{M}_\theta) \big]_\lambda \\
&= 1 + \mathbb{E}_{s \sim P_2^{0,\mathcal{M}_\theta}(\cdot | s_{\text{good}}, a)} \big[ V_3^{\pi,\lambda}(s; \mathcal{M}_\theta) \big] \\
&= 2.
\end{aligned}
\tag{B.45}
$$

$$
\begin{aligned}
Q_2^{\pi,\lambda}(s_{\text{bad}}, a; \mathcal{M}_\theta) &= 0 + \inf_{P(\cdot) \in \Delta(\mathcal{S})} \mathbb{E}_{s \sim P(\cdot)} \big[ V_3^{\pi,\lambda}(s; \mathcal{M}_\theta) + \lambda D_{\text{TV}}(P(\cdot) \| P_2^{0,\mathcal{M}_\theta}(\cdot | s_{\text{bad}}, a)) \big] \\
&= 0 + \mathbb{E}_{s \sim P_2^{0,\mathcal{M}_\theta}(\cdot | s_{\text{bad}}, a)} \big[ V_3^{\pi,\lambda}(s; \mathcal{M}_\theta) \big]_\lambda \\
&= 0 + \mathbb{E}_{s \sim P_2^{0,\mathcal{M}_\theta}(\cdot | s_{\text{bad}}, a)} \big[ V_3^{\pi,\lambda}(s; \mathcal{M}_\theta) \big] \\
&= \begin{cases} p, & a = \theta \\ q, & a = 1 - \theta \end{cases}
\end{aligned}
\tag{B.46}
$$

By Proposition 3.6, we know there is no distribution shift in the second step, i.e., the worst case transition kernel is the nominal kernel. By the robust bellman equation (3.4), we have

$$2 = V_2^{\pi,\lambda}(s_{\text{good}}; \mathcal{M}_\theta) > V_2^{\pi,\lambda}(s_{\text{bad}}; \mathcal{M}_\theta) = \pi_2(\theta | s_{\text{bad}}) \cdot p + \pi_2(1 - \theta | s_{\text{bad}}) \cdot q.$$

For the first step $h = 1$, for any $a \in \mathcal{A}$

$$
\begin{aligned}
Q_1^{\pi,\lambda}(s_{\text{good}}, a; \mathcal{M}_\theta) &= 1 + \inf_{P(\cdot) \in \Delta(\mathcal{S})} \mathbb{E}_{s \sim P(\cdot)} \big[ V_2^{\pi,\lambda}(s; \mathcal{M}_\theta) + \lambda D_{\text{TV}}(P(\cdot) \| P_1^{0,\mathcal{M}_\theta}(\cdot | s_{\text{good}}, a)) \big] \\
&= 1 + \mathbb{E}_{s \sim P_1^{0,\mathcal{M}_\theta}(\cdot | s_{\text{good}}, a)} \big[ V_2^{\pi,\lambda}(s; \mathcal{M}_\theta) \big]_\lambda \\
&= 1 + \lambda.
\end{aligned}
\tag{B.47}
$$

By Proposition 3.6, we know the worst case transition at the first step is

$$P_1^{\pi,\lambda,\dagger}(\cdot | s_{\text{good}}, a) = \mathbb{1}\{\cdot = s_{\text{bad}}\}, \quad \forall a \in \mathcal{A}, \tag{B.48}$$

say, in the worst case the initial state will transit to the bad state $s_{\text{bad}}$ with probability 1. Lastly, by the robust bellman equation (3.4), we have $V_1^{\pi,\lambda}(s_{\text{good}}; \mathcal{M}_\theta) = Q_1^{\pi,\lambda}(s_{\text{good}}, a; \mathcal{M}_\theta), \forall a \in \mathcal{A}$.

To proceed, we substitute the ground truth value functions into the performance difference lemma to get a lower bound on the regret. In particular, for step $h = 3$, by (B.44) we have

$$\sum_{a \in \mathcal{A}} \big( \pi_3^{\star,\mathcal{M}_\theta}(a | s_3) - \pi_3(a | s_3) \big) \cdot Q_3^{\pi,\lambda}(s_3, a; \mathcal{M}_\theta) = 0, \quad \forall s \in \{s_{\text{good}}, s_{\text{bad}}\}. \tag{B.49}$$

For step $h = 1$, by (B.47) we have

$$\sum_{a \in \mathcal{A}} \big( \pi_1^{\star,\mathcal{M}_\theta}(a | s_1) - \pi_1(a | s_1) \big) \cdot Q_1^{\pi,\lambda}(s_1, a; \mathcal{M}_\theta) = 0, \quad s_1 = s_{\text{good}}. \tag{B.50}$$

For step $h = 2$, by (B.45), we have

$$\sum_{a \in \mathcal{A}} \left( \pi_2^{\star, \mathcal{M}_\theta}(a|s_{\text{good}}) - \pi_2(a|s_{\text{good}}) \right) \cdot Q_2^{\pi, \lambda}(s_{\text{good}}, a; \mathcal{M}_\theta) = 0. \tag{B.51}$$

By (B.46), and the fact that the optimal policy of $\mathcal{M}_\theta$ at step $h = 2$ and $s_{\text{bad}}$ is $\pi_2^{\star, \mathcal{M}_\theta}(\theta|s_{\text{bad}}) = 1$ (according to (B.46)), we have

$$\begin{aligned}
&\sum_{a \in \mathcal{A}} \left( \pi_2^{\star, \mathcal{M}_\theta}(a|s_{\text{bad}}) - \pi_2(a|s_{\text{bad}}) \right) \cdot Q_2^{\pi, \lambda}(s_{\text{bad}}, a; \mathcal{M}_\theta) \\
&= p - \left( \pi_2(\theta|s_{\text{bad}}) \cdot p + \pi_2(1 - \theta|s_{\text{bad}}) \cdot q \right) \\
&= \frac{p - q}{2} \left( \left| \pi_2^{\star, \mathcal{M}_\theta}(\theta|s_{\text{bad}}) - \pi_2(\theta|s_{\text{bad}}) \right| + \left| \pi_2^{\star, \mathcal{M}_\theta}(1 - \theta|s_{\text{bad}}) - \pi_2(1 - \theta|s_{\text{bad}}) \right| \right) \\
&= (p - q) D_{\text{TV}} \left( \pi_2^{\star, \mathcal{M}_\theta}(\cdot|s_{\text{bad}}) || \pi_2(\cdot|s_{\text{bad}}) \right).
\end{aligned} \tag{B.52}$$

Combining (B.49), (B.50), (B.51) and (B.52) with (B.41), we have

$$\begin{aligned}
&V_1^{\pi^\star, \lambda}(s_{\text{good}}; \mathcal{M}_\theta) - V_1^{\pi, \lambda}(s_{\text{good}}; \mathcal{M}_\theta) \\
&\geq \mathbb{E}_{a_1 \sim \pi_1^{\star, \mathcal{M}_\theta}(\cdot|s_{\text{good}}), s_2 \sim P_1^{\pi^\star, \lambda, \dagger}(\cdot|s_{\text{good}}, a)} \left[ \sum_{a \in \mathcal{A}} \left( \pi_2^{\star, \mathcal{M}_\theta}(a|s_2) - \pi_2(a|s_2) \right) \cdot Q_2^{\pi, \lambda}(s_2, a; \mathcal{M}_\theta) \right] \\
&= P_1^{\pi^\star, \lambda, \dagger}(s_{\text{bad}}|s_{\text{good}}, 0) \cdot (p - q) \cdot D_{\text{TV}} \left( \pi_2^{\star, \mathcal{M}_\theta}(\cdot|s_{\text{bad}}) || \pi_2(\cdot|s_{\text{bad}}) \right) \\
&= (p - q) \cdot D_{\text{TV}} \left( \pi_2^{\star, \mathcal{M}_\theta}(\cdot|s_{\text{bad}}) || \pi_2(\cdot|s_{\text{bad}}) \right),
\end{aligned}$$

where the last equation holds by the definition of the worst case transition kernel (B.48). Thus, for any algorithm executing $\pi^1, \cdots, \pi^K$, its regret is lower bounded as follows

$$\begin{aligned}
\text{Regret}(K; \mathcal{M}_\theta, \mathcal{ALG}) &= \sum_{k=1}^{K} V_1^{\pi^\star, \lambda}(s_{\text{good}}; \mathcal{M}_\theta) - V_1^{\pi^k, \lambda}(s_{\text{good}}; \mathcal{M}_\theta) \\
&\geq (p - q) \cdot \sum_{k=1}^{K} D_{\text{TV}} \left( \pi_2^{\star, \mathcal{M}_\theta}(\cdot|s_{\text{bad}}) || \pi_2^k(\cdot|s_{\text{bad}}) \right).
\end{aligned}$$

However, the agent does not have access to any information over the bad state $s_{\text{bad}}$ by interacting with the nominal environment. As a result, the estimate $\pi_2^k(\cdot|s_{\text{bad}})$ of $\pi_2^{\star, \mathcal{M}_\theta}(\cdot|s_{\text{bad}}) = \mathbb{1}\{\cdot = \theta\}$ can do no better than the random guess. Formally, consider

$$\begin{aligned}
&\sup_{\theta \in \{0,1\}} \mathbb{E}_{\mathcal{M}_\theta, \mathcal{ALG}} \left[ \text{Regret}(K; \mathcal{M}_\theta, \mathcal{ALG}) \right] \\
&\geq (p - q) \cdot \sup_{\theta \in \{0,1\}} \mathbb{E}_{\mathcal{M}_\theta, \mathcal{ALG}} \left[ \sum_{k=1}^{K} D_{\text{TV}} \left( \pi_2^{\star, \mathcal{M}_\theta}(\cdot|s_{\text{bad}}) || \pi_2^k(\cdot|s_{\text{bad}}) \right) \right] \\
&= (p - q) \cdot \sup_{\theta \in \{0,1\}} \sum_{k=1}^{K} \mathbb{E}_{\mathcal{ALG}} \left[ \pi_2^k(1 - \theta|s_{bad}) \right],
\end{aligned}$$

where in the last inequality we drop the subscription of $\mathcal{M}_\theta$ because the algorithm outputs $\pi_2^k$ are independent of $\theta$ due to

out previous discussion. To proceed, we have

$$\sup_{\theta \in \{0,1\}} \sum_{k=1}^{K} \mathbb{E}_{\mathcal{ALG}}\big[\pi_2^k(1-\theta|s_{bad})\big] \geq \frac{1}{2} \sum_{\theta \in \{0,1\}} \sum_{k=1}^{K} \mathbb{E}_{\mathcal{ALG}}\big[\pi_2^k(1-\theta|s_{bad})\big]$$

$$= \frac{1}{2} \sum_{k=1}^{K} \sum_{\theta \in \{0,1\}} \mathbb{E}_{\mathcal{ALG}}\big[\pi_2^k(1-\theta|s_{bad})\big]$$

$$= \frac{1}{2} \sum_{k=1}^{K} 1$$

$$= \frac{K}{2}.$$

Thus we can conclude that

$$\inf_{\mathcal{ALG}} \sup_{\theta \in \{0,1\}} \mathbb{E}_{\mathcal{M}_\theta, \mathcal{ALG}}\big[\text{Regret}(K; \mathcal{M}_\theta, \mathcal{ALG})\big] \geq (p-q) \cdot \frac{K}{2} = \Omega(K).$$

This completes the proof. $\qquad\square$

## C. Auxiliary Lemmas

**Lemma C.1.** *Levy et al. (2020, Section A.1.2) Let $P$ be a distribution on $\mathcal{S}$ and $l : \mathcal{S} \to \mathbb{R}$ be the loss function. For any distribution $Q$ on $\mathcal{S}$ such that $Q \ll P$, define the $\phi$-divergence as*

$$D_\phi(Q||P) = \sum_{s \in \mathcal{S}} \phi\Big(\frac{Q(s)}{P(s)}\Big) P(s).$$

*Then we have the following strong duality result:*

$$\sup_{P \ll P_0} \mathbb{E}_P\big[l(X) - \lambda D_\varphi(P, P_0)\big] = \inf_{\eta \in \mathbb{R}} \left\{ \lambda \mathbb{E}_{P_0}\Big[\varphi^\star\Big(\frac{l(X) - \eta}{\lambda}\Big)\Big] + \eta \right\},$$

*where $\phi^\star = \sup_{t \geq 0}\{st - \phi(t)\}$ is the Fenchel conjugate function of $\phi$.*

**Lemma C.2** (Lemma 4.1 of Jin et al. (2018)). *The following properties hold for $\alpha_t^i$:*

*(a) $\frac{1}{\sqrt{t}} \leq \sum_{i=1}^{t} \frac{\alpha_t^i}{\sqrt{i}} \leq \frac{2}{\sqrt{t}}$ for every $t \geq 1$.*

*(b) $\max_{i \in [t]} \alpha_t^i \leq \frac{2H}{t}$ and $\sum_{i=1}^{t}(\alpha_t^i)^2 \leq \frac{2H}{t}$ for every $t \geq 1$.*

*(c) $\sum_{t=i}^{\infty} \alpha_t^i = 1 + \frac{1}{H}$ for every $i \geq 1$.*

**Lemma C.3.** *Suppose (B.14) in Lemma B.5 holds. For any $h \in [H]$, let $\phi_h^{k,\lambda} := V_h^{k,\lambda} - V_h^{\star,\lambda}(s_h^k)$, and letting $w = (w_1, \cdots, w_k)$ be a nonnegative vector, we have:*

$$\sum_{k=1}^{K} w_k \phi_h^{k,\lambda} \leq O\big(SA\|w\|_\infty \min\{1+\lambda, H\}\sqrt{H^3\iota} + \min\{1+\lambda, H\}\sqrt{SA\|w\|_1\|w\|_\infty H^3\iota}\big),$$

*where $\phi_h^{k,\lambda} := (V_h^{k,\lambda} - V_h^{\star,\lambda})(s_h^k)$.*

*Proof of Lemma C.3.* For any fixed $(k, h) \in [K] \times [H]$, let $t = N_h^k(s_h^k, a_h^k)$, and suppose $(s_h^k, a_h^k)$ was previously taken at step $h$ of episodes $k_1, \cdots, k_t \leq k$. We then have, for some absolute constant $c$:

$$\phi_h^{k,\lambda} = \big(V_h^{k,\lambda} - V_h^{\star,\lambda}\big)(s_h^k)$$

$$\leq \big(Q_h^{k,\lambda} - Q_h^{\star,\lambda}\big)(s_h^k, a_h^k)$$

$$\leq \alpha_t^0 \min\{1+\lambda, H\} + \sum_{i=1}^{t} \alpha_t^i \big([V_{h+1}^{k_i,\lambda}]_\lambda - [V_{h+1}^{\star,\lambda}]_\lambda\big)(s_{h+1}^{k_i}) + 4c_2 \min\{1+\lambda, H\}\sqrt{\frac{H\iota}{t}},$$

where the first inequality holds from $V_h^{k,\lambda}(s_h^k) \le \max_{a' \in \mathcal{A}} Q_h^k(s_h^k, a_h^k)$ and the robust Bellman optimality equation $V_h^{\star,\lambda}(s_h^k) = \max_{a' \in \mathcal{A}} Q_h^{\star,\lambda}(s_h^k, a') \ge Q_h^{\star,\lambda}(s_h^k, a_h^k)$. The second inequality holds by (B.14). For any $(s, h) \in \mathcal{S} \times [H]$, we know

$$
\begin{aligned}
V_{h+1}^{k_i,\lambda}(s) &= \left[ Q_{h+1}^{k,\lambda}(s, a_{h+1}^k) \right]_{\min\{1+\lambda, H\}} \\
&\ge \left[ Q_{h+1}^{k,\lambda}(s, \pi_{h+1}^\star(s)) \right]_{\min\{1+\lambda, H\}} \\
&\ge \left[ Q_{h+1}^{\star,\lambda}(s, \pi_{h+1}^\star(s)) \right]_{\min\{1+\lambda, H\}} = V_{h+1}^{\star,\lambda}(s),
\end{aligned}
$$

where the last inequality holds by Lemma B.2 and the last equation holds by Proposition B.1. Then we can upper bound the differences between those clipped value functions by the differences between original value functions,

$$
\begin{aligned}
\phi_h^{k,\lambda} &\le \alpha_t^0 \min\{1+\lambda, H\} + \sum_{i=1}^t \alpha_t^i \big( V_{h+1}^{k_i,\lambda} - V_{h+1}^{\star,\lambda} \big)(s_{h+1}^{k_i}) + 4c_2 \min\{1+\lambda, H\} \sqrt{\frac{H\iota}{t}} \\
&= \alpha_t^0 \min\{1+\lambda, H\} + \sum_{i=1}^t \alpha_t^i \phi_{h+1}^{k_i,\lambda} + O\left( \min\{1+\lambda, H\} \sqrt{\frac{H\iota}{t}} \right).
\end{aligned}
$$

Next, we compute the summation $\sum_{k=1}^K w_k \phi_h^{k,\lambda}$. Denoting $n_h^k = N_h^k(s_h^k, a_h^k)$, we have

$$
\sum_{k=1}^K w_k \alpha_{n_h^k}^0 \min\{1+\lambda, H\} = \sum_{k=1}^K \min\{1+\lambda, H\} w_k \cdot \mathbb{1}[n_h^k = 0] \le \min\{1+\lambda, H\} SA \|w\|_\infty,
$$

and following the same argument as the derivation of (C.20) in (Jin et al., 2018), we have

$$
\sum_{k=1}^K w_k \min\{1+\lambda, H\} \sqrt{\frac{H\iota}{n_h^k}} \le O\big( SA\|w\|_\infty + \sqrt{SA\|w\|_1 \|w\|_\infty} \big) \min\{1+\lambda, H\} \sqrt{H\iota}.
$$

The rest of proof follows the proof of lemma C.7 in (Jin et al., 2018), and we finally have

$$
\sum_{k=1}^K w_k \phi_h^k \le O\big( SA\|w\|_\infty \min\{1+\lambda, H\} \sqrt{H^3 \iota} + \min\{1+\lambda, H\} \sqrt{SA\|w\|_1 \|w\|_\infty H^3 \iota} \big).
$$

This finishes the proof. $\qquad\square$

# D. Additional Experiment Results

For the target Inverted Pendulum experiments, we further add perturbation to actions. If action perturbation is exerted, then with a certain probability, the action output from a policy will be replaced by a random action. Experiments are shown in Figure 7a and Figure 7b. The patterns are similar to Figure 5, except that all policies show certain performance degradation due to action perturbation.

## D.1. Ablation Study

**Hyperparameter $\lambda$ should be properly chosen, overly small $\lambda$ could hurt nominal performance.** We show the results of clipped DDQN with $\lambda = 80, 70$ in Figure 8. They perform worse than the standard DDQN under testing environments with moderate pole length perturbation (0.5-0.8). When the perturbation is large they still outperform the standard DDQN.

**The emergence of the robustness behavior requires a stable baseline method.** Next, we show that the value clipping mechanism highly depends on the algorithm it is inserted to. An unstable algorithm incorporating the value clipping operator may not lead to robustness behaviors. To illustrate this, we replace the DDQN with DQN (Mnih et al., 2015), which is subject to Q-function overestimation. Experiment results are shown in Figure 9. There are two observations. First, the DQN can not reliably achieve the optimal performance even under the exact training environment (one out of ten seeds could have non-optimal performance). Second, though clipped DQN with $\lambda = 90, 100$ has slightly better performance in the nominal

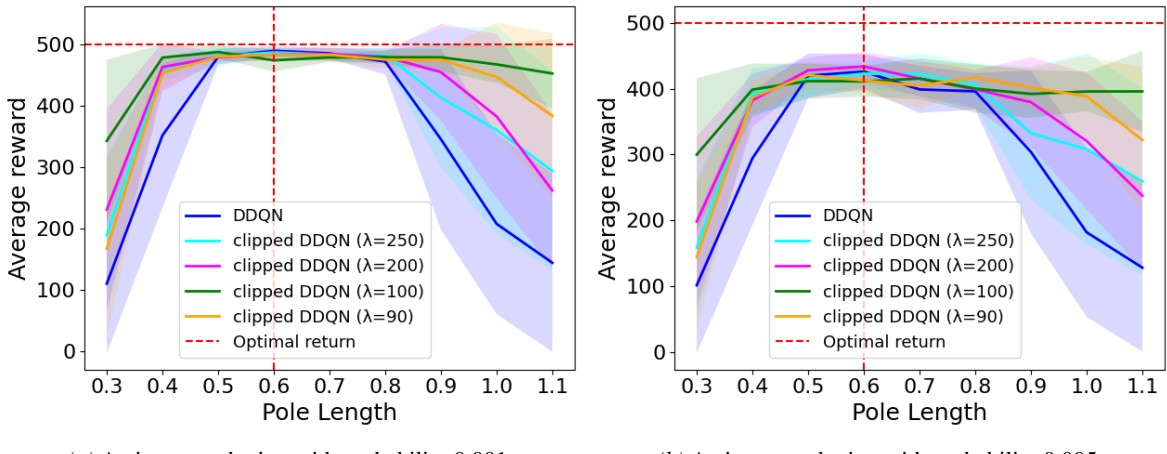

*(a)* Action perturbation with probability 0.001.

*(b)* Action perturbation with probability 0.005.

*Figure 7.* Testing results under both pole length and action perturbation. The Optimal return represents the largest cumulative reward can be achieved at the test time. The vertical red line represents the pole length in the nominal environment.

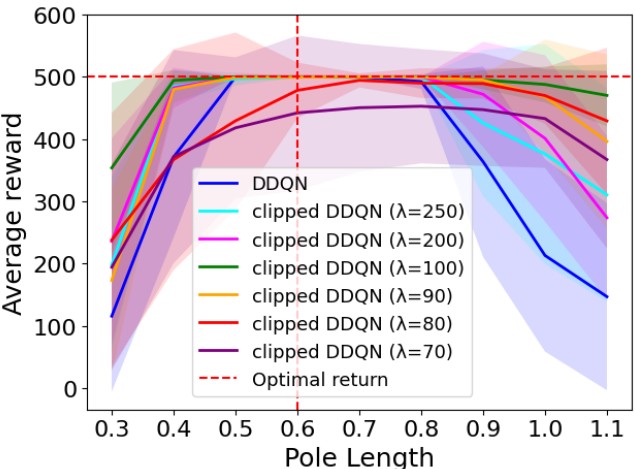

*Figure 8.* Illustration of robustness performances under different clipping threshold $\lambda$. The choice of hyperparameter $\lambda$ is important. Overly small $\lambda$, such as 70 and 80, could hurt the performance in the training environment. The Optimal return represents the largest cumulative reward can be achieved at the test time. The vertical red line represents the pole length in the nominal environment.

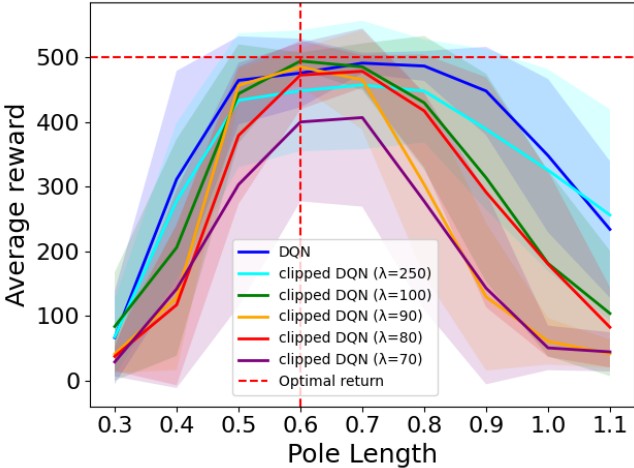

*Figure 9.* Evaluation results for DQN and its clipped versions. No action perturbation is employed in the testing environment. There is no clear evidence of robustness behavior for all clipped DQNs. The Optimal return represents the largest cumulative reward can be achieved at the test time. The vertical red line represents the pole length in the nominal environment.

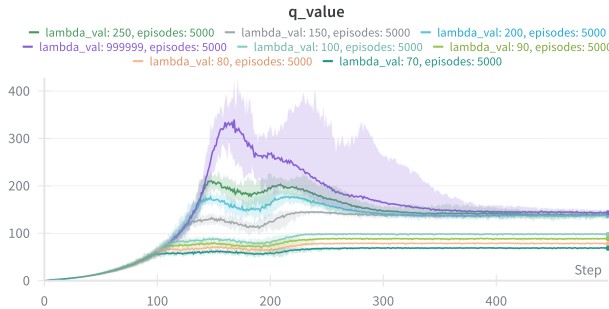

*Figure 10.* Illustration of the learning dynamics of Q-function estimation by DQN and its clipped versions. DQN overestimates the Q-function.

environment, in general they do not show robustness property against pole-length perturbation. We suspect this is due to the estimation error in DQN, to illustrate this, we show the learning dynamics of the Q-value, which is calculated as the mean value of Q-function in replay buffer, in Figure 10. The purple line is DQN without clipping, the Q-function estimation after convergence is about 140, much higher than the true optimal Q-function, which is about 100. This suggests that DQN is prone to overestimation, which can degrade performance. And the clipped version of DQN can also suffer from the biased Q-function estimation, cannot show robustness property. Instead, we show the learning dynamics of DDQN and its clipped versions in Figure 11. DDQN addresses the overestimation problem and can stably converge to the true optimal Q-function. Based on this stable learning process, clipped versions of DDQN show robustness properties as illustrated in Figure 8.

**A finer discretization of the action space with 21 discrete actions.**    Next, we discretize the action space to be of size 21 (previously 11). We train all algorithms 3000 episodes. Other factors remain the same. Experiment results are shown in Figures 12 to 14. When the action space becomes larger, DDQN requires longer training episodes to converge. However, the value clipping operator accelerates the convergence rate by shrinking the unreasonably large Q-function estimation. In general, clipped DDQN with $\lambda = 80, 90$ achieve very robust performances in terms of testing time environment perturbation.

We further train DDQN for 4,000 episodes, which is sufficient for convergence. Evaluation results are reported in Figures 12, 16 and 17. Clipped DDQN with $\lambda \in 100, 200, 250$ exhibits clear robustness properties. However, the best robust performance is achieved by clipped DDQN with smaller clipping thresholds, $\lambda \in 80, 90$, when trained for 3,000 episodes, as shown in Figure 12. This observation suggests that fully exploiting the value-clipping mechanism in modern deep RL algorithms requires more careful design, motivating future work on principled strategies for integrating and tuning value clipping.

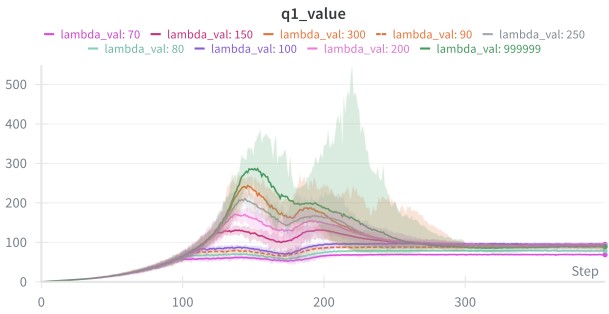

*Figure 11.* Illustration of the learning dynamics of Q-function estimation by DDQN and its clipped versions. DDQN avoids the overestimation issue. Moreover, the clipping operator tends to accelerate the convergence.

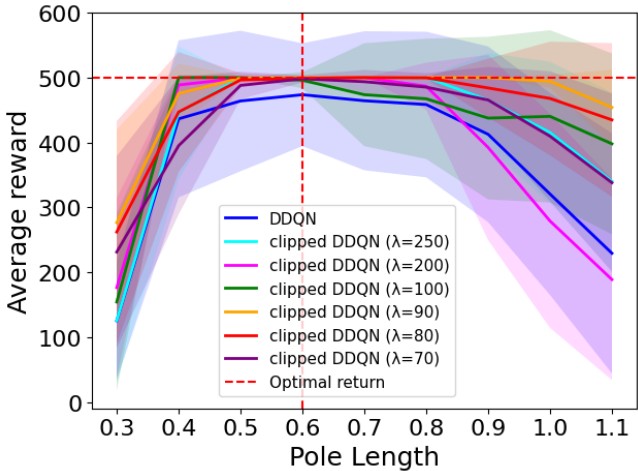

*Figure 12.* There is no action perturbation in the test environment. The optimal performances are achieved by clipped DDQN with $\lambda = 80, 90$. The Optimal return represents the largest cumulative reward that can be achieved at the test time. The vertical red line represents the pole length in the nominal environment. Shaded area represents the standard deviation.

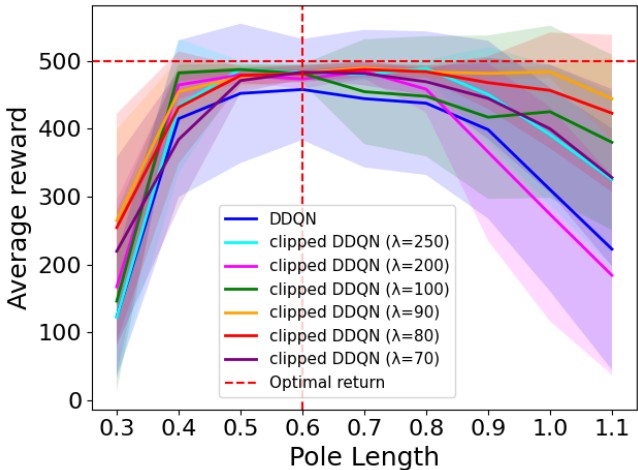

*Figure 13.* Action perturbation with probability 0.001. The Optimal return represents the largest cumulative reward that can be achieved at the test time. The vertical red line represents the pole length in the nominal environment. Shaded area represents the standard deviation.

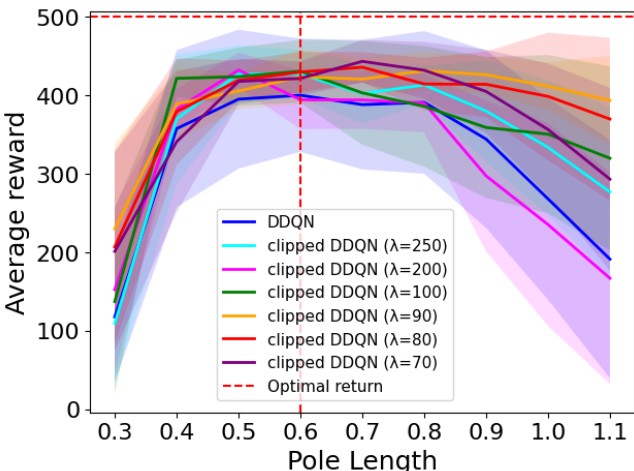

*Figure 14.* Action perturbation with probability 0.005. The Optimal return represents the largest cumulative reward that can be achieved at the test time. The vertical red line represents the pole length in the nominal environment. Shaded area represents the standard deviation.

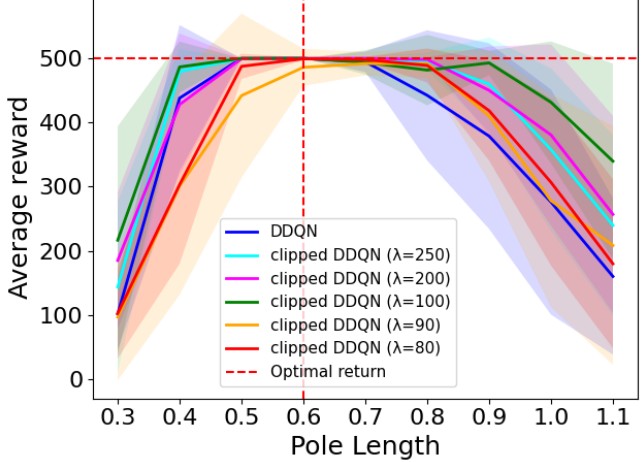

*Figure 15.* There is no action perturbation in the test environment.The Optimal return represents the largest cumulative reward that can be achieved at the test time. The vertical red line represents the pole length in the nominal environment. Shaded area represents the standard deviation.

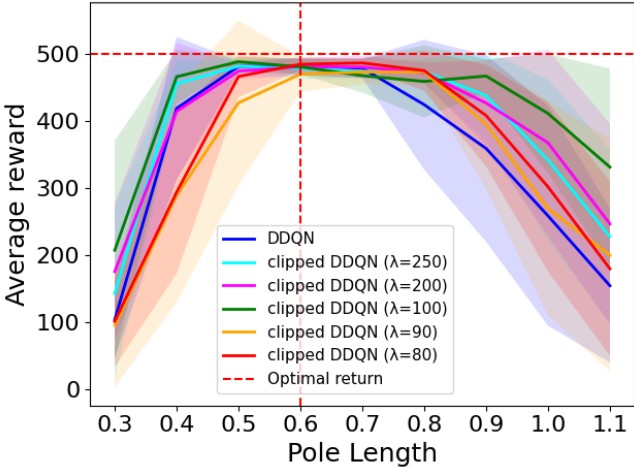

*Figure 16.* Action perturbation with probability 0.001.The Optimal return represents the largest cumulative reward that can be achieved at the test time. The vertical red line represents the pole length in the nominal environment. Shaded area represents the standard deviation.

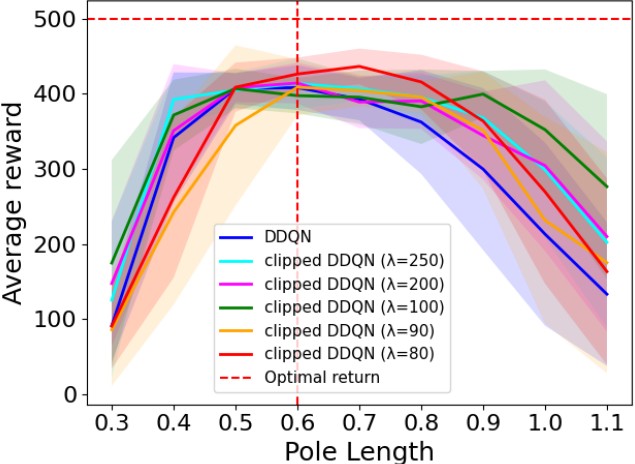

*Figure 17.* Action perturbation with probability 0.005.The Optimal return represents the largest cumulative reward that can be achieved at the test time. The vertical red line represents the pole length in the nominal environment. Shaded area represents the standard deviation.

