# OpenReview forum: "Clipped Q-Learning: Your Value Clipping Is Secretly A Robust Operator"
_ICML.cc/2026/Conference — ICML 2026 regular_

### Official Review · Reviewer_AG9y · 2026-02-16

**Soundness:** 3
**Presentation:** 3
**Significance:** 3
**Originality:** 3
**Overall Recommendation:** 5
**Confidence:** 3

**Summary:**

The paper proposes the *clipping Bellman operators* as the optimal solution to RRMDPs, where the adversary state dynamics is introduced by a soft TV regularization for discrete control. The authors conduct optimality analysis where the clipping backup & minimizer dynamics are identified, with a natural extension on the UCB variants under RRMDPs, along with corresponding regret guarantees. Experiments on the Inverted Pendulums are conducted to validate the performance of the new algorithm.

**Compliance With Llm Reviewing Policy:**

Affirmed.

**Final Justification:**

My main concerns have been addressed, and I am willing to recommend the paper for acceptance given its overall quality.

**Key Questions For Authors:**

Besides the aforementioned assessment, there are some additional questions that I have for the authors.

- - -

1. Theoretically, have the authors considered RRMDPs for continuous control, where the state dynamics necessitates more fine-grained continuous measure analyses?

- - -

2. Could the authors consider extending the experiments to more realistic control tasks to better assess the generalizability of their algorithms?

**Limitations:**

It is recommended that the authors include a standalone section discussing the limitations and future works.

**Strengths And Weaknesses:**

# **Strengths**

- - -

The readability of this paper is satisfactory, which features in clear and consistent formulas, a coherent logical structure, well-designed explanatory figures, and persuasive experiment results. The connection between soft adversary RRMDPs and the clipping robust operators is well revealed by the authors, which is both novel and natural according to my assessment. Readers may find it comparatively easy to grasp the flow of the paper, thanks to the appropriate examples, unambiguous mathematical definitions, and helpful theorems.

- - -

# **Weaknesses**

- - -

However, the following factors do affect my evaluation of the paper.

- - -

1. A more natural way to introduce the regularized variants of value functions is to start with an explicit objective $J(\pi)=\mathbb{E}\left[\sum \gamma^t \left[ r_t+\texttt{reg term}\right]\right]$ from which the optimal $Q, V$ functions are straightforwardly derived via dynamic programming. The authors are encouraged to adopt this presentation to enhance readability and ensure logical completeness.

- - -

2. From my perspective, Sec. 3.1 and Sec 3.2 largely cover the same optimization problem, where the former focuses on the optimal objective and the latter discusses the corresponding $\arg\inf$. It is recommended that the presentation be streamlined, and both sections be combined. The same applies to the paragraph “Healthcare Interpretation”, which seems to deviate somewhat from the main theoretical focus of the paper and may be condensed.

- - -

3. L324-328 (left column) uses the notation $\iota$, which is visually similar to the subscript of the previous character $H$. Perhaps a different symbol can be chosen to prevent such confusion.

- - -

4. In standard UCB analysis, the uncertainty term $b_t$ is decoupled from function updates, where $b_t$ is typically used to guide *action selection* (corresponding to Line 5 in Alg. 1, where the authors claim that the action is from $\arg\max Q$) rather than being directly incorporated into the *value update* itself (Line 7 in Alg. 1). Have the authors considered this standard approach, or alternatively, is there a particular motivation for including $b_t$ within the Bellman backup? Clarifying this design choice would help readers understand the implications for convergence and exploration.

- - -

On the whole, I believe that the aforementioned weaknesses shall be outweighed by all the merits of this paper. And it is recommended that the paper be considered for acceptance given its solidity.

---

> ### Author Rebuttal · Authors · 2026-03-31
>
> We thank the reviewer for your valuable time and effort in providing detailed feedback on our work. We hope our response will fully address all your questions.
>
> ---
> **Q1:**
> Start with an explicit objective $J(\pi)=\mathbb{E}[\sum \gamma^t[r_t+\textbf{reg term}]]$ to enhance readability and ensure logical completeness.
>
> **A1:** Thank you for the advice. With more space in the camera ready version, we can definitely adopt the presentation and include more details as suggested by the reviewer.
>
> ---
> **Q2:** Presentation of Section 3.
>
>
> **A2:** We will consider reorganize the Section 3, as currently it includes too many results and discussions as a single Preliminary Section. We might use a single section for Section 3.1 and Section 3.2 for better readability.
>
> ---
> **Q3:** Notation confusion of $\iota$.
>
> **A3:** We will use $\xi$ to replace $\iota$ to avoid any confusion.
>
> ---
> **Q4:** On the uncertainty term $b_t$.
>
> **A4:** This is an interesting question. The UCB estimation procedure follows a line of works on provably efficient RL [1,2,3], where the UCB term is incorporated into value function estimation. It is indeed different from the UCB used in Monte-Carlo Tree Search (MCTS), where UCB only helps action selection, rather than being incorporated into the value function update. The essence of incorporating UCB term in value function lies in the theoretical analysis, where we require the estimated Q-function $\hat{Q}^{k,\lambda}_h$ be an optimistic estimation of the true optimal Q-function $Q^{\star, \lambda}_h$, say, $\hat{Q}^{k,\lambda}_h(s,a) \geq Q^{\star, \lambda}_h(s,a)$ uniformly for all $(s,a,h)\in\mathcal{S}\times\mathcal{A}\times[H]$. We refer the reviewer to Lemma B.3 and Lemma B.7 for more details.
>
>
> ---
> **Q5:** Extending RRMDPs for continuous control, where the state dynamics necessitates more fine-grained continuous measure analyses; extending the experiments to more realistic control tasks to better assess the generalizability of their algorithms.
>
> **A5:** Extending RRMDPs to continuous state–action spaces would require appropriate function approximation techniques, along with new theoretical tools to handle the finer structure of continuous dynamics. At present, we do not have a complete answer to this question.
>
> We agree that evaluating our method on more realistic large-scale control tasks (e.g., Gym or DeepMind Control) would further strengthen the impact of this work. However, such extensions would entail substantial additional effort, including: identifying task-specific factors whose perturbations adversarially affect performance, selecting stable baseline methods that can handle high-dimensional or continuous action spaces, and systematically assessing whether the clipping module improves robustness under these perturbations.
>
> These challenges introduce significant experimental and methodological complexity that goes beyond the scope of this work. We therefore leave this direction for future study.
>
> ---
> **Q6:** limitation section
>
> **A6:** With more space for camera ready version, we will include a separate limitation section.
>
> While our work provides a principled connection between value clipping and TV-regularized robust MDPs, several limitations remain.
>
> First, our finite-sample analysis relies on a fail-state assumption to control the range of the value function and enable tractable regret bounds. While this assumption is standard in theoretical analyses, it may not hold or be easily verifiable in more general environments.
>
> Second, the robustness induced by the TV-RRMDP framework is tailored to a specific class of adversarial perturbations. In particular, it primarily hedges against “large-value” attacks, where probability mass is shifted toward low-value states. As a result, it does not capture more general forms of model misspecification or structured perturbations that may arise in practice.
>
> Third, although we provide preliminary empirical evidence in a simple control setting, it remains unclear when and how the clipping module behaves in more complex environments. In particular, the interaction between clipping and function approximation in deep RL is not yet well understood, and further empirical and theoretical investigation is needed to assess its effectiveness in high-dimensional settings.
>
> We leave these directions for future work.
>
> ---
> *References:*
>
> [1] Azar et al. Minimax Regret Bounds for Reinforcement Learning
>
> [2] Jin et al. Is Q-learning Provably Efficient?
>
> [3] Jin et al. Provably efficient reinforcement learning with linear function approximation
>
>
> ---
> We hope we have addressed all of your questions/concerns. If you have any further questions, we would be more than happy to answer them and if you don’t, would you kindly consider increasing your score?

---

> > ### Author Rebuttal · Reviewer_AG9y · 2026-04-01
> >
> > Thank you to the authors for your response. I would be willing to raise my evaluation to a score of 5.

---

> > > ### Author Response · Authors · 2026-04-07
> > >
> > > We are pleased that your previous concerns have been addressed. Thank you again for your support of our work.

---

### Official Review · Reviewer_vVvK · 2026-03-02

**Soundness:** 4
**Presentation:** 4
**Significance:** 3
**Originality:** 4
**Overall Recommendation:** 5
**Confidence:** 4

**Summary:**

This paper shows that clipping the next-state value in Q-learning's Bellman backup at threshold \(\lambda\) yields an unbiased estimator of a robust Bellman operator for TV-regularized robust MDPs (RRMDPs), where adversaries shift mass from high-value states to minimal-value ones.  It characterizes the worst-case kernel, proves \(\tilde{O}(\sqrt{SAH^3K})\) regret for UCB variants (improved by range shrinkage to \(\min\{1+\lambda,H\}\)), justifies fail-state assumption via info-deficit hardness, and demos clipped Double DQN robustness on Inverted Pendulum under dynamics shifts.

**Compliance With Llm Reviewing Policy:**

Affirmed.

**Key Questions For Authors:**

1. Prop. 3.6 assumes \(V_\text{min}=0\); general \(V_\text{min}\) kernel? Clarifies generality.
2. DDQN clip: optimal \(\lambda\) tuning? Ablation raises significance.
3. Beyond tabular/pendulum: Atari/DeepMind Control? Boosts impact.

**Limitations:**

Yes (info deficit, tabular focus); standard impact stmt.

**Strengths And Weaknesses:**

**Soundness**: Excellent. Clean duality (Props. 3.2/3.4) links clipping to TV-RRMDP; worst-kernel explicit (Prop. 3.6). Regret via novel tools (range shrinkage, perf-diff lemma). Hardness justifies Ass. 4.1. Expts confirmatory (MDP sim, DDQN robust w/o nominal loss).

**Presentation**: Excellent. Crisp: clipping → duality → kernel → algs → hardness → expts. Toy ex. intuitive (healthcare interp.). Alg. 1 repro.; apps detailed.

**Significance**: Good. Modular clip for robust Q-learning; range shrinkage stat-efficient. Relevant for dynamics-uncertain domains (robotic/healthcare). Specialized (tabular+toy FA) but unlocks clip-as-robust-operator lens.

**Originality**: Excellent. Novel: general-\(\lambda\) clip = TV-RRMDP solver (vs. prior ceiling-only). Worst-case mass-shift interp. fresh; bridges Q-learning + RRMDPs.

---

> ### Author Rebuttal · Authors · 2026-03-31
>
> We thank the reviewer for your valuable time and effort in providing detailed feedback on our work. We hope our response will fully address all your questions.
>
>
> ---
> **Q1:** Prop. 3.6 assumes ($V_\text{min}=0$); general ($V_\text{min}$) kernel? Clarifies generality.
>
>
>
>
> **A1:** Proposition 3.6 does not assume $V_{\text{min}} = 0$ and applies to a general $V_{\text{min}}$. In contrast, Theorems 4.2 and 4.3 rely on Assumption 4.1, which requires $V_{\text{min}} = 0$, and the justification for this assumption is provided in Section 4.1--without this assumption learning can be exponentially hard or imporssible due to an information deficit issue, and this assumption mitigate the issue. It remains an open question whether alternative conditions, other than Assumption 4.1, can guarantee learnability.
>
>
> ---
> **Q2:** DDQN clip: optimal (\lambda) tuning? Ablation raises significance.
>
>
> **A2:** We tune the optimal $\lambda$ based off of the following procedure. First we train a vanilla DDQN, and observe the range of the learned value function on historical states, then we choose several $\lambda$ accross the range. Second, we train the clipped DDQN with various chosen $\lambda$ and obtain learned robust policies. Third, we test robust policies in various perturbed environment, and get the one that achieve overall best performance. In the inverted Pendulumn experiment, the optimal $\lambda = 90,100$. We did a bunch of ablation study in **Appendix D** to test different choices of $\lambda$, choice of the baseline algorithm DQN/DDQN, and other perturbations, etc.
>
>
> ---
> **Q3:** Beyond tabular/pendulum: Atari/DeepMind Control? Boosts impact.
>
> **A3:** We agree that including large-scale deep RL experiments (e.g., Atari or DeepMind Control) would further strengthen the impact of this work. However, such extensions would require substantial additional effort, including but not limited to: identifying task-specific factors whose perturbations adversarially affect performance, selecting stable baseline methods that can handle high-dimensional or continuous action spaces, and systematically evaluating whether the clipping module effectively improves robustness under these perturbations.
>
> These challenges introduce significant experimental and methodological complexity that goes beyond the scope of this work. We therefore leave this direction for future study.

---

> > ### Author Rebuttal · Reviewer_vVvK · 2026-04-03
> >
> > The writer addressed all the claims

---

> > > ### Author Response · Authors · 2026-04-07
> > >
> > > We are pleased that your previous concerns have been addressed. Thank you again for your support of our work.

---

### Official Review · Reviewer_NdRf · 2026-03-14

**Soundness:** 1
**Presentation:** 2
**Significance:** 1
**Originality:** 1
**Overall Recommendation:** 2
**Confidence:** 5

**Summary:**

The paper introduces clipped-Q learning, an approach that clips Q-values when above a certain threshold. Previous work uses a heuristic approach to clip Q-values within the maximal range ($R_{\max}/(1-\gamma)$). This work analyzes the theoretical guarantees induced by clipping when the value is arbitrary, specifically when it is below the maximal range of Q-values. It provides an equivalent formulation with TV-distance regularized robustness, two variants of Clipped-Q learning with UCB bonus, and their regret bounds. A double-DQN variant of clipped-Q-learning is then run on the Inverted Pendulum.

**Compliance With Llm Reviewing Policy:**

Affirmed.

**Key Questions For Authors:**

- How does robustness with respect to initial state distribution induce robustness to transition probability, according to the authors? If it does.

- How do the authors understand the notion of distributionally robust MDPs? Based on [3], these formulate a minimum over distributions of transition kernels, i.e., $\min_{\mu} \mathbb{E}_{P\sim\mu}[v^{\pi}_{P}(s)]$, not over transitions $\min_P$.

- How do the authors map the correspondence between value functions in RRMDPs and Prop. 3.4? On my end, I do not see any minimization on the transitioning states in Prop 3.4., as opposed to RRMDPs that minimize value under the worst next state transition.

**Limitations:**

The contributions of this work are overstated and mischaracterized. The robustness guarantee arising from clipping is with respect to the initial state distribution (not the transition probability or the rewards).

The authors adequately discuss the limited numerical experiments in the conclusion, stating their intention to run more extensive experiments. Given the incremental contribution of value clipping (which itself has already been done heuristically with value upper bound), it would have been more valuable to see extensive experiments, analyzing the empirical value in clipping for low $\lambda$-values.

**Strengths And Weaknesses:**

**Strengths**
- The paper is written clearly and is easy to follow.

**Weaknesses**

*Soundness*
- Several relevant works are missing [1, 2, 3, 4], but more importantly, there seems to be confusion between regularized, robust, distributionally robust, and what uncertain variable the optimization problem aims to be robust to. Specifically, Props. 3.2 and Prop. 3.4 highlight that clipping leads to robustness with respect to the initial state distribution, which is very different from the transition robustness claimed in the introduction and throughout the paper.

*Significance*
- The paper lacks some practical motivation as to why clipping is necessary or useful, algorithmically or theoretically. It sounds more like it starts from an empirical tweak that the authors tried to elaborate upon. The theoretical study seems to include several misunderstandings, and the empirical experiments are not sufficiently convincing to assess the usefulness of clipping. More exhaustive experiments on different large environments would have been helpful here.

*Originality*
- Props 3.2 and 3.4 appear to follow straightforwardly from Tang et al (2025). Indeed, their assumption that the initial state distribution is fully supported on $\mathcal{S}$ does not lose generality as long as the optimization focuses on its support (Prop 3.2).

*Presentation*
- Section 3 is very long, even more so for a Preliminary one.
- Fig 4 shows that $\lambda = 90$ is less robust than $\lambda = 100$, which seems to contradicts the authors' claim.

[1] Geist, Matthieu, Bruno Scherrer, and Olivier Pietquin. "A theory of regularized Markov decision processes." International conference on machine learning. PMLR, 2019.
[2] Derman, Esther, Matthieu Geist, and Shie Mannor. "Twice regularized MDPs and the equivalence between robustness and regularization." Advances in Neural Information Processing Systems 34 (2021): 22274-22287.
[3] Xu, Huan, and Shie Mannor. "Distributionally robust Markov decision processes." Advances in Neural Information Processing Systems 23 (2010).
[4] Derman, Esther, and Shie Mannor. "Distributional robustness and regularization in reinforcement learning." arXiv preprint arXiv:2003.02894 (2020).

---

> ### Author Rebuttal · Authors · 2026-03-31
>
> We thank the reviewer for your valuable time and effort in providing detailed feedback on our work. We hope our response will fully address all your questions.
>
> ---
> **Q1:** Initial state distribution robustness v.s. transition perturbation robustness
>
> **A1:** This point reflects a **key misunderstanding** that affects several conclusions in the review. In the RRMDP framework, robustness is formally defined in Eq (3.1) and (3.2), where the uncertainty is explicitly **over transition probabilities at every stage**. In contrast, Propositions 3.2 and 3.4 which were referred to by the reviewer are **general duality results** that serve as analytical tools; they are applied to the robust Bellman optimality equation (3.5) **at each stage $h\in[H]$** as discussed above Definition 3.3.
>
> To make this concrete, fix a $h\in[H]$, the $\mu$ in Proposition 3.2/3.4 corresponds to $P'(\cdot)$ in (3.5), and $V$ corresponds to $V^{\star,\lambda}_{h+1}$. Applying Proposition 3.2/3.4 to (3.5), we have the simple closed form solution for robust Bellman optimality equation at each stage $h\in[H]$:
> $Q _h^{\star,\lambda}(s,a) = r _h(s,a) + \mathbb{E} _{s'\sim P^0 _h(\cdot|s,a)}[V _{h+1}^{\star, \lambda}(s')] _{V^{\star,\lambda} _{h,\min} + \lambda}.$
>
> Under this correspondence, the infimum in Propositions 3.2/3.4 precisely captures adversarial perturbations over next-state transitions. Therefore, the resulting robustness is with respect to **transition dynamics**, consistent with the claims throughout the paper.
>
> ---
> **Q2:** Motivation of the clipping modular, and contribution of this paper
>
> **A2:** The clipping module is rigorously motivated by the TV-regularized RRMDP framework, through the closed-form robust Bellman optimality equation, as discussed in **A1**. It is **not** merely an empirical heuristic. Our work is the **first** to establish a principled connection between RRMDPs and the clipping behavior in the foundational Q-learning algorithm, which constitutes the main contribution of this work. Our experiments are specifically designed to validate this theoretical link and provide strong empirical support for our claims. Conducting large-scale deep RL experiments would require substantial additional effort and is beyond the scope of this work; we leave this direction for future study.
>
> ---
> **Q3:** Originality of Proposition 3.2 and 3.4.
>
> **A3:** While Tang et al. (2025) present results with a similar high-level idea to Propositions 3.2 and 3.4, their analysis is not fully rigorous and implicitly relies on a critical absolute continuity assumption. In contrast, we provide a **completely new and rigorous proof**, and further **extend** the results to settings that allow support shifts, supported by **new theoretical analysis**. We emphasize that Propositions 3.2 and 3.4 serve as analytical tools rather than the main contributions of the paper. Beyond these results, we also characterize the worst-case behavior of both the transition kernel and the policy in Section 3.2, two variants of clipped Q-learning algorithms and their finite sample analysis, which, to the best of our knowledge, are novel and not addressed in prior work.
>
> ---
> **Q4:** Figure 4 show $\lambda=90$ is less robust than $\lambda=100$, which seems to contradicts the authors' claim.
>
> **A4:** This does **not** contradict our claim, as we explicitly state in Line 73-76 "We equip Double DQN with the value-clipping module and evaluate it **heuristically** on the classical control task Inverted Pendulum." This experiment is **heuristic** because DDQN involves neural network for general function approximation, which does **not** fall into our analysis regime. Note that the behavior of  deep neural network in deep RL remains not fully understood, so we should **not** expect it to perfectly obey analysis in the tabular setting. For example, the learning dynamics in Figure 10 shows that the learned value functions in the middle of training can significantly exceed the theoretical ceiling 100 due to overestimation. And clipping indeed happens in the middle with theoretical ceiling $\lambda=100$. That said, our empirical results still support the main claim: moderate values of $\lambda$ improve robustness compared to no clipping.
>
> ---
> **Q5:** Understanding the distributionally robust MDPs.
>
> **A5:** We do **not** adopt the same definition of distributionally robust MDPs as in [3]. As shown in Eq (3.1) and (3.2) of our paper, we focus on a single worst-case transition kernel—equivalently, a Dirac distribution over the set of admissible kernels. This formulation corresponds to the classical robust MDP framework, which has been extensively studied in the literature. We refer the reader to the related work section for a more detailed discussion.
>
> ---
> We respectfully request the reviewer to reconsider the points above, correct any misunderstandings, and re-evaluate our paper accordingly to provide a fair and reasonable assessment. We welcome any further questions.

---

> > ### Author Rebuttal · Reviewer_NdRf · 2026-04-03
> >
> > The authors clarified a misunderstanding I had on their Props. 3.2 and 3.4. To avoid further confusion, it would be helpful to state the final result in terms of the next state transition instead of keeping it with $s\sim \mu$. This refactorization is part of the proof in my opinion, not the final result.
> >
> > I do understand that neural network approximation can break some theoretical guarantees, at least partially. It is precisely for this reason that I was suggesting the authors to investigate other domains more exhaustively, confirming whether their theoretical insight consistently holds despite NN approximation. Numerical experiments are still limited in that respect.
> >
> > The originality of this work is still overstated in my opinion  with many similar works that have not been discussed in the rebuttal (+ in the paper). I deplore the tone adopted by the authors to address my concerns, which aimed to initiate a constructive discussion rather than shutting my questions down.

---

> > > ### Author Response · Authors · 2026-04-07
> > >
> > > We thank the reviewer for acknowledging the misunderstanding. Due to space limit in the first round rebuttal, we were not able to provide a detailed discussion on those related works. Thanks for bring them to our attention, we will add the following discussion in the related work section of the paper.
> > >
> > >
> > > **Regularized MDPs** There is a rich body of work on regularized Markov Decision Processes. In particular, [1] develops a general theoretical framework for regularized MDPs, with a focus on policy regularization. Building on this line of work, [2] shows that policy-regularized MDPs are equivalent to reward-robust MDPs, and further establishes that MDPs with both transition and reward uncertainty sets can be reformulated as MDPs with policy and value regularization. In a related direction, [3] studies Wasserstein distributionally robust MDPs and characterizes a dual relationship between robustness to model uncertainty and value function regularization. Our work is orthogonal to these prior studies. Rather than focusing on policy or value regularization, we investigate transition-regularized MDPs in the online setting, and propose a robust variant of Q-learning with finite-sample guarantees.
> > >
> > > To avoid term conflict, we change **Distributionally Robust MDPs** to **Distributionally Robust RL**, and replace "distributionally robust Markov (DRMDP)" with "distributionally robust MDP (RMDP)" as is typically done in prior works on RMDPs. We will cite [4] in this revised related work section.
> > >
> > > ---
> > > ###
> > > We agree that including additional large-scale deep RL experiments (e.g., Atari or DeepMind Control) would further strengthen the impact of this work. However, such extensions would require substantial additional effort, including (i) identifying task-specific factors whose perturbations adversarially affect performance, (ii) selecting stable baseline methods that can reliably operate in high-dimensional or continuous action spaces, and (iii) systematically evaluating whether the proposed clipping module improves robustness under these perturbations. These challenges introduce significant experimental and methodological complexity that goes beyond the scope of the current paper.
> > >
> > > Instead, this work focuses on establishing a principled theoretical foundation. In particular, we characterize the worst-case behavior of TV-regularized RRMDPs, formalize the connection between value clipping in Q-learning and TV-regularized RRMDPs, and propose two variants of Q-learning algorithms with finite-sample guarantees. We believe these contributions are substantial on their own, and we leave the exploration of larger-scale domains to future work.

---

### Official Review · Reviewer_77Yf · 2026-03-15

**Soundness:** 3
**Presentation:** 4
**Significance:** 3
**Originality:** 3
**Overall Recommendation:** 4
**Confidence:** 4

**Summary:**

The authors analyze Clipped Q-learning, an algorithmic variant of Q-learning which upper bounds the bootstrap of the 1-step backup sample by a chosen value $\lambda > 0$. They derive a connection to Regularized Robust MDPs (RRMDPs) using the Total Variation (TV) distance, showing theory that describes the robust Bellman operator and its regret bounds with different variations of UCB exploration. Finally, some small experiments are conducted to compare the robustness of clipped Q-learning to the standard baseline, by perturbing tabular MDPs and plotting the average reward.

**Compliance With Llm Reviewing Policy:**

Affirmed.

**Final Justification:**

After much deliberation, I have decided to keep my original score. I still lean toward acceptance, but the following concerns remain:
- The requirement of nonnegative rewards is a significant limitation in my opinion, as negative rewards are very common in practice despite what the authors say. It's therefore unclear if the proposed robust-operator theory holds generally or only under favorable conditions. The authors **must** disclose this more explicitly.
- The convergence analysis is unidimensional in the sense that it only addresses regret rate and not other trade-offs (e.g., suboptimality), and therefore strongly implies that clipping is uniformly good. As discussed with the authors, this is an overly simplified explanation which makes clipped Q-learning sound better than the reality. The authors have agreed to tailor this claim accordingly.
- The $\lambda$ hyperparameter is very brittle. Although it is true that deep RL already has many hyperparameters, most are relatively robust and transferrable across environments (e.g., minibatch size, learning rate, exploration). This does not seem to be the case for $\lambda$ here, which will be a pain point when trying to apply the insights from this paper.

None of these are too catastrophic, but all are not easily addressed without major changes.

The paper is still conceptually interesting in my opinion. The analysis and discussions are thought-provoking, stimulating, and worth sharing with the ICML community. For this reason, I still believe the paper can be accepted, as long as the claims are adjusted, as discussed in the rebuttal.

**Key Questions For Authors:**

Nothing additional. Please just respond to the Weaknesses and Limitations.

**Limitations:**

- As mentioned in the weaknesses, the trade-off between convergence speed and quality is not sufficiently discussed. This is the biggest limitation of the paper overall in my opinion.
- The analysis only pertains to $\gamma=1$ in episodic tasks. Why?
- Rewards appear to be bounded in $[0,1]$. Why? Since these are episodic tasks, negative rewards can change optimal behavior.

**Strengths And Weaknesses:**

## Strengths

- The theory reveals an interesting connection between a simple modification to a classic algorithm (Q-learning) and RRMDPs. I am not an expert in the latter area, but I reviewed the results in Section 3 closely and they appear to be correct. Connecting these two disparate areas helps to interpret and justify an otherwise seemingly unprincipled approach, leading to new insights and potentially new algorithms.
- The paper has great descriptive analysis intermixed with the theoretical results. The authors spent the necessary time to explain the setup, intuition, and significance of the results. This makes the theoretical discussions highly engaging.
- There are lots of provided examples and small experiments to add clarity and offer extra evidence that supports the theoretical claims.

## Weaknesses

- Clipping introduces significant bias into the value estimation, degrading policy performance by precluding convergence to $Q^*$. The obtained policy is only optimal in the reformulated RRMDP problem with “fictitious” nominal transition kernels, but is suboptimal in the original problem. This is a nontrivial trade-off which I think should be made clear in the paper.
- The *Healthcare Interpretation* example is nice for understanding how clipping could be more robust, but one could argue that the reward function is simply ill-specified for this problem. A dying patient should probably yield a very negative reward instead of just 0. This does not mean Q-learning isn’t robust, it just means that failure is not sufficiently punished in this problem definition. We could easily construct many dual-reward counterexamples where standard Q-learning outperforms clipped Q-learning. Does this mean classic Q-learning is more robust? The nuance must be clarified.
- The theoretical results in Section 4 are slightly misleading, I think. The paper claims the convergence rate is improved by a factor of $H / \min(1 + \lambda, H)$ with respect to a regret bound, where $H$ is the horizon. This implies we could just set $\lambda=0$ to achieve the fastest convergence rate, but this would degenerate to $Q(s,a) = r(s,a)$ for all $(s,a)$, which obviously would not make sense. The theorems appear to be correct, but I think the nuance of the interpretation needs to be explicitly clarified. In reality, there is a major trade-off between convergence rate and convergence quality which is not highlighted by the paper, as I mentioned in the previous two points.
- The choice of $\lambda$ is not scale invariant and is highly dependent on a given reward function. This means it probably does not generalize well across different environments and may require tuning per environment. It would be nice to have some more guidance for choosing this value. The experiments seem to just show a few hand-picked values between 0 and 1, which happen to be appropriate for the toy examples with bounded returns in $[0,1]$, but I do not recall seeing a strong motivation for them.

## Additional Edits (did not impact score)

- The experiment in 5.1 could use more details. How long are the rolled out trajectories? Are the agents retrained for each value of $q \in [0,1]$ or just trained on one and evaluated on the spectrum? I don’t think I could reproduce these results from the current description.
- Ideally $\lambda$ would not be used for the clip value as this is commonly reserved for TD($\lambda$) algorithms in RL, especially in the context of value-function operators like $\mathcal{T}^\lambda$. Perhaps a less common symbol can be used instead?
- I am torn by the equations in the introduction. On one hand, they help clarify the authors’ intended direction for analysis immediately. On the other, notation has not been properly set up and defined at this point, which makes the discussion less precise than it should be (e.g., MDP notation hasn’t even been introduced yet). Preferably, the introduction would offer a high-level intuition for the problem and planned contributions, and then these equations could be clearly defined in the background with an appropriate amount of space and rigor.
  - As an example, Eq. 1.2 is technically incomplete because it only shows the partial update for the taken state-action pair $(s,a)$. The other values must be carried over too to be complete: i.e., $Q_{t+1}(s’,a’) = Q_t(s',a'), \forall (s’,a’) \neq (s,a)$. This kind of implied understanding makes the paper less rigorous and less accessible to a wide audience.
- It would be nice to add a figure for Example 3.7.
- Line 77, left: citations are double parenthesized.
- Line 94, left: Did not capitalize Bellman.

---

> ### Author Rebuttal · Authors · 2026-03-31
>
> We thank the reviewer for your valuable time and effort in providing detailed feedback on our work. We hope our response will fully address all your questions.
>
> ---
> **Q1:** The robust policy is not optimal in the nominal env.
>
> **A1:** We added clarification: The optimal policy $\pi^{\star,\lambda}$ is the solution to the max-min optimization problem and achieves optimal performances in the worst-case environment. However, $\pi^{\star,\lambda}$ may sacrifice performance under the nominal dynamics in order to hedge against worst-case deviations, reflecting an inherent trade-off between robustness and nominal optimality.
>
> ---
> **Q2:** $[0,1]$-bounded reward.
>
> **A2:** We restrict the discussion in the $[0,1]$-bounded reward regime. This is a standard assumption in RL theory [1,2] to simplify analysis and ensure bounded value functions. It can be obtained without loss of generality via appropriate rescaling of arbitrary bounded reward in real applications. The illustrative toy Example 3.7, is also designed to fit into this regime. We agree it does not perfectly reflect the actual healthcare setting, but it suffices to illustrate the idea--robust policy can be different from standard optimal policy.
>
> Relaxing the assumption may indeed alter the optimal and robust behaviors, as the reviewer pointed out. At this stage, we yet have a precise characterization on how the results would extend beyond this setting, and we consider this an interesting direction for future work. Nevertheless, in many real-world applications, one can construct an equivalent MDP through appropriate normalization so that the rewards lie within $[0,1]$.
>
>
> ---
> **Q3:** On convergence speed and convergence quality.
>
> **A3:** The reviewer is correct that a smaller $\lambda$ can lead to faster convergence, but typically at the cost of poorer performance. This trade-off is illustrated in Figures 7 and 10, where the clipped DDQN converges more quickly with smaller thresholds $\lambda$, while the best robust performance is achieved at moderate values such as $\lambda=90,100$. We add the following paragraph:
>
> Theorem 4.2 suggests that a smaller $\lambda$ leads to lower sample complexity. In the extreme case of $\lambda=0$, no learning is required, since the worst-case kernel deterministically transits to state with smallest reward. However, such overly conservative behavior will result in poor performance. In practice, one should **prioritize** selecting an appropriate $\lambda$ that balances robustness and performance, rather than sacrificing performance solely for faster convergence.
>
> ---
> **Q4:** Guidance on choosing $\lambda$.
>
> **A4:** $\lambda$ is a tuning parameter in our clipped Q-learning and should be selected on a per-task basis. To determine an optimal $\lambda$, we propose the following two-step procedure.
>
> First, train a standard Q-learning (or DDQN) algorithm until convergence. Use the learned value function to evaluate states in the replay buffer. This yields an empirical distribution of values over historical states. Second, select percentiles of this distribution—e.g., 50%, 70%, and 90% percentiles—as candidate values for $\lambda$. Then, train clipped Q-learning with each candidate $\lambda$ and evaluate the resulting robust policies under various perturbed environments. This evaluation step is typically efficient, as it does not require significant additional data compared to training. The final choice of $\lambda$ can then be made based on the best empirical performance across these evaluations.
>
> ---
> **Q5:** Extension to the infinite-horion discounted setting.
>
> **A5:** Just like the analysis of standard Q-learning with UCB [1,3], our results can potentially be extended to the infinite-horizon discounted setting with a general discount factor $\gamma \in [0,1]$. However, such an extension would require additional technical developments and is beyond the scope of this work. We therefore leave this direction for future study.
>
> ---
> **Q6:** Minor issues.
>
> * Each evaluation consists of 500 rollout episodes. For each $\lambda$, the agent is trained once, and the learned policy is then evaluated across a range of perturbed environments. We have included our code in the supplementary material, which can reproduce all experimental results.
>
> * Change $\lambda$ to $\alpha$.
>
> * Make the equations in the introduction more complete and rigorous. At the same time, we expect readers to have basic familiarity with the foundational Q-learning algorithm.
>
> * Figure 1 is for Example 3.7.
>
> * We have revised all typos.
>
> ---
> *References:*
>
> [1] Jin et al., Is q-learning provably efficient?
>
> [2] Agarwal et al., Reinforcement Learning: Theory and Algorithms.
>
> [3] Dong et al., Q-learning with UCB Exploration is Sample Eﬃcient for Infinite-Horizon MDP.
>
> ---
> We hope we have addressed all of your questions/concerns. If you have any further questions, we would be more than happy to answer them and if you don’t, would you kindly consider increasing your score?

---

> > ### Author Rebuttal · Reviewer_77Yf · 2026-04-05
> >
> > Thanks for the authors' response. Several of my concerns have been resolved but there are still some significant issues that I believe need to be addressed:
> >
> > ---
> >
> > **Q2:** [0,1]-bounded reward.
> >
> > > We restrict the discussion in the [0,1]-bounded reward regime. ... It can be obtained without loss of generality via appropriate rescaling of arbitrary bounded reward in real applications.
> >
> > I am not sure about this. As I said previously, under episodic termination, rescaling the reward function *does* change the optimal policy, and so I believe there would be loss of generality. This explains, for example, why the solution to the Healthcare Interpretation changes when negative rewards are allowed.
> >
> > Unless "in real applications" is liberally interpreted to mean "nonnegative rewards," I don't see how this is true.
> >
> > ---
> >
> > **Q3:** On convergence speed and convergence quality.
> >
> > > Theorem 4.2 suggests that a smaller $\lambda$ leads to lower sample complexity. In the extreme case of $\lambda=0$, no learning is required, since the worst-case kernel deterministically transits to state with smallest reward. However, such overly conservative behavior will result in poor performance. In practice, one should prioritize selecting an appropriate $\lambda$ that balances robustness and performance, rather than sacrificing performance solely for faster convergence.
> >
> > This is better, but consequently it is quite misleading to claim convergence rate improvements when there is such a critical trade-off here. In reality, as you said, convergence rate must be balanced with other considerations, but the theory does not yet capture this. If, for example, the theory could describe the error (bias) at convergence due to the clip operator, then the trade-off could be better characterized and the paper would be stronger.
> >
> > ---
> >
> > **Q4:** Guidance on choosing $\lambda$.
> >
> > > $\lambda$ is a tuning parameter in our clipped Q-learning and should be selected on a per-task basis. To determine an optimal $\lambda$, we propose the following two-step procedure.
> >
> > > First, train a standard Q-learning (or DDQN) algorithm until convergence. Use the learned value function to evaluate states in the replay buffer. This yields an empirical distribution of values over historical states. Second, select percentiles of this distribution—e.g., 50%, 70%, and 90% percentiles—as candidate values for $\lambda$. Then, train clipped Q-learning with each candidate $\lambda$ and evaluate the resulting robust policies under various perturbed environments. This evaluation step is typically efficient, as it does not require significant additional data compared to training. The final choice of $\lambda$ can then be made based on the best empirical performance across these evaluations.
> >
> > This procedure sounds quite expensive, especially since it must be repeated for each task. I do think this is a significant weakness of the method. Training multiple agents just to select one hyperparameter, independently in every new environment, undermines much of the motivation for applying robust MDP methods to safety-critical domains. This maybe works when the environment can be simulated and the deployed policy remains static, but it limits practicality and generality.
> >
> > ---
> >
> > **Q6:** Minor issues.
> >
> > > We have included our code in the supplementary material, which can reproduce all experimental results.
> >
> > Sharing code is good, but ideally the paper would be self-contained and descriptive enough to reproduce the results on its own.
> >
> > Code can also change over time; in the future, it may become unclear which version of the code was used to generate results. The paper serves as documentation to verify that the code is correct.
> >
> > > Change $\lambda$ to $\alpha$.
> >
> > $\alpha$ may actually be more problematic than $\lambda$ because it's used for the step size, which is very common. If you must use $\lambda$, then it is probably fine in the context of this paper, but I do recommend considering a less common symbol.

---

> > > ### Author Response · Authors · 2026-04-07
> > >
> > > Thank you for the detailed feedback, we provide further discussion in the following.
> > >
> > > ---
> > > **On Q2:**  To clarify the $[0,1]$ reward setting, we mean first normalize the reward to lie in $[0,1]$, then our theoretical analysis applies. As for general (possibly negative) reward functions, it is true there is no closed answer yet.
> > >
> > > ---
> > > **On Q3:** We adopt the reviewer's advice and will consider removing the claim of convergence rate improvement, since it might be misleading to practitioners. The current results only focus on showing how fast the clipped Q-Learning algorithm with a particular $\lambda$ converges to the true optimal robust policy $\pi^{\star,\lambda}$ corresponding to this $\lambda$, by providing an upper bound on the regret defined as $\text{Regret(K)}=\sum_{k=1}^K [V^{\star,\lambda}(s^k_1) - V^{\pi^k,\lambda}(s^k_1)]$. While in practice the $\lambda$ should be carefully chosen based on the extend of perturbation the agent needs to hedge against with. Probably the reviewer's idea can be formulated as this: if the oracle threshold (the threshold that perfectly captures the worst case perturbation without being over-conservative) is $\lambda_1$ while we chose $\lambda_2$ (might be smaller or larger than $\lambda_1$), how to quantify the new performance gap, which might be defined as something like $V^{\star, \lambda_1}(s^k_1) - V^{\pi^k(\lambda_2),\lambda_1}$. This is an interesting point to investigate in the future, thanks for bringing it up.
> > >
> > >
> > > ---
> > > **On Q4:** Standard deep RL algorithms also need quite large scale hyperparameter sweeping, and our algorithm only introudce one more hyperparameter to tune compared to the baseline deep RL algorithms (like DDQN). That said, we do agree in safety-critical domains this could be problematic. We will discuss this in limitation section.
> > >
> > > ---
> > > **On Q6:** Yes, we will add more details on experiment setup. In terms of the threshold notation, we will use $\beta$ for the threshold.

---

### Decision · Program_Chairs · 2026-04-30

**Decision:**

Accept (regular)

**Comment:**

The reviews were mixed, but on balance I recommend weak accept. Reviewers generally agreed that the paper offers an interesting and technically solid perspective on value clipping, with a clean connection to robust MDPs and meaningful theoretical development. The main concerns are about practical scope and positioning: the empirical study is limited, the clipping threshold appears brittle and task dependent, and the paper should be much clearer about the tradeoff between robustness, nominal performance, and regret guarantees. One negative review also raised concerns about the technical interpretation and originality, though the discussion clarified an important misunderstanding and narrowed the disagreement mostly to positioning and scope. On balance I think the theoretical contribution clears the bar, but the final version should sharpen the claims, discuss limitations more candidly, and improve engagement with related work.